# Structural dynamics of human fatty acid synthase in the condensing cycle

Wooyoung Choi[1,4], Chengmin Li[1,4], Yifei Chen[2,3], YongQiang Wang[2] & Yifan Cheng[1,2 ✉]

Long-chain fatty acids are the building blocks of fat in human bodies. In mammals, fatty acid synthase (FASN) contains multiple enzymatic domains to catalyse all chemical reactions needed for de novo fatty acid synthesis[1]. Although the chemical reactions carried out by these enzymatic domains are well defined, how the dimeric FASN with an open architecture continuously catalyses such reactions to synthesize a complete fatty acid remains elusive. Here, using a strategy of tagging and purifying endogenous FASN in HEK293T cells for single-particle cryo-electron microscopy studies, we characterized the structural dynamics of endogenous human FASN. We captured conformational snapshots of various functional substates in the condensing cycle and developed a procedure to analyse the particle distribution landscape of FASN with different orientations between its condensing and modifying wings. Together, our findings reveal that FASN function does not require a large rotational motion between its two main functional domains during the condensing cycle, and that the catalytic reactions in the condensing cycle carried out by the two monomers are unsynchronized. Our data thus provide a new composite view of FASN dynamics during the fatty acid synthesis condensing cycle.

In mammals, fatty acids are essential components that can be synthesized de novo within cells or supplied from the diet. De novo syntheses are carried out by a large multidomain enzyme, FASN[1]. During development, FASN function is essential because *FASN* gene knockout causes embryonic lethality[2]. However, the expression level of *FASN* in adults remains low in most tissues owing to the dietary supply of fatty acids[3], except in adipocytes, glandular cells[4] and in numerous tumorigeneses including breast, prostate and lung cancers, in which high demand for lipid supply upregulates *FASN* expression levels[5–8]. In addition, enveloped viruses hijack endogenous FASN to aid viral replication[9,10]. Recent studies also show causality between FASN and metabolic diseases, immune responses and lipid-mediated neurodegenerative diseases[11–13].

The process of de novo fatty acid synthesis is chemically well defined and highly conserved through evolution. It comprises a set of recurring chemical reactions that can be simplified to two key processes: a decarboxylative Claisen condensation reaction to extend an acet(yl) moiety with two carbons supplied from a malonyl moiety resulting in β-ketoacyl, and a modifying process with three steps to produce a saturated acyl moiety that is ready to be elongated further. These processes are repeated multiple times until the substrate reaches the desired length. This conserved process is carried out by six to seven enzymes, including a phosphopantetheine (Ppant)-modified acyl carrier protein (ACP). The entire cycle of fatty acid synthesis requires that ACP carries the substrate from one enzyme to another repeatedly. In bacteria and plants, enzymes catalysing fatty acid synthesis are individual proteins. An ACP carries substrates from one enzyme to another, sequentially following the chemical reactions[14]. In metazoans, all essential enzymes are included in a single polypeptide chain and are arranged into two main functional domains, each carrying out a key chemical process and named correspondingly the condensing and modifying domains (Fig. 1a,b).

FASN functions as a homodimer with its condensing and modifying domains each forming a dimeric wing with the same domain of the other monomer, and the two wings are arranged in parallel with a short linker between them[15,16] (Fig. 1a). This architecture prompted a hypothesis that rotational motions between the two wings facilitate shuttling of ACP to different enzymatic domains[16]. Later structural studies of mammalian FASN revealed large rotational motions between the two wings[17,18], further promoting the notion that rotational motion and ACP shuttling are correlated[19,20]. However, in the absence of direct evidence that high conformational variability is required for FASN function, and that the crowded intracellular environment constrains protein dynamics[21], it remains unclear whether the large rotational motion between the two wings in FASN is indeed correlated with ACP shuttling.

Here we studied functional human endogenous FASN and characterized its conformational dynamics by single-particle cryo-electron microscopy (cryo-EM). By visualizing ACP engagements with either the β-ketoacyl synthase (KS) or malonyl/acetyl transferase (MAT) domains, we captured FASN at specific functional substates during the condensing cycle and revealed that the enzymatic reactions carried out by the two monomers in the condensing cycle are unsynchronized. We developed a procedure to analyse the particle distribution landscapes of relative orientations between the two wings, and reveal that the large motion is not required to shuttle substrate to different enzymatic domains by ACP during the condensing cycle.

[1]Department of Biochemistry and Biophysics, University of California San Francisco, San Francisco, CA, USA. [2]Howard Hughes Medical Institute, University of California San Francisco, San Francisco, CA, USA. [3]Present address: Department of Pharmaceutical Chemistry, University of California San Francisco, San Francisco, CA, USA. [4]These authors contributed equally: Wooyoung Choi, Chengmin Li. ✉e-mail: yifan.cheng@ucsf.edu

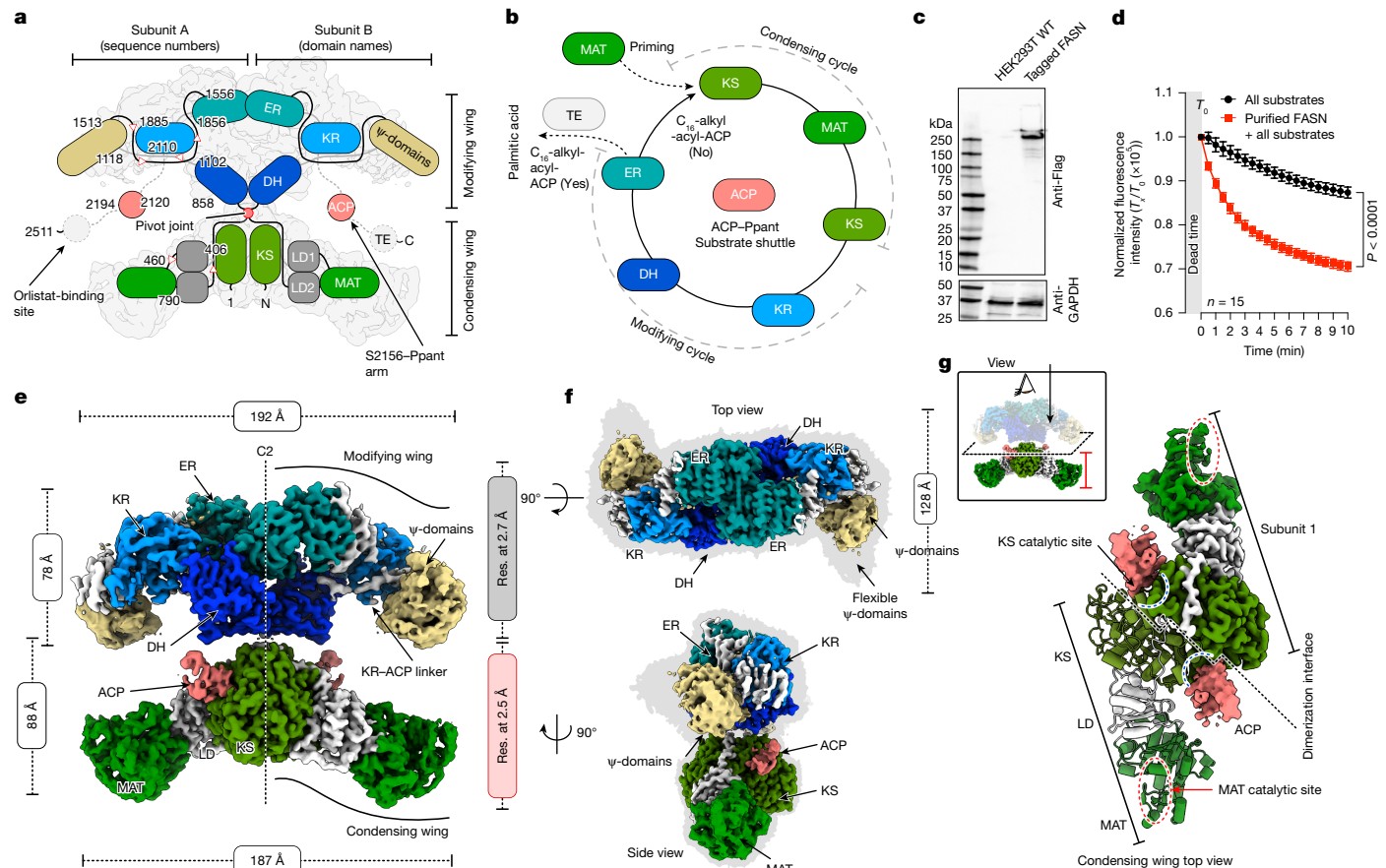

**Fig. 1 | Overall architecture of human endogenous FASN. a**, A schematic representation of the domain arrangement in the structure of human FASN without specifying a *trans* or *cis* arrangement of the two monomers in the dimer. Name (right subunit) and its corresponding amino acid sequence number (left subunit) for each domain are indicated. DH, dehydratase; ER, NADPH-dependent enoyl reductase; LD, linker domain. **b**, A schematic illustration of the catalytic cycle of palmitic acid synthesis, in which Ppant-modified ACP shuttles the substrates through different enzymatic domains. **c**, Anti-Flag immunoblotting reveals successful tagging in HEK293T cells. Anti-GAPDH immunoblotting serves as a loading control. Endogenous tagging following the established procedure[22] was performed three times. For raw data, see Supplementary Fig. 9. PAGE gel concentration gradient, 4–15%. WT, wild type.

**d**, NADPH consumption levels indicate the functionality of purified FASN. The graph shows normalized values from biological triplicate experiments. $T_0$ and $T_x$ indicate the starting (0 min) and measuring ($x$ min) point, respectively. The two-tailed unpaired parametric Student's $t$-test was used to calculate the $P$ value, indicated in the figure. The data are shown as the mean ± s.e.m. **e**, A composite cryo-EM map of human endogenous FASN with ACP bound to the KS domain. Domains are coloured as in **a**,**b**. Res., resolution. **f**, Top (upper panel) and side (bottom panel) views of the composite map. **g**, Top view of the condensing domain alone, with the top panel illustrating the viewing orientation. From this view, the catalytic sites of the KS and MAT domains are in opposite sides of the condensing domain. An ACP is attached to the catalytic site of the KS domain and the loop of the linker domain is well resolved.

## Structure of human endogenous FASN

To obtain functional FASN with Ppant-modified ACP, we tagged the carboxy terminus of FASN in HEK293T cells[22], and purified full-length human endogenous FASN by tandem affinity purification followed by size-exclusion chromatography (Fig. 1c and Supplementary Figs. 1 and 2). Mass spectrometry analysis confirms that purified FASN is full length with major post-translation modifications including Ppant modification on S2156, ensuring that FASN contains holo-ACP (Supplementary Fig. 3). Purified FASN is enzymatically active, it consumes NADPH in the presence of acetyl-CoA and malonyl-CoA and it produces an intermediate acyl chain at the tip of Ppant (Fig. 1d and Extended Data Fig. 1).

We subjected purified FASN to single-particle cryo-EM studies (Extended Data Fig. 2a). As was the case in previous cryo-EM studies[18,23,24], we observed large variations in the orientation between the condensing and modifying wings, preventing determination of a consensus high-resolution structure of the entire FASN. Aiming to reduce such heterogeneity, we incubated the purified FASN with orlistat, an FDA-approved drug against the thioesterase (TE) domain of human FASN[25,26], or 1,3-dibromopropane (DBP), which was shown to

crosslink ACP with the KS domain[27,28] (Supplementary Fig. 1h,i), and acquired two additional datasets, one from each sample. These three datasets were processed separately following the same procedure (Extended Data Figs. 2–5).

For each dataset, three-dimensional (3D) classifications isolate a subset, from which 3D reconstruction shows a fuzzy density associated with the KS domain. Focusing on these subsets, we refined the condensing and modifying wings separately to high resolutions with the best to 2.5 Å and 2.7 Å, respectively, from the 1,3-DBP sample (Extended Data Figs. 4 and 5). From these two separate reconstructions, we generated a composite map of human FASN incubated with 1,3-DBP, in which an ACP is seen bound to the KS domain (Fig. 1e–g). This composite map is of sufficient quality for accurate modelling of most of the modifying and condensing wings (Extended Data Fig. 5 and Extended Data Table 1).

## Tracking holo-ACP in condensing cycle

During the condensing cycle, ACP shuttles its substrates between MAT and KS via a Ppant that is covalently linked to S2156 (Fig. 2a). To capture intermediate conformations of ACP engaged with either KS or MAT, we

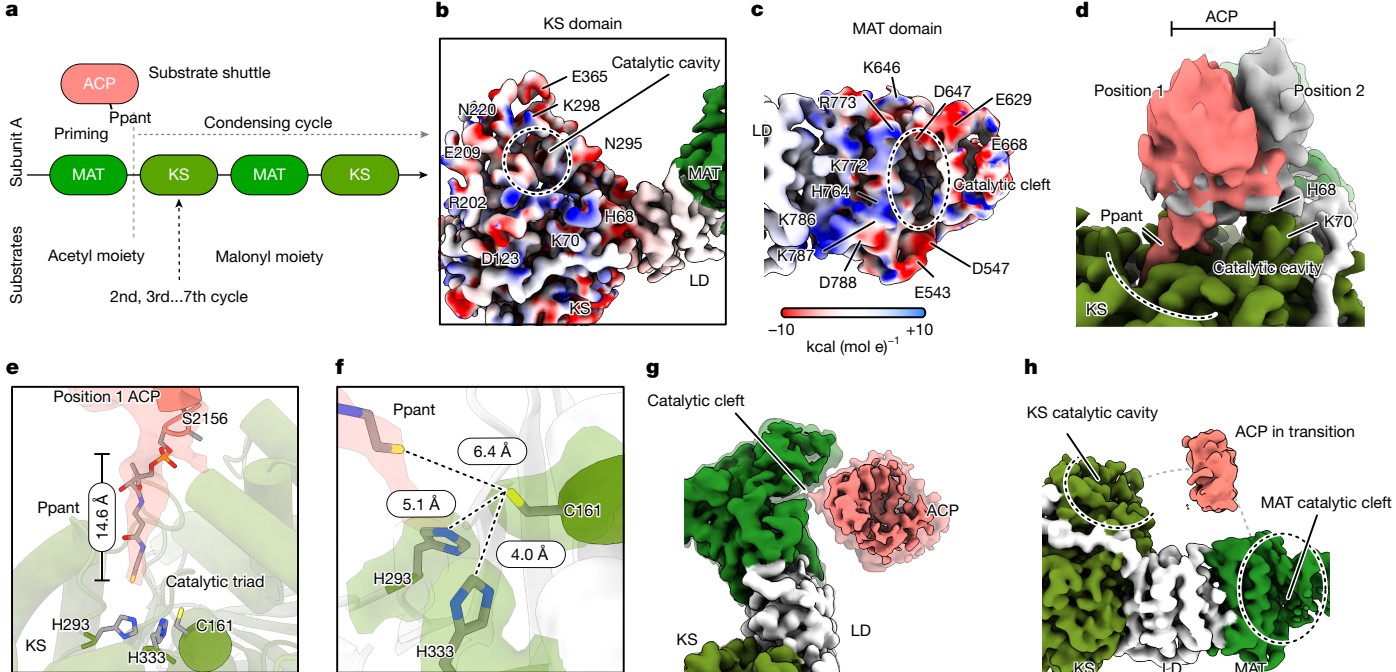

**Fig. 2 | Tracking ACP during condensing cycle. a**, A schematic illustration of Ppant-modified ACP hops through MAT and KS during the condensing cycle. **b,c**, Close-up views of the density maps around the catalytic cavity (outlined with a dashed line) in the KS (**b**) and MAT (**c**) domains. The density maps are coloured with the electrostatics of individual residues in the atomic model. Residues around the cavity are marked. **d**, ACP engagement with KS catalytic cavity was captured in two different positions (salmon and grey). Position 1 (salmon) shows a density inserted into the catalytic cavity, marked as Ppant.

In position 2 (grey), a helical density from ACP engages with H68 and K70 of KS. **e**, A non-protein density (transparent grey) in the catalytic cavity is modelled as Ppant attached to S2156 of ACP and reaches the catalytic triad of the KS domain. **f**, An enlarged view of the catalytic triad and the tip of Ppant. Residues and distances are labelled. **g**, Attachment of ACP density to the catalytic cleft of the MAT domain. **h**, An ACP density is seen not attached to the condensing domain but located in between KS and MAT.

combined particles of ACP-containing subsets from all three datasets to produce a single consensus composite map and subjected each wing to symmetry expansion followed by focused classification. This analysis produced 11 meaningful classes, which we grouped into 5 functional groups—ACP engaged with KS; ACP engaged with MAT; ACP between KS and MAT; and two different groups with ACP not seen (Extended Data Fig. 6). In all of these groups, the catalytic cavities of KS and MAT are well resolved. The surfaces surrounding these catalytic sites contain mainly charged residues (Fig. 2b,c), suggesting that the engagements of ACP to KS or MAT are probably mediated by electrostatic interactions, similar to those seen in yeast FAS[29–31].

In the group of ACP engaged with KS (ACP–KS), an ACP density is captured in two positions attached to KS, revealing different levels of engagement (Fig. 2d and Extended Data Fig. 7a–c). In one position (the grey density), a clear helical density is resolved in ACP interacting with H68 and K70 on the KS domain next to the catalytic cavity. In the other position (salmon density), a non-protein density extends about 15 Å from S2156 in the docked ACP model into a cavity towards catalytic C161 surrounded by H293 and H331, which have been suggested to be the catalytic triad of KS[1,32,33] (Fig. 2e,f, Extended Data Fig. 7d and Supplementary Video 1). We modelled this density as a Ppant arm attached to S2156. Visualizing Ppant density engaging with the KS catalytic site suggests that this is indeed a relevant snapshot of FASN in the condensing cycle.

In the ACP–MAT group, we observe a density resembling the shape of ACP loosely attached to a well-resolved catalytic cleft in MAT (Fig. 2g and Extended Data Fig. 6b,e). We interpret this density as an ACP engaging with MAT without stable interactions (Extended Data Fig. 7e and Supplementary Video 2). Unlike in ACP–KS, no density of Ppant is seen to engage with the MAT catalytic cleft. It is likely that the engagement of ACP to MAT is transient, consistent with the finding that MAT has a higher turnover than KS[34,35].

In the ACP group in between KS and MAT, a less well-defined extra density is seen that engages with neither KS nor MAT. We interpret it as ACP in transition between the two domains (Fig. 2h, Extended Data Fig. 7f and Supplementary Video 3). Together, these snapshots represent three different substates in the condensing cycle.

In both the no-ACP and dynamic MAT groups, ACP is not seen. A key difference between these two groups is that intact MAT is clearly resolved in the no-ACP group but only partially resolved in the dynamic MAT group, revealing different levels of dynamics in the catalytic cleft (Extended Data Fig. 7g). Without seeing where ACP is, it is difficult to assign these snapshots to specific substates.

For the modifying wing, we also applied symmetry expansion followed by 3D classification to separate particles of the modifying domain into two main categories, with (about 32%) and without (about 68%) well-resolved Ψ-domains attaching to β-ketoreductase (KR) (Extended Data Fig. 7h). Comparison between the two reveals motions of the Ψ-domains against the rest of the modifying domain, which remains stable (Extended Data Fig. 7i,j). Both KR and Ψ–KR have a Rossmann-like fold, containing two α-helical domains separated by a β-sheet (Extended Data Fig. 7j). Such an α/β/α sandwich architecture is known to facilitate domain fluctuations[36–38]. Indeed, we observed that the β-sheet of KR serves as a hinge allowing progressively increased motion from Ψ–KR towards Ψ–methyltransferase (Ψ–ME) (Extended Data Fig. 7i,j).

Finally, we convert all symmetry-expanded particles back to restore the intact dimeric form, leading to four combinations of dimeric modifying wing, with two, one on either side or no resolved Ψ-domains, and fifteen combinations of dimeric condensing wing, with each monomeric domain assigned to a specific functional intermediate (Extended Data Fig. 6d,e). In total, there are 60 combinations of condensing and modifying wings identified above, representing 60 specific substates,

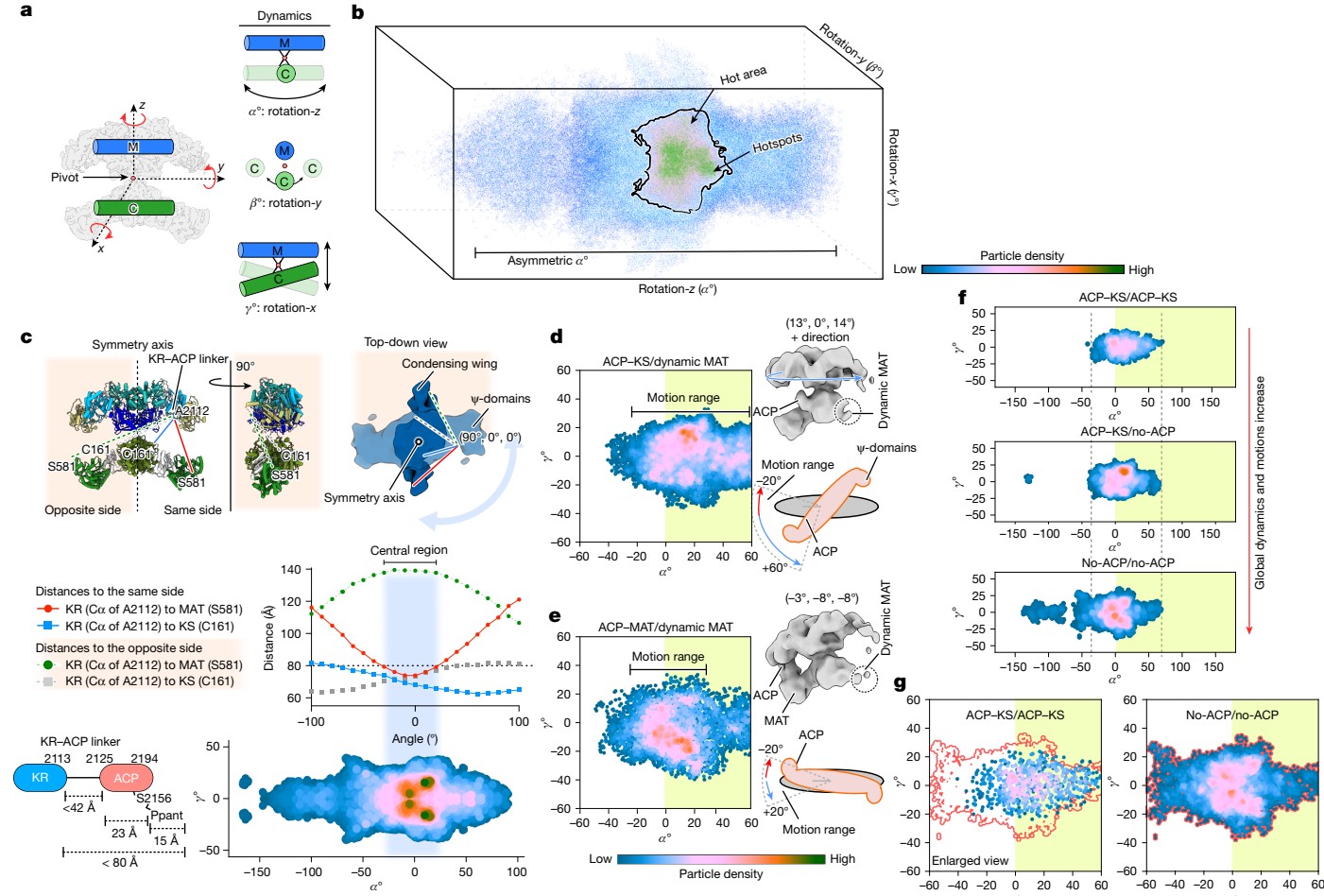

**Fig. 3 | ACP engagements with condensing wing restrict FASN dynamics.** **a**, The orientation between the modifying and condensing wings of a FASN particle is described by three angles, $\alpha$, $\beta$ and $\gamma$. Rotation-$z$ ($\alpha°$), rotation-$y$ ($\beta°$) and rotation-$x$ ($\gamma°$) indicate rotation around the $z$, $y$ and $x$ axis, respectively. **b**, A 3D plot of the particle distribution landscape of the orientation between the two wings. The plot is colour-coded with the local particle density. **c**, ACP engagement with MAT or KS. Upper panel: the distances from residue A2112, where ACP is linked to the KR domain, to the catalytic site of MAT (S581) and KS (C161) on the same side (solid red and blue lines), or the opposite side (dashed green and grey lines). Lower panel: distances from A2112 to C161 and S581 plotted against the $\alpha$ angle. The dashed line marks the maximum reach of holo-ACP. The shaded band indicates the range of the $\alpha$ angle, within which ACP can engage both KS and MAT. The putative reach of ACP is calculated on the basis of the maximum stretch of the linker, the size of ACP and the size of Ppant.

**d**, The conformational landscape of the ACP–KS/dynamic MAT substate (left), a reconstruction calculated from particles around (13°, 0°, 14°) (upper right) and a schematic illustration showing the range of $\alpha$ with ACP engaged with KS (lower right). **e**, The conformational landscape of the ACP–MAT/dynamic MAT substate (left), a reconstruction calculated from particles around (−3°, −8°, −8°) (upper right) and a schematic illustration showing the range of $\alpha$ with ACP engaged with MAT (lower right). **f**, The landscapes of three different substates, ACP–KS/ACP–KS (upper), ACP–KS/no-ACP (middle) and no-ACP/no-ACP (bottom), each with intact pseudo domains. $\alpha$ motion increases without ACP engagements on the condensing wings. **g**, Enlarged views of the central region of the conformational landscape showing a side-by-side comparison of ACP–KS/ACP–KS (left) and no-ACP/no-ACP (right). The landscape contour for no-ACP/no-ACP from the right panel is overlaid in red in the left panel.

mostly in the condensing cycle (Extended Data Fig. 6f). They illustrate how ACP shuttles between KS and MAT in each monomer during the condensing cycle with the modifying wing changing its conformation accordingly. As we did not observe any coordination between the ACP shuttling in the two monomers in our sample, it is likely that the catalytic steps of the two monomers in FASN are unsynchronized in the condensing cycle.

We then calculated the reconstruction from each substate, hoping to resolve both the condensing and modifying wings with a specific orientation between them. However, for every substate or snapshot, we failed to reconstruct such a map. Instead, each of the two wings can be resolved only individually, suggesting that FASN particles of each substate still have large variation in the orientation between the two wings. To fully appreciate the conformational dynamics of FASN, it is necessary to analyse the embedded orientational variations between the two wings of FASN in any specific substates.

## Conformational landscape of human FASN

When we reconstruct the composite map (Fig. 1e), the modifying and condensing wings are refined and determined separately but from the same particles. Thus, every particle has two sets of assigned Euler angles, one for each wing, from which the relative orientation between the two can be calculated as rotational angles $\alpha$, $\beta$ and $\gamma$ around three orthogonal axes $z$, $y$ and $x$ (Fig. 3a and Extended Data Fig. 8a). In a Cartesian coordinate system with $\alpha$, $\beta$ and $\gamma$ as the axes, each particle is represented as a point. Plotting all particles in this coordinate system produces a particle distribution landscape, which we refer to as the conformational landscape, with all orientations between the two wings (Fig. 3b). Particle densities at any given coordinate in this landscape represent the population of FASN particles with a specific orientation between the two wings defined by the coordinate (Extended Data Fig. 8b,c). Particles selected from any specific coordinate produce a 3D

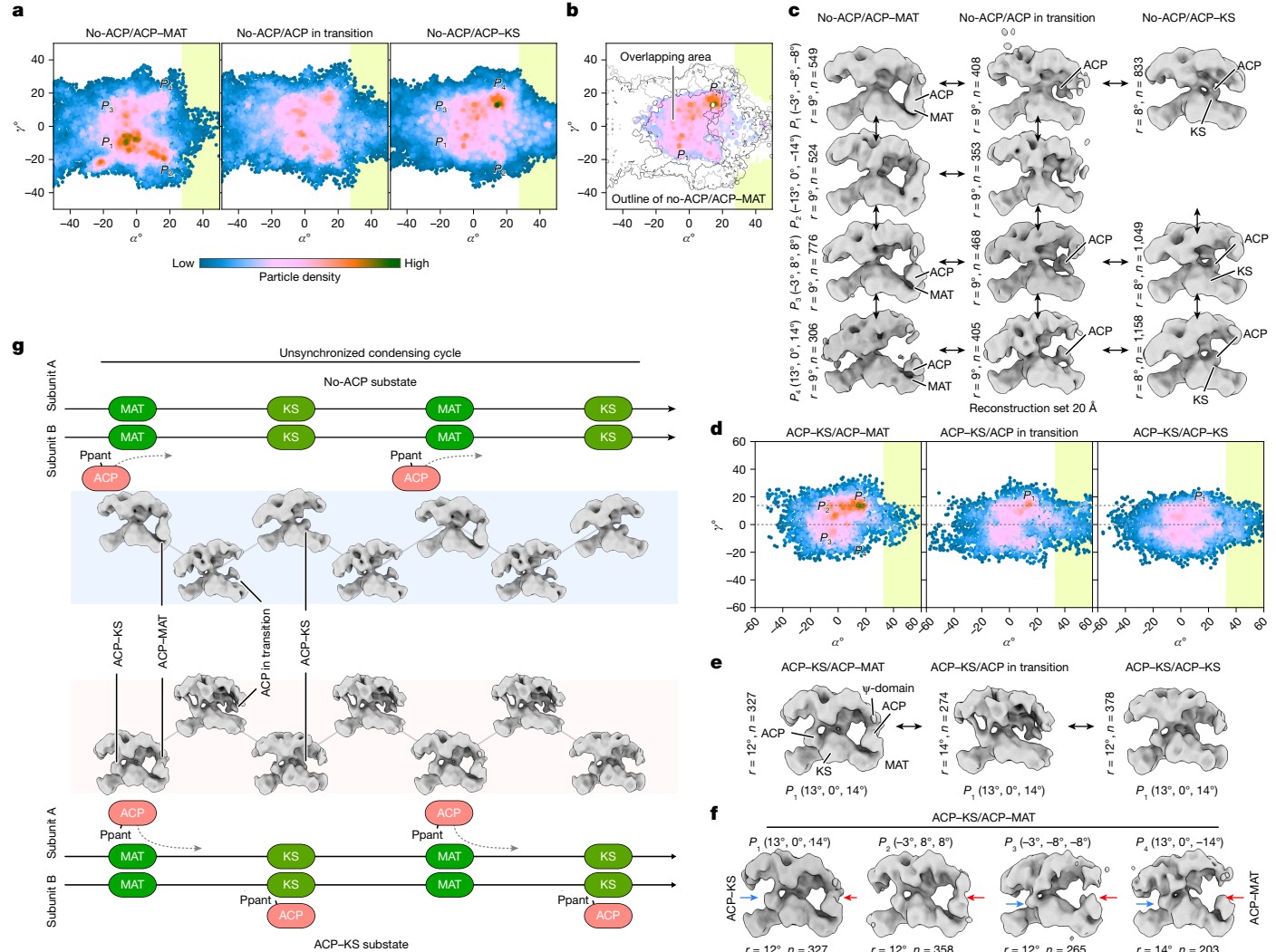

**Fig. 4 | ACP shuttling does not require larger rotation between two wings.**
**a**, The landscapes of substates associated with three steps in the condensing cycle of one monomer. **b**, The hot areas of these landscapes overlap considerably. **c**, 3D reconstructions calculated from four different locations in the landscape of each substate are arranged in columns. Comparisons of the reconstructions in each row reveal that ACP can hop between the KS and MAT domains without changing the orientation between the two domains. The reconstructions are calculated from $n$ particles within a radius of $r$ around the selected coordinates as labelled. **d**, The landscapes of three other substates. **e**, 3D reconstructions calculated from the same coordinates of the landscapes. Similarly, with a fixed

orientation between the condensing and modifying domains, one ACP keeps its engagement with the KS in one monomer and another ACP hops between KS and MAT in the other monomer. The reconstruction parameters are marked. **f**, 3D reconstructions calculated from different locations in the landscape of the ACP–KS/ACP–MAT substate show that different orientations between the condensing and modifying domains allow the same ACP engagement with the condensing domain. Reconstruction parameters are marked. **g**, A schematic view of the condensing cycle in human FASN, in which the reconstructions of different substates can be placed in different steps of the condensing cycle, revealing that the enzymatic reactions carried out by the two monomers are not correlated.

reconstruction with well-defined orientation between the modifying and condensing wings, validating the algorithm that we developed to calculate the conformational landscape (Extended Data Fig. 8d–f and Supplementary Figs. 4 and 5).

Pooling together particles of all substates identified above, the landscape analysis shows that the range of angles between the two wings is larger in $\alpha$ (swivelling) than in $\beta$ and $\gamma$ (swinging; Fig. 3b and Extended Data Fig. 8b,d). This is reasonable considering the constraints imposed by rotating two elongated wings against each other around a pivot joint in the middle. Notably, the landscape shows regions with higher particle densities (hot areas) and even a few distinct hotspots. We interpret these hot areas and hotspots to represent relatively favourable orientations between the two wings. The landscape is also asymmetrical along $\alpha$. Overall, the shape and the appearance of the landscape provide the orientations between the two wings that a FASN particle can experience. Notably, we observed that small molecules, orlistat or 1,3-DBP, do not

alter the overall shapes of the conformational landscapes but generate a few hotspots within the central region (Supplementary Fig. 4).

The conformational landscape analysis described here treats each wing as a rigid body. Thus, it characterizes only the orientation between the two wings without accounting for any conformational heterogeneity within each wing. Next, we apply this analysis to each substate or snapshot identified from classifications to provide a more comprehensive description of FASN dynamics associated with ACP shuttling in the condensing cycle.

## ACP engagements constrain the landscape

ACP is linked to KR in the modifying wing with a short linker (Fig. 1a). Engagement of ACP with KS or MAT in the condensing wing can occur only if the distance between KR and the catalytic site of KS or MAT is shorter than the maximum reach of the Ppant arm, estimated to

be around 80 Å (ref. 18; Fig. 3c). By measuring the distance between A2112 in the KR domain, where the ACP is linked, and the surface of catalytic cavities of KS or MAT either in the same or opposite sides at different $\alpha$ angles, we found that ACP–KS is favoured when $\alpha$ is in the range from about −50° to beyond +50°, whereas ACP–MAT is favoured within a more confined range, from −20° to +10°. Indeed, the central hot area roughly matches the range of $\alpha$ angles within which ACP can engage with both the KS and MAT domains (Fig. 3c). Taking two substates, ACP–KS/dynamic MAT and ACP–MAT/dynamic MAT, as examples, the hot area of the ACP–KS/dynamic MAT substate extends from −20° to +60°, whereas that of the ACP–MAT/dynamic MAT substate is confined within −20° to +20° (Fig. 3d,e). This difference can be explained by the fact that KS is located closer to the linker and that the catalytic sites of KS and MAT face opposite sides of the condensing domain. More examples are shown in Extended Data Fig. 9. Without ACP engagement to the condensing wing, there are more particles with $\alpha$ angles beyond −50° (Fig. 3f,g). Obviously, particles with a large $\alpha$ angle cannot engage their ACPs to the condensing wing; thus, they are incapable of carrying out the catalytic reactions of the condensing cycle.

## Snapshots in the condensing cycle

Next we ask the question of whether, within the central region of the landscape, any specific enzymatic steps of the condensing cycle correlate with a specific orientation between the two wings. During one condensing cycle, including the priming step, ACP cycles through engagements with MAT and KS (Fig. 2a). In one example, three substates identified from classification (that is, no-ACP/ACP–MAT, no-ACP/ACP in transition and no-ACP/ACP–KS) are associated with these steps. Here, one monomer of FASN steps through these catalytic steps while the other monomer has no ACP engaged with the condensing domain. Their landscapes show a similar particle distribution around the central section with considerable overlaps, yet the hotspots of each substate are located differently (Fig. 4a,b). Within the overlapping region, we calculated reconstructions from four different locations in each landscape (Fig. 4c, columns). These reconstructions show that, for each substate, engagement of ACP with a specific catalytic domain (MAT, KS or in between) is maintained at different orientations between the two wings. Similarly, reconstructions calculated from different substates but from the same location in the landscape demonstrate that, at any given orientation between the two wings, ACP also shuttles substrates between KS and MAT (Fig. 4c, rows).

In another example, three substates, ACP–KS/ACP–MAT, ACP–KS/ACP in transition and ACP–KS/ACP–KS, can also be assigned to a condensing cycle in one monomer, while the other maintains ACP engagement with KS. With one ACP engaged with KS, the landscapes are more constrained (Fig. 4d). Similarly, the same conclusion can be drawn from the reconstructions calculated from the same location of each substate, such as at a hotspot of the ACP–KS/ACP–MAT substate (13°, 0°, 14°) (Fig. 4e) and from three different locations of the same substate (Fig. 4f). Together, reconstructions calculated here can be placed at different catalytic steps during the condensing cycle of a FASN dimer, without distinguishing which substrate, acetyl or malonyl moiety, the ACP carries (Fig. 4g).

More examples are shown in Extended Data Fig. 9 and Supplementary Fig. 5. One feature revealed from all of these landscapes is that the hot (pink) area of each substate is confined in a relatively small range and they overlap considerably with each other (Fig. 4, Extended Data Fig. 9 and Supplementary Fig. 5). Although we observed ACP engaged only with the condensing domain located in the same side, we cannot conclude that ACP engages only with the condensing domain of the same monomer. Structurally, ACP could engage with the condensing domain of either monomer, as previously suggested[39]. Nevertheless, we conclude the following: during the condensing cycle, ACP shuttles

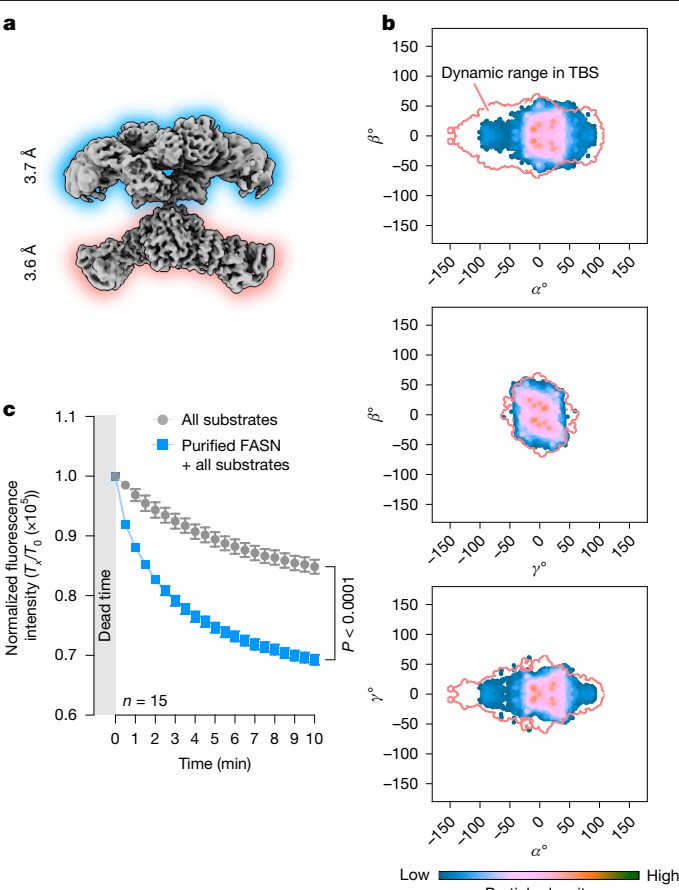

**Fig. 5 | FASN with reduced rotational motion remains functional.**
**a**, A composite cryo-EM map of human endogenous FASN in PEG 6000 buffer. **b**, The conformational landscape of FASN in PEG 6000 buffer overlaid with that in TBS buffer (red contour). PEG 6000 buffer reduces the range of rotation between the two wings. **c**, NADPH consumption by FASN with substrates in PEG 6000 buffer. $T_0$ and $T_x$ indicate the starting (0 min) and measuring ($x$ min) point, respectively. The two-tailed unpaired parametric Student $t$-test was implemented to obtain the $P$ value, indicated in the figure. The data are shown as the mean ± s.e.m. Prism was used to generate the graph and calculate the statistics.

substrates between MAT and KS at any given orientation within the central region in the landscape; the catalytic reactions in the condensing cycle of each monomer in FASN are unsynchronized with each other; and we cannot identify any specific path in the landscape that would suggest a specific trajectory of change of orientation between the two wings leading from one catalytic step to another.

## Large motion is not essential for FASN

Our landscape analysis suggests that the catalytic reactions of the condensing cycle can occur at a relatively confined range of orientation between two wings. To further support this, we isolated particles from two hotspots in the landscape of all substrates combined, and subjected particles to conformational variation analysis[40]. In both hotspots, while maintaining the same orientation between two wings, ACP is seen to engage with KS, MAT or in between (Supplementary Figs. 6 and 7). Considering the distance that a Ppant-modified ACP can reach, particles within the central region can carry out all reactions in the modifying cycle. Together, the findings of our analysis here show that FASN particles within the central range of the landscape can perform the entire enzymatic reaction of palmitate synthesis. Thus, a large rotation is probably not required for FASN function. Indeed, the

central regions have higher particle densities than the regions of large rotations (Fig. 3 and Extended Data Fig. 9).

Next we demonstrate that, without a large rotation, FASN is fully functional in consuming NADPH and elongating acyl chains. As rotational motion between the two wings is not energy driven, we reason that a higher-viscosity buffer, such as PEG 6000, would reduce the rotational motion between the two wings. Indeed, we observed that the landscape of FASN in PEG 6000 buffer has a similar distribution in the central region but the population of particles with large rotational angles is noticeably reduced (Fig. 5a,b and Supplementary Fig. 8). However, FASN in PEG 6000 buffer remains functional (Fig. 5c and Extended Data Fig. 1c–e), supporting the conclusion that a large rotational motion is not required for FASN function.

## Discussion and conclusion

Together, our findings suggest a scenario in which rotations between the two FASN wings are stochastic and random. They are constrained by ACP engagements in two monomers with KS or MAT. There is no defined path in the conformational landscape that would suggest how such rotations facilitate ACP shuttling from one enzyme to another. Thus, rotations between the two wings are not correlated with the catalytic steps of the condensing cycle. Transitions between different substates occur in almost any orientation within the central region.

Previous studies suggested three plausible models of ACP movement in FASN and related type I polyketide synthases (PKSs), free diffusion, medium conformational constraint and excessive conformational constraint[19,41]. Our current study suggests that ACP movement in FASN follows the medium constraint model. When the orientation between the two wings is within the central region of the landscape, ACP stochastically shuttles substrates between different enzymatic domains. Small rotations may aid transition between different catalytic steps, as localized hot areas are seen in the landscape of different substrates. However, no condensing reaction can be carried out when the angle between the two wings is too large (Fig. 4, Extended Data Fig. 9 and Supplementary Fig. 5).

A major architectural difference between yeast FAS and metazoan FASN is the so-called reaction chamber, which is closed in yeast but open in metazoans. Within a closed reaction chamber such as in yeast FAS, ACP can reach all enzymatic domains to complete acyl chain synthesis[31]. By contrast, with an open and dynamic reaction chamber of metazoan FASN, acyl chain synthesis can be effectively accomplished only if the angle between the two wings is confined within a certain range. Beyond this range, FASN molecules are incapable of completing the entire acyl chain synthesis, unless they can actively rotate back.

We can speculate on the enzymatic advantage of having two dynamic open chambers in FASN. If multiple FASN molecules are located close enough, an open chamber would allow ACP to be shuttled across molecules. If that occurs, FASN molecules with large rotational angles between their two wings are then capable of completing acyl chain synthesis. When rapid de novo fatty acid synthesis becomes necessary, such as in cancer cells or during viral infection by enveloped viruses, *FASN* expression is upregulated[9,42]. As shown in Dengue and classical swine fever viral infections, viral non-structural proteins redistribute host FASN to sites of viral replication, increasing local FASN concentration and cellular fatty acid synthesis[10,43]. A speculation from this observation is that the increased local FASN concentration causes congregation of FASN so that catalytic steps of fatty acid synthesis can be carried out by multiple FASN particles clustered together.

Beyond metazoan FASN, bacterial PKSs also have open reaction chambers[19,44,45], and some are considered to have similar rotational motion between functional wings to shuttle ACP[46,47]. Congregations of PKSs were observed in bacterial cells[48], and high-order organization of PKSs was proposed[49]. Without further experimental evidence, we can only speculate that the open reaction chamber in PKSs may also

allow enzymatic reactions across different molecules to increase the overall enzyme productivity.

The conformational landscape analysis presented here was inspired by similar approaches taken previously[50–53]. In our study, the conformational landscape is analysed in real space and directly interpretable as the angles between the two bodies of a molecule. This approach is suitable for characterizing near-rigid-body motions, particularly rotations, between different domains within a protein or a complex. The calculation of the landscape depends on deterministic assignment of Euler angles to specific domains as near-rigid bodies, and its accuracy depends on alignment and classification of heterogeneous particles into defined classes. Combining this approach with other methods, such as 3D variability analysis, deep learning based cryoDRGN or other newly developed flexibility analysis tools[40,54], would provide a more rigorous analysis of conformational dynamics.

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

# Methods

### FASN stable cell line generation

Endogenous tagging of human FASN in HEK293T was described previously[22]. Specifically, adherent HEK293T cells were cultured in DMEM medium (Gibco) supplemented with 10% FBS (v/v) at 37 °C with 8% $CO_2$. When the cells reached approximately 50% confluence, knock-in materials, including 300 ng of px458 and 500 ng of homology-directed repair plasmid donors, were transiently transfected using Lipofectamine 2000 (ThermoFisher). After two passages of the HEK293T cells, cells were chemically selected using puromycin (ThermoFisher) at a concentration of 1–2 µg ml$^{-1}$. Continuous puromycin treatment eliminated unedited cells for 10–14 days (10–14 cycles of selection), and validation of the stable cell line was performed using immunoblotting against both anti-Flag–peroxidase (HRP) (Sigma-Aldrich, A8592) and anti-GAPDH (Proteintech, CL594-60004), and anti-mouse IgG–HRP (Invitrogen, 31450; Fig. 1c and Supplementary Fig. 9). Immunoblotting procedures were adapted as previously described[22].

Genetically engineered HEK293T cells show similar cell proliferation compared to wild-type HEK293T cells (Supplementary Fig. 1c), indicating that tagging FASN does not perturb endogenous FASN function. Cell growth was measured using crystal violet staining[55,56]. FASN stable cells were seeded into one well in a 6-well plate with a density of $0.25 \times 10^6$ cells ml$^{-1}$ (real count) and grown overnight in DMEM supplemented with 10% FBS (v/v%) at 37 °C with 8% $CO_2$ to completely attach to the plate. Every 24 h, medium was gently removed, and the cells were washed twice using 1 ml of phosphate-buffered saline (PBS) and then once using 1 ml of 4% formaldehyde (initial 16% (w/v) (Thermo Scientific)) to fix the cells. After fixing for 1 h at room temperature, the solution was gently aspirated and washed twice using 1 ml of PBS. To count cell numbers, cell staining solution, crystal violet (Abcam, ab232855), was gently applied into the plate, and the plate was slowly shaken at room temperature for 1 h. The crystal violet staining solution was aspirated, and the cells were rinsed with ice-cold PBS twice. Cells were destained for 1 h in the presence of 10% citric acid solution, and 100 µl of the destaining solution was taken from the plate to measure absorbance at 570 nm ($OD_{570nm}$) using a SpectraMAX M5 (Molecular Devices).

### Fluorescence size-exclusion chromatography

Endogenous FASN size and accessibility of nanobody binding for secondary affinity purification were validated using fluorescence size-exclusion chromatography (FSEC) by coupling an eGFP-conjugated ALFA nanobody (ALFANb) as previously described[22]. The FASN stable cells were collected and resuspended in 180 µl ice-cold TBS buffer, and then combined with 20 µl of 100 mM $n$-dodecyl-β-D-maltoside and 20 mM cholesteryl hemisuccinate to lyse cells. The mixtures were rotated at 4 °C for 30–45 min followed by spin down at 21,000$g$ for 20 min at 4 °C. The supernatant was incubated with 1 µl of purified ALFANB–eGFP (0.1 mg ml$^{-1}$) for 1 h at 4 °C, and then directly injected onto a Superose 6 Increase 10/300 GL column (Cytiva), pre-equilibrated with TBS buffer, at a flow rate of 0.5 ml min$^{-1}$. GFP signal was detected by an HPLC system (Shimazu) equipped with an RF-20A fluorescence detector (excitation 488 nm and emission 508 nm).

### Human endogenous FASN purification

FASN stable cells were transferred to a T75 flask and grown to approximately 70–90% confluence. Cells were then transferred to 20 ml Freestyle293 medium (Gibco) supplemented with 1% FBS (v/v) to adapt to suspension culture, which typically takes a week. Suspension-adapted cells were grown at 37 °C with 5% $CO_2$ in the presence of puromycin at a concentration of 1 µg ml$^{-1}$. Cells were then grown without puromycin until protein purification.

The FASN stable cells were grown to a density of $3–4 \times 10^6$ cells ml$^{-1}$ in 400–800-ml cultures before collection by centrifugation at 3,000–4,000$g$ for 10 min at 4 °C in a floor centrifuge (Beckman Coulter rotor

JLA-8.1). The cell pellet was resuspended with ice-cold TBS buffer supplemented with protease inhibitor cocktail (Sigma-Aldrich, S8820). Cells were lysed with a probe sonicator over an ice–water mixture with gentle stirring to prevent the samples from overheating. The cell debris was then discarded by two follow-up centrifugation steps, one at 8,000$g$ for 20 min at 4 °C using the rotor JA-25.50 (Beckman Coulter), and then at 126,000$g$ for 1 h at 4 °C using the rotor Ti45 or 50.2Ti (Beckman Coulter), sequentially. The final supernatant was applied to 1 ml pre-equilibrated anti-Flag M2 affinity gel (Sigma-Aldrich, A2220) and incubated for 2 h. The beads were extensively washed with 50 column volumes (CVs) of 500 mM NaCl and 20 mM Tris-HCl pH 8.0, followed by 50 CVs of TBS buffer. To elute the proteins, 2 CVs of TBS buffer supplemented with 0.25 mg ml$^{-1}$ of 3×Flag peptide (GlpBio, GP10149) was applied twice. For secondary affinity purification, 2 mg of an in-house-purified ALFANb–eGFP was supplied to the elution and incubated for 30 min at 4 °C. The mixture was loaded onto 0.5 ml of TALON resin (Takara Bio) for 2 h to further isolate a more pure FASN sample from other contaminants, such as methylosome and proteosome[22]. The TALON resin was washed using 10 CVs of 20 mM imidazole pH 8.0-containing ice-cold TBS, and FASN was eluted using 3 CVs of ice-cold TBS supplemented with 300 mM imidazole pH 8.0.

The eluate was further purified by loading onto the size-exclusion chromatography column in TBS buffer, Superose 6 Increase 10/300 GL, and the fractions containing the target proteins were determined by FSEC and SDS–polyacrylamide gel electrophoresis (PAGE). The fractions were pooled together and concentrated.

### Mass spectrometry validation

A 1–3 µg quantity of purified FASN was subjected to mass spectrometry (MS) to validate the identity of human FASN and identify any possible post-translational modifications. Five sample volumes of ice-cold acetone were added to the FASN sample and left at −20 °C overnight. The precipitated sample was collected at 21,000$g$ for 30 min at 4 °C, and the supernatant was discarded gently. Residual acetone was air-dried for an additional 30 min at room temperature. The samples were submitted to the University of California San Francisco (UCSF) MS facility for liquid chromatography–MS analysis. In brief, the peptides from the in-solution tryptic digestion recovered by C18 ZipTips (Millipore) were analysed on a QExactive Plus (Thermo Scientific), connected to a NanoAcquityUltraPerformanceUPLC system (Waters). A 15-cm EasySpray C18 column (Thermo Scientific) was used to resolve peptides (60-min 2–30% gradient with 0.1% formic acid in water as mobile phase A and 0.1% formic acid in acetonitrile as mobile phase B). The mass spectrometer was operated in positive ion mode and in data-dependent mode to automatically switch between MS and MS/MS. MS spectra were acquired between 350 and 1,500 $m/z$ with a resolution of 70,000.

After spectrometric data acquisition, peak lists generated were searched against the *Homo sapiens* (HEK293T cells) SwissProt. 2022.05.26. random.concat database (containing 567,483 entries) using Protein Prospector[57] with the default parameters: enzyme specificity was set as trypsin, and one missed cleavage per peptide was allowed; *N*-acetylation of the amino terminus of the protein, loss of protein N-terminal methionine, pyroglutamate formation from peptide N-terminal glutamines and oxidation of methionine were allowed as variable modifications. Mass tolerance was 10 ppm in MS and 30 ppm in MS/MS. The false positive rate was estimated by searching the data using a concatenated database that contains the original Swiss-Prot database, as well as a version of each original entry in which the sequence has been randomized. A 1% false discovery rate was permitted at the protein and peptide levels. All spectra identified as matches to peptides of FASN (peptide spectral matches) were reported.

For Ppant identification, in-gel tryptic digestion was performed in the presence of dithiothreitol and IAA for reduction and alkylation, respectively.

## NADPH consumption assay

We measured the decrease of fluorescence intensity generated from NADPH as a measure of its consumption. Using a SpectraMax ID5 plate reader (Molecular Device) with a 350-nm excitation light, the emission spectrum from a control sample at 37 °C that contains all reaction reagents, except purified FASN, shows a peak at 460–480 nm (Extended Data Fig. 1a).

The endogenous FASN was purified as described above. For each biological replicate, 1,600–2,000 ml of FASN stable cells were used for FASN purification. After size-exclusion chromatography, all fractions collected were subjected to SDS–PAGE. Fractions containing pure FASN were pooled together and concentrated to 200 nM. The 10× stock solutions of the three substrates, NADPH (Sigma, 481973), acetyl-CoA (Sigma, A2056) and malonyl-CoA (Sigma, M4263), were each prepared in TBS buffer at concentrations of 1.6 mM, 1.6 mM and 4 mM, respectively. The 5× stock solution of PEG 6000 (90 mg ml$^{-1}$) was prepared in TBS buffer. To set each reaction, 20 µl room-temperature TBS buffer or PEG 6000 buffer was first added into a single well of a Corning transparent 96-well plate, followed by addition of 10 µl of each substrate sequentially. Finally, 50 µl of 200 nM FASN protein or the same amount of TBS buffer (for control) was added to the reaction. The final reaction volume was 100 µl. Fluorescence intensity from each reaction well at 37 °C was measured every 30 s for 10 min. The photomultiplier tube gain of the plate reader was set to 600 V with a read height of 0.67 mm. The same procedure was repeated five times for technical replicates. Three complete biological replicates were conducted from cell growth, protein purification to enzymatic assay. Prism was used to generate the graphs and calculate the statistics.

## Validation of Ppant modification on S2156 and intermediate enzymatic products

After the NADPH consumption measurement described above, samples from five wells were pooled together, concentrated to 40 µl, and subjected to SDS–PAGE (Extended Data Fig. 1d and Supplementary Fig. 9). FASN bands were excised and subjected to in-gel tryptic digestion followed by the MS analysis described above. The tryptic peptides were analysed on an Orbitrap Lumos Fusion (Thermo Scientific) in positive ion mode. MS spectra were acquired from 375 to 1,500 *m/z* with a resolution of 120,000.

Following data acquisition, peak lists were generated, and an initial protein ID search was conducted using the same parameters as above. A second search included variable post-translational modifications such as phosphorylation on S/T/Y, N6-acetylation on lysine and *O*-(pantetheine 4′-phosphoryl) Ppant modification on S2156 for samples with reduction and alkylation. For S2156 adduct identification, mass modifications ranging from 300 to 1,000 on serine and tri-oxidation on cysteine were included.

## Stability of endogenous FASN in the presence of small molecules

The FASN samples were prepared as described above in the section entitled Fluorescence size-exclusion chromatography. The different concentrations of the small-molecule drugs, orlistat (Sigma-Aldrich, O4139) and 1,3-DBP (Sigma-Aldrich, 125903), were added to the supernatant and incubated for 1 h at 4 °C. The FASN mixtures with the different concentrations of small molecules were loaded to a pre-equilibrated Superose 6 Increase 10/300 GL column (Cytiva) in ice-cold TBS at a flow rate of 0.5 ml min$^{-1}$. The alteration of GFP intensity was measured by an HPLC system (Shimazu) equipped with an RF-20A fluorescence detector (excitation 488 nm and emission 508 nm).

## Cryo-EM sample preparation and data collection

Purified FASN samples were concentrated to 0.5–1.0 mg ml$^{-1}$ using a 100-kDa-cutoff protein concentrator (ThermoFisher). In addition to FASN alone, we also incubated concentrated FASN with 3 mM 1,3-DBP (Sigma-Aldrich,125903) or 1 mM orlistat (Sigma-Aldrich, O4139) 30 min before preparing cryo-EM grids. For each grid, a 3 µl droplet of sample was applied to gold Quantifoil grids, 1.2 and 1.3-µm size and hole space, 200 mesh, blotted for 1.5–2 s with 0 blotting force at 100% humidity using a Vitrobot Mark III, and plunge-frozen in liquid ethane cooled by liquid nitrogen. Frozen grids were first screened with a Talos Arctica operated at 200 kV and equipped with a Gatan K3 camera. A screening dataset was collected and processed, yielding preliminary reconstructions of the modifying and condensing wings that were later used as references. Grids with suitable ice thickness and particle distribution were transferred to a Titan Krios operated at 300 kV and equipped with a Gatan K3 camera and a BioQuantum energy filter with the zero loss energy slit set to 10 eV. Videos were recorded in super-resolution mode at a nominal magnification of 105,000, resulting in a super-resolution pixel size of 0.4175 Å per pixel. Each video stack was dose-fractionated into 80 frames using a total exposure time of 2 s at 0.025 s per frame. The total dose was 45.8 (e$^-$ Å$^{-2}$). A total of 16,222 videos from FASN alone, 17,376 videos from FASN incubated with 1 mM orlistat, and 13,101 videos from the sample of FASN incubated with 3 mM 1,3-DBP were collected.

To reduce the dynamics of the two wings, 18 mg ml$^{-1}$ PEG 6000 was added to purified FASN and the same freezing procedure was performed. The grids were screened by a Talos Arctica electron microscope (ThermoFisher-FEI), operated at 200 kV, and equipped with a Gatan K3 camera (Gatan). A total of 4,911 videos were acquired.

## Cryo-EM data processing

Raw video stacks were subjected to motion correction, dose weighting and Fourier cropping to 0.835 Å per pixel using MotionCor2 implemented in Relion[58,59]. The contrast transfer function was estimated by CTFFIND-4.1 (ref. 60). By using relion_star_handler, micrographs with a resolution lower than the preset resolution were triaged (MaxRes cutoff 5.0 to 5.5). A total of 12,670 (FASN alone), 14,934 (FASN + orlistat) and 10,383 (FASN + 1,3-DBP) motion-corrected and dose-weighted micrographs remained for further processing.

Initial particle picking was performed in Relion using Laplacian-of-Gaussian picking from 200 randomly selected micrographs of the FASN dataset. Extracted particles were 4× binned by Fourier cropping to 3.34 Å per pixel, followed by 2D classification. 2D class averages with well-defined features were selected to train a Topaz neural network[61]. The trained Topaz model was used to pick particles from the same 200 micrographs without binning, followed by particle extractions, 4× binning, 2D classification and another round of retraining of the Topaz neural network with selected 2D class averages. The trained model was then applied to pick particles from the original micrographs followed by particle extraction, binning to 3.34 Å per pixel and 2D classification. Only particles from obvious junk classes were discarded. The remaining particles were extracted again from micrographs with 2× binning (1.67 Å per pixel) and subjected to iterative 3D classification using ab initio and heterogeneous refinements implemented in CryoSPARC[62]. The same procedure was applied to three datasets separately, from which two classes were identified. One shows a well-defined density for the modifying wing but not for the condensing wing, and it is found in all three datasets (Extended Data Figs. 2 and 4). The other shows a well-resolved condensing wing but not a modifying wing, and it is found only in datasets for FASN alone and FASN with 3 mM 1,3-DBP.

Further 3D refinement was carried out in Relion for each class separately. Initial global search and refinement was performed using 2× binned particles. For the class with a resolved modifying wing, the initial reference map is a composite map low-pass filtered to 30 Å. It was generated by combining the resolved modifying and condensing wings from the two classes identified above, following the atomic model of *Sus scrofa* FASN (Protein Data Bank (PDB): 2VZ8). For the class with a resolved condensing wing, the initial reference map is the density map obtained from CryoSPARC of this class, but low-pass filtered to 30 Å.

Further local refinements were performed using particles without binning (0.835 Å per pixel). In the iterative local refinements of each class, a mask was applied to either the modifying or the condensing wings of the reference map for the masked refinement, CTF-refinement and Bayesian polishing. This was followed by a focused classification without alignment to further isolate a subset of particles producing a reconstruction with well-defined structural details. This subset of particle was then subjected again to iterative masked refinement, CTF-refinement and Bayesian polishing. Finally, we obtained a well-resolved modifying wing from one class, in which the condensing wing appears as a blob. From the other class, we obtained a well-resolved condensing wing, but the modifying wing appears as a blob. Details are also shown in Extended Data Figs. 2–4. Until this step, from each class, we were able to obtain well-defined density only of either the modifying wing or the condensing wing but not both. The same procedure was applied to all three datasets, with some minor modifications.

To resolve both wings from the same subset of the particles, we focused on the final subset of particles obtained above. For the subset in which the modifying wing is resolved, we generated a reverse loose mask around the modifying wing to subtract background in the original particle images followed by iterative refinement using the density of the well-resolved modifying wing as the initial reference model. In parallel, a tight mask around the modifying wing was used to subtract the modifying wing from the original particle images. Then, the modifying wing-subtracted particles were subjected to a global search followed by local refinement using the density of the condensing wing obtained from the other class as an initial reference model. This procedure produced two separate reconstructions of the modifying and condensing wings from the same subset of particles. Different from the initial reference, the final reconstruction of the condensing domain obtained from this subset contains a fuzzy density of ACP.

The same procedure was applied to the final subset of the other class, in which the condensing wing is well resolved but the modifying wing is not. Similarly, two separate reconstructions of the modifying and condensing wings were reconstructed from the same subset of particles. However, no ACP density is seen in the reconstruction of the condensing domain. We did not pursue further analysis of particles in this class.

The same procedure was applied to three datasets, FASN, FASN + orlistat, FASN + 1,3-DBP. $C_2$ symmetry was applied in these procedures. Symmetry expansion was applied separately to the modifying and condensing domains, followed by focused 3D classification without image alignment ($T = 60$, $k = 4$ for the modifying domain and $T = 15$, $k = 12$ for the condensing domain).

The FASN dataset in PEG 6000 buffer from Artica was processed followed by on-the-fly motion correction[63] and mainly processed in CryoSPARC as described in Supplementary Fig. 8.

### Model building

The modifying domain of the *Sus scrofa* FASN crystal structure (PDB: 2VZ8) was used as a starting model with its amino acid sequence replaced with the human sequence using Modeller[64]. For the condensing domain, the crystal structure of the human FASN condensing domain (PDB: 3HHD) was used as the starting model. For ACP, Ppant-modified rat ACP (PDB: 2PNG) was used as the starting model, with its sequence replaced with human sequence. A complete starting model was generated by docking these atomic structures into the cryo-EM density map using USCF ChimeraX[65]. The initial models were manually modified using Coot[66], followed by several rounds of real-space refinement in Phenix[67]. Final human endogenous FASN models were validated in Coot and using Molprobity[68].

### Conformational landscape

As described above, FASN contains two main domains that can rotate with respect to each other. Such rotations prevent determination of a single high-resolution reconstruction with both domains. Instead, we determined high-resolution reconstructions of the modifying and condensing domains separately from the same subset of particles. As such, each particle has two sets of assigned Euler angles, one for each domain. Thus, for each particle, a $3 \times 3$ rotational matrix that describes the relative orientation between the two domains can be represented as the product of the inverse rotational matrix of one domain and the rotational matrix of the other domain. Here, for each domain, the rotational matrix is derived from three Euler angles assigned to the particle during the refinement. The product matrix is further converted into Euler angles following the $z$–$y$–$x$ sequence and represented in a Cartesian coordinate system in which the relative orientation between the two domains of each particle is represented as a point with coordinate ($\alpha$, $\beta$, $\gamma$), as shown in Fig. 3b. The procedure of calculating the landscape is implemented in the program CryoROLE.

To calculate a 3D reconstruction from particles around a certain coordinate, particles within a given radius from a specific coordinate are selected, and the Euler angle assigned to the condensing domain of the particles is used to calculate a 3D reconstruction with the relion_reconstruct tool in RELION 4.0 (ref. 58). A smaller radius focuses on a more narrowly defined conformational space, but it may result in fewer particles for reconstruction. Typical for the FASN datasets, a reconstruction calculated from about 200 particles can produce a reconstruction with a resolution of about 20 Å with clearly defined modifying and condensing domains with a relative orientation exactly matching the selected orientational coordinate.

### CryoDRGN analysis

CryoDRGN[40] analysis was performed on particles isolated from four spots in the combined landscape: at coordinate (13°, 0°, 14°) with a radius of 14° containing 41,851 particles; another hotspot at coordinate (−3°, 8°, 8°) with a radius of 14° containing 54,974 particles; an unfavourable spot at coordinate (−40°, 0°, 12°) with a radius of 15° containing 16,779 particles; and another unfavourable spot at coordinate (40°, 0°, 12°) with a radius of 15° containing 19,453 particles. All images were down sampled to 192 × 192. The cryoDRGN models were trained with a latent dimension size of 8 and 25 epochs. The output results were generated with a $k$-sample set to 40.

### Reporting summary

Further information on research design is available in the Nature Portfolio Reporting Summary linked to this article.

### Data availability

Composite maps of FASN alone (composite map: EMDB-44230, consensus map: EMDB-44231, focused modifying domain map: EMDB-44263, and focused condensing domain map: EMDB-44264), with orlistat (EMDB-44267, EMDB-44268, EMDB-44269 and EMDB-44270) and with 1,3-DBP (EMDB-44300, EMDB-44301, EMDB-44330 and EMDB-44329) have been deposited into the Electron Microscopy Data Bank (EMDB) database. The atomic models (modifying domain–PDB: 9B80 and condensing domain–PDB: 9B7Z) have been deposited, accordingly. Also, maps of the condensing and modifying wings calculated from combining all three datasets have also been deposited together with their symmetry-expanded 3D focused classification maps: modifying domain (EMDB-44371) and condensing domain (EMDB-44378). The symmetry-expanded map of the ACP–KS group is an additional map associated with EMDB-44378 and linked with the atomic model of the condensing wing with Ppant and partially docked ACP (PDB: 9MJ9). The pYC homology-directed repair donor plasmid is available via Addgene at https://www.addgene.org/Yifan_Cheng/.

## Code availability

The program CryoROLE (cryo-EM relative orientation landscape) used in this study is open source and available via GitHub at https://github.com/yifancheng-ucsf/cryorole with instructions.

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

**Acknowledgements** We thank J. Gross, A. Smith, U. S. Chio, M. Jin, K. Choi and R. Kalia for reading the manuscript and providing feedback; and T. Goddard and the UCSF Chimera team for generating code to convert the conformational landscape to a volume format for visualization as a 3D plot in ChimeraX. MS analysis was provided by the Mass Spectrometry Resource at UCSF (A. L. Burlingame) supported by the Dr. Miriam and Sheldon G. Adelson Medical Research Foundation (AMRF) and the UCSF Program for Breakthrough Biomedical Research (PBBR). This work is partially supported by grants from the National Institutes of Health (NIH; R35GM140847 and U54AI170792 to Y. Cheng, U54AI170792 to W.C.). Instruments at the UCSF Cryo-EM facility are partially supported by grants from the NIH (S10OD020054, S10OD021741 and S10OD026881) and Howard Hughes Medical Institute. The UCSF cryo-EM facility used in this study is managed by D. Bulkley and G. Gilbert. Y. Cheng is an Investigator of the Howard Hughes Medical Institute.

**Author contributions** W.C. initiated and performed genetic tagging of FASN in HEK293T cells, purification, biochemical, structural and functional studies and analysis. C.L. developed CryoROLE to calculate the conformational landscape. Y. Chen participated in data processing and model building. Y.W. performed MS analysis. Y. Cheng supervised the project. All authors participated in manuscript writing.

**Competing interests** Y. Cheng is a non-shareholder member of the scientific advisory boards at ShuiMu BioSciences and Pamplona Therapeutic Co. All other authors declare no competing interests.

**Additional information**
**Correspondence and requests for materials** should be addressed to Yifan Cheng.

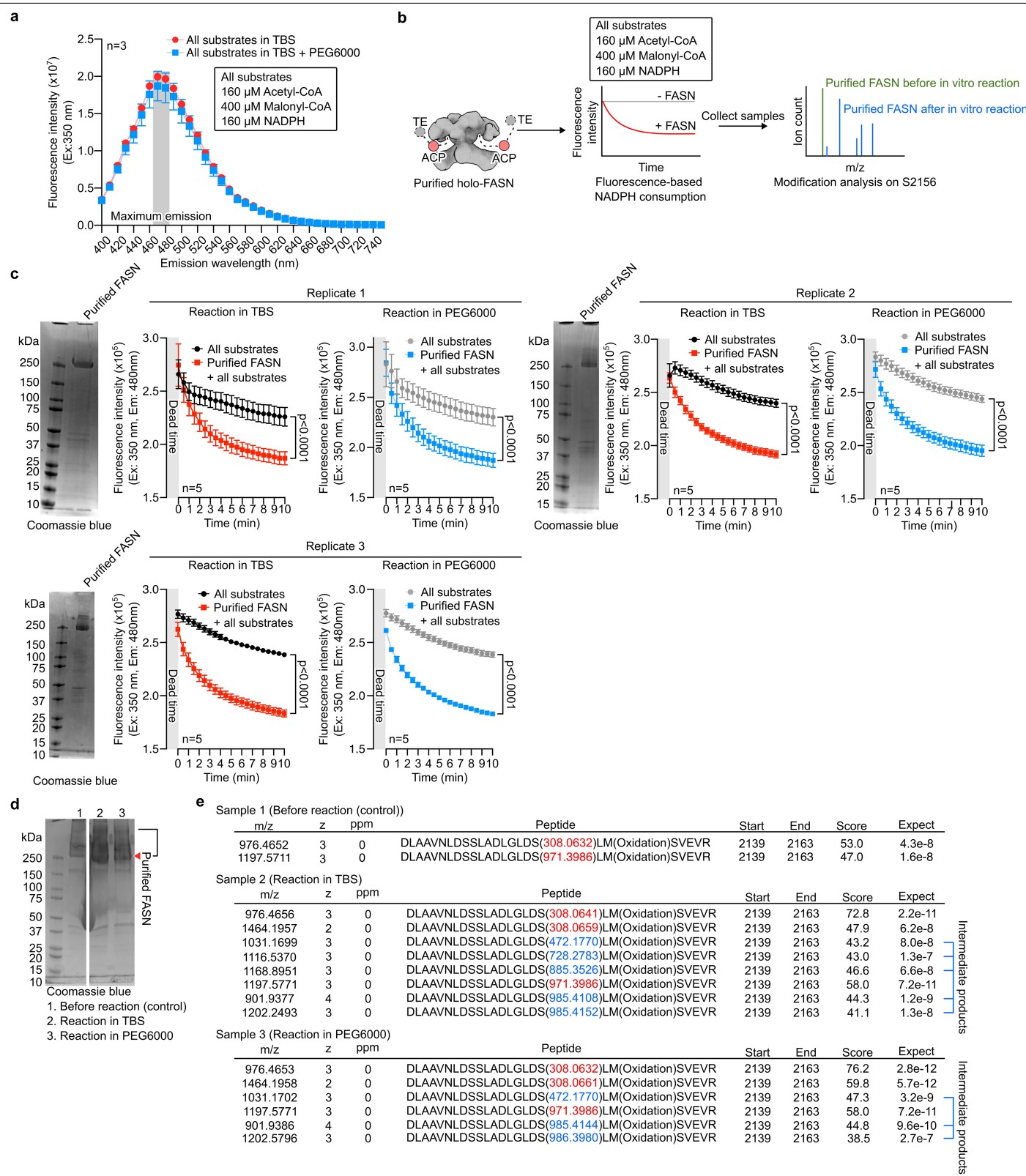

**Extended Data Fig. 1 | The functional validation of purified FASN. a**, NADPH in all substrate mixture shows fluorescence emission spectrum in the TBS with/without PEG6000 at 350 nm excitation. **b**, The schematic view of FASN assay process. The purified FASN is subjected to NADPH consumption assay, and then, the reacted samples are subjected to LC-MS to identify additional enzymic intermediates on tip of Ppant. **c**, Raw data of NADPH consumption assay using purified FASN in vitro. The purified FANS quality is demonstrated by SDS-PAGE image (left) and shows NAPDH consumption in the presence/absence of PEG6000 (right). The experiments are conducted in 3 times of biological replicates and 5 different batches of samples are measured in each purification. The two-tail unpaired parametric student T-test was implemented to obtain P-value, as labeled in the figure. The center values are means and the error bars are s.e.m. Prism is used to generate the graph and calculate statistics. **d**, Samples after reaction is subjected to SDS-PAGE analysis. For gel source data, see Supplementary Fig. 9. **e**, LC-MS of FASN bands larger than 200 kDa shows additional enzymic intermediates on Ppnat after incubating with all substrates in vitro (sample 2 and 3) versus control (sample 1). The LC-MS experiment was repeated two times.

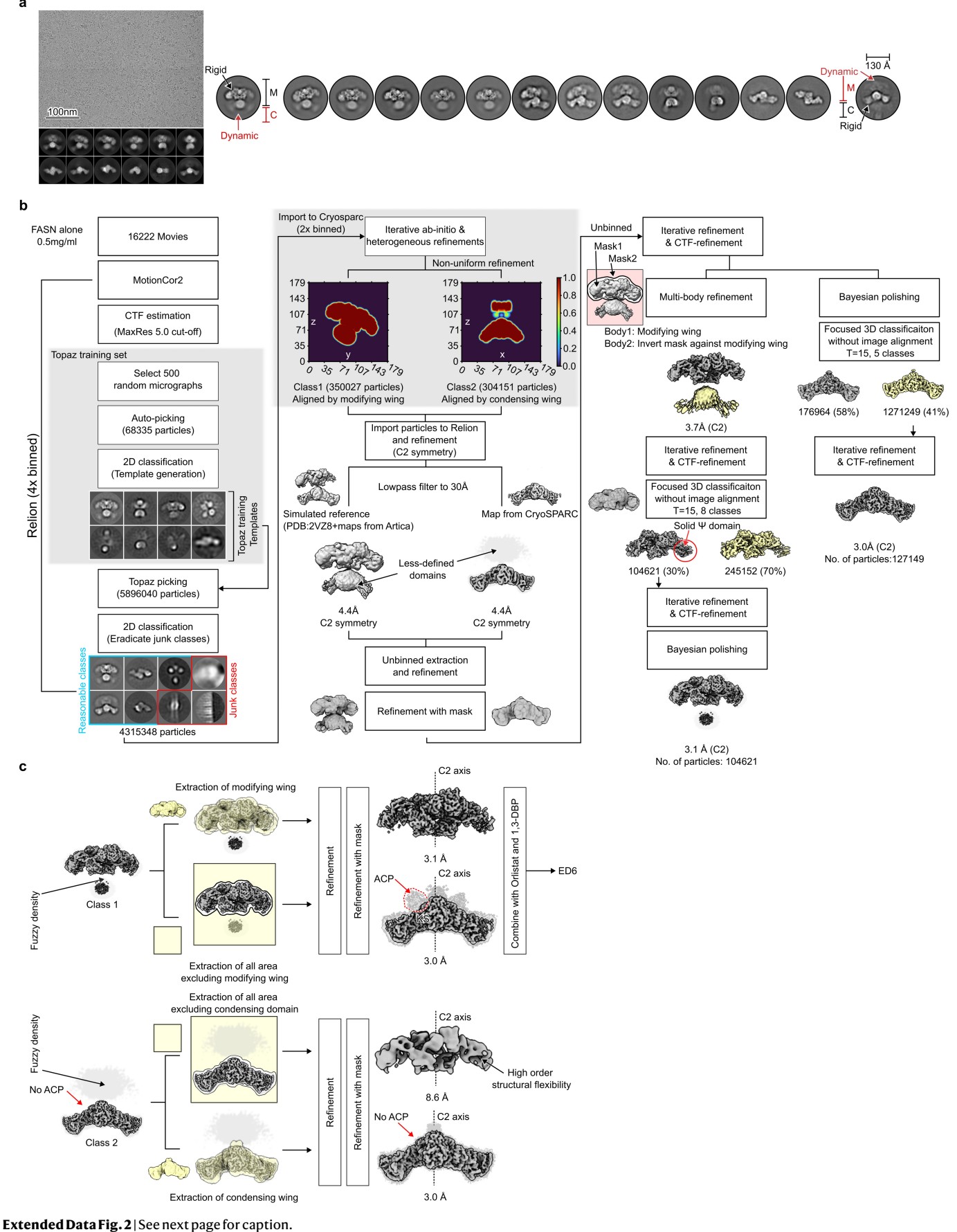

**Extended Data Fig. 2** | See next page for caption.

**Extended Data Fig. 2 | Flowchart of processing FASN alone. a**, A representative micrograph from a total of 3,150 micrographs from frozen hydrated FASN with the scale bar labeled and 2D average images. 2D average images show large rotations between the condensing and modifying domains. **b**, A detailed data processing flowchart illustrates image processing procedure of FASN alone. Two classes of particles were identified, each produces a high-resolution reconstruction of one wing, either condensing or modifying wing. **c**, A flowchart illustrating signal subtraction followed by separate alignment and refinement. From each class, final reconstructions of condensing and modifying wing are determined independently from the same subset of particles. Class with ACP is subjected to further classification, illustrated in Extended Data Fig. 6.

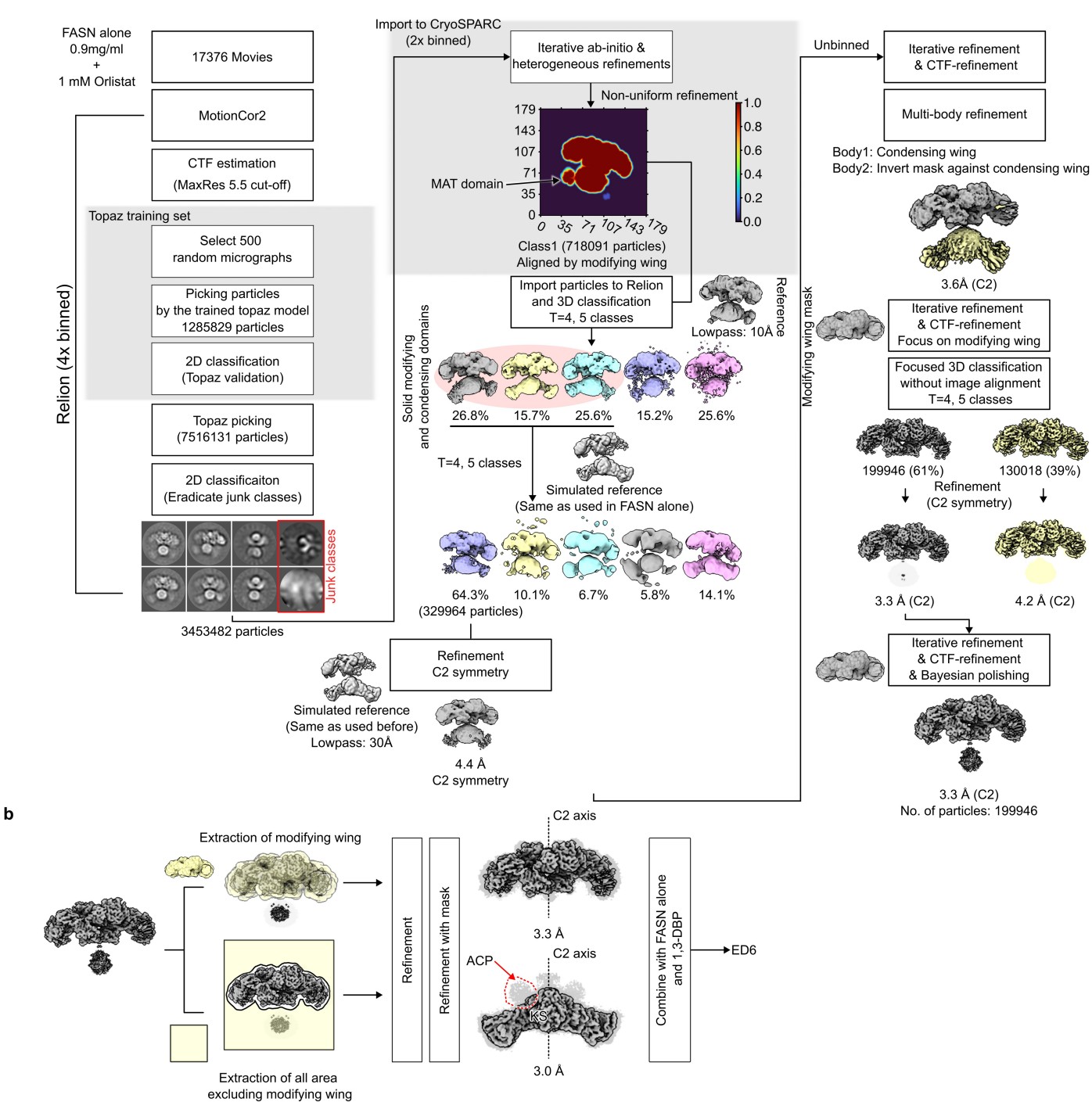

**Extended Data Fig. 3 | Flowchart of processing FASN with Orlistat.**
**a**, A detailed data processing flowchart illustrates image processing procedure of FASN incubated with Orlistat. **b**, A flowchart illustrating signal subtraction followed by separate alignment and refinement to determine final reconstructions of condensing and modifying wings independently from the same subset of particles, which are subjected to further classification, illustrated in Extended Data Fig. 6.

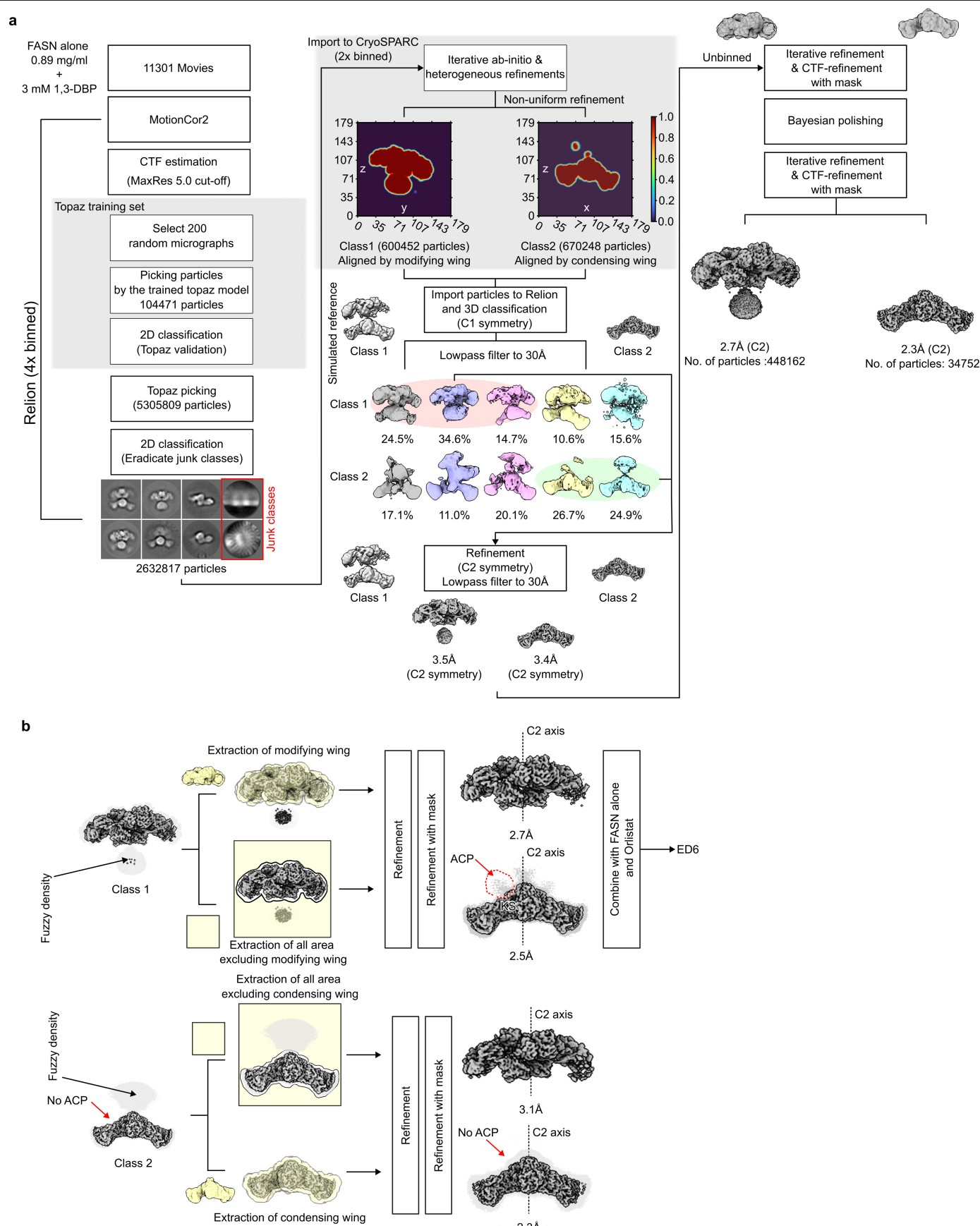

**Extended Data Fig. 4** | See next page for caption.

**Extended Data Fig. 4 | Flowchart of processing FASN with 1,3-DBP.**
**a**, A detailed data processing flowchart illustrates image processing procedure of FASN incubated with 1,3-DBP. Two classes of particles were identified, each produces a high-resolution reconstruction of one wing. **b**, A flowchart illustrating signal subtraction followed by separate alignment and refinement. From each class, final reconstructions of condensing and modifying wings are determined independently from the same subset of particles. Class with ACP is subjected to further classification, illustrated in Extended Data Fig. 6.

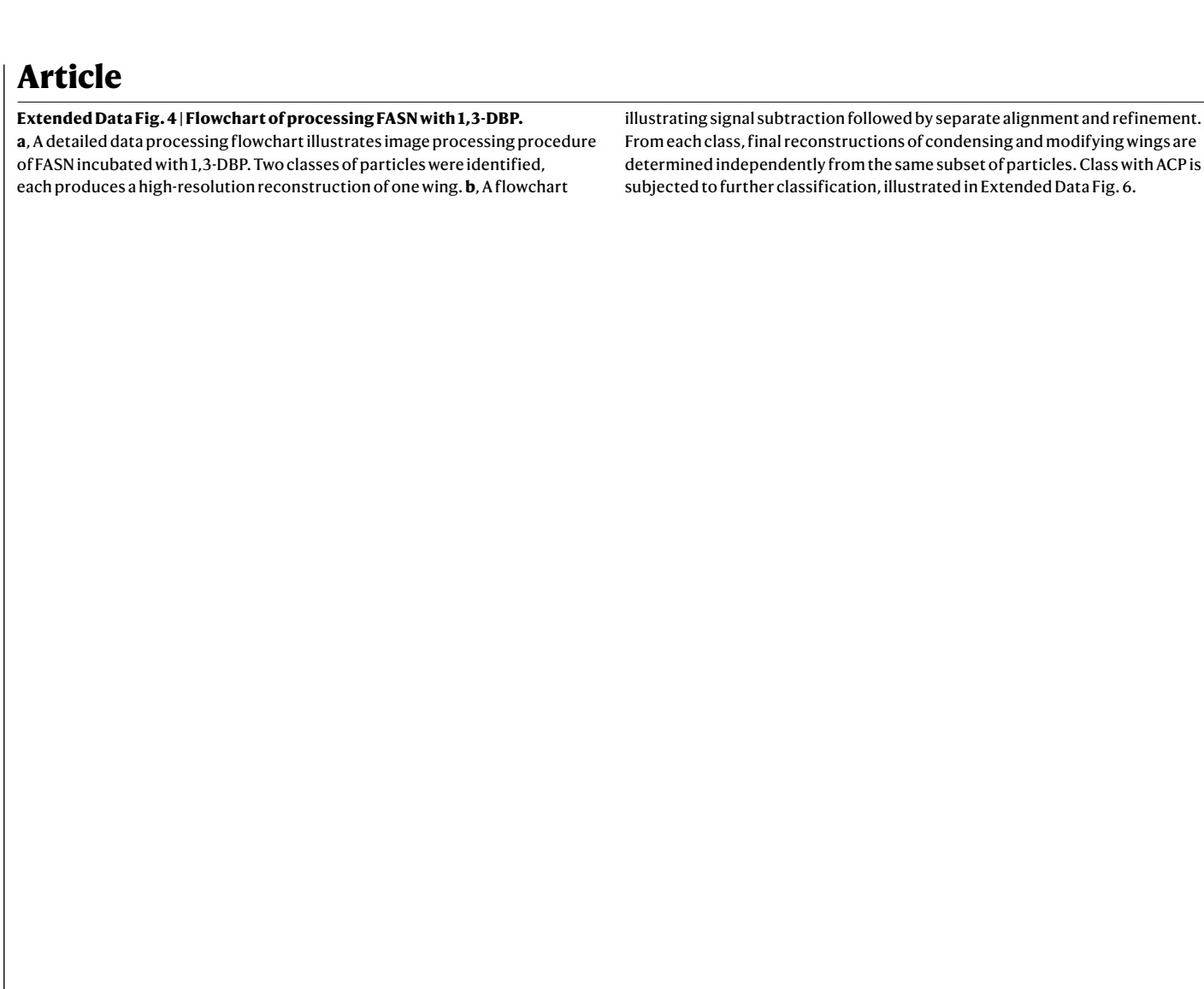

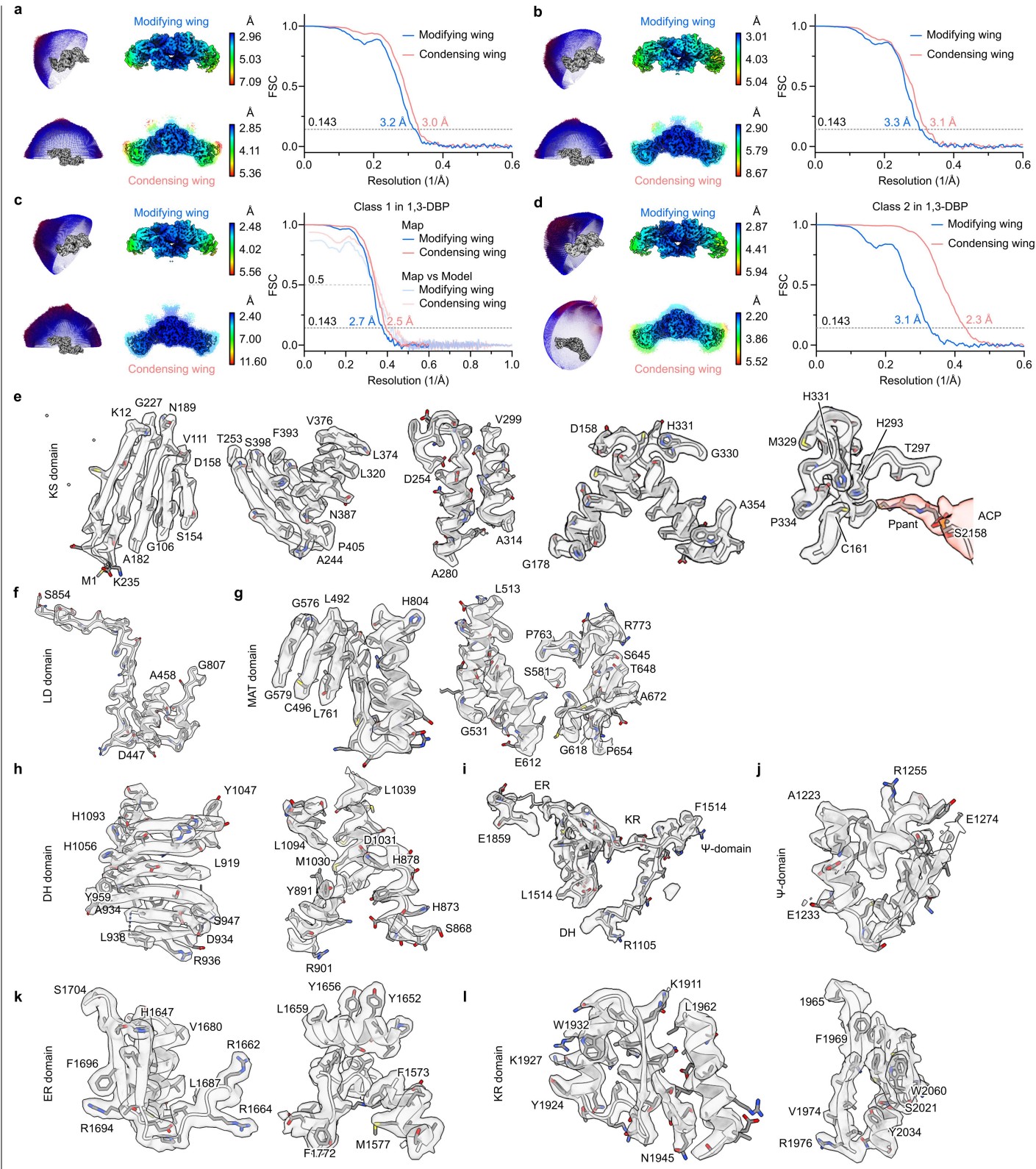

**Extended Data Fig. 5 | Resolution estimation and representative densities.**
**a-d**, Angular distribution, local resolution with resolution scale bar, and FSC curves of modifying (upper row and blue curve) and condensing (lower row and red curve) wings of FASN alone (**a**), FASN with Orlistat (**b**), FASN with 1,3-DBP with model-to-map FSC inserted (**c**), and the class identified from FASN with 1,3-DBP in which no ACP is seen (**d**). **e-l**, Representative densities of various domains from FASN with docked atomic model. Domains and amino acid sequence are labeled in each panel.

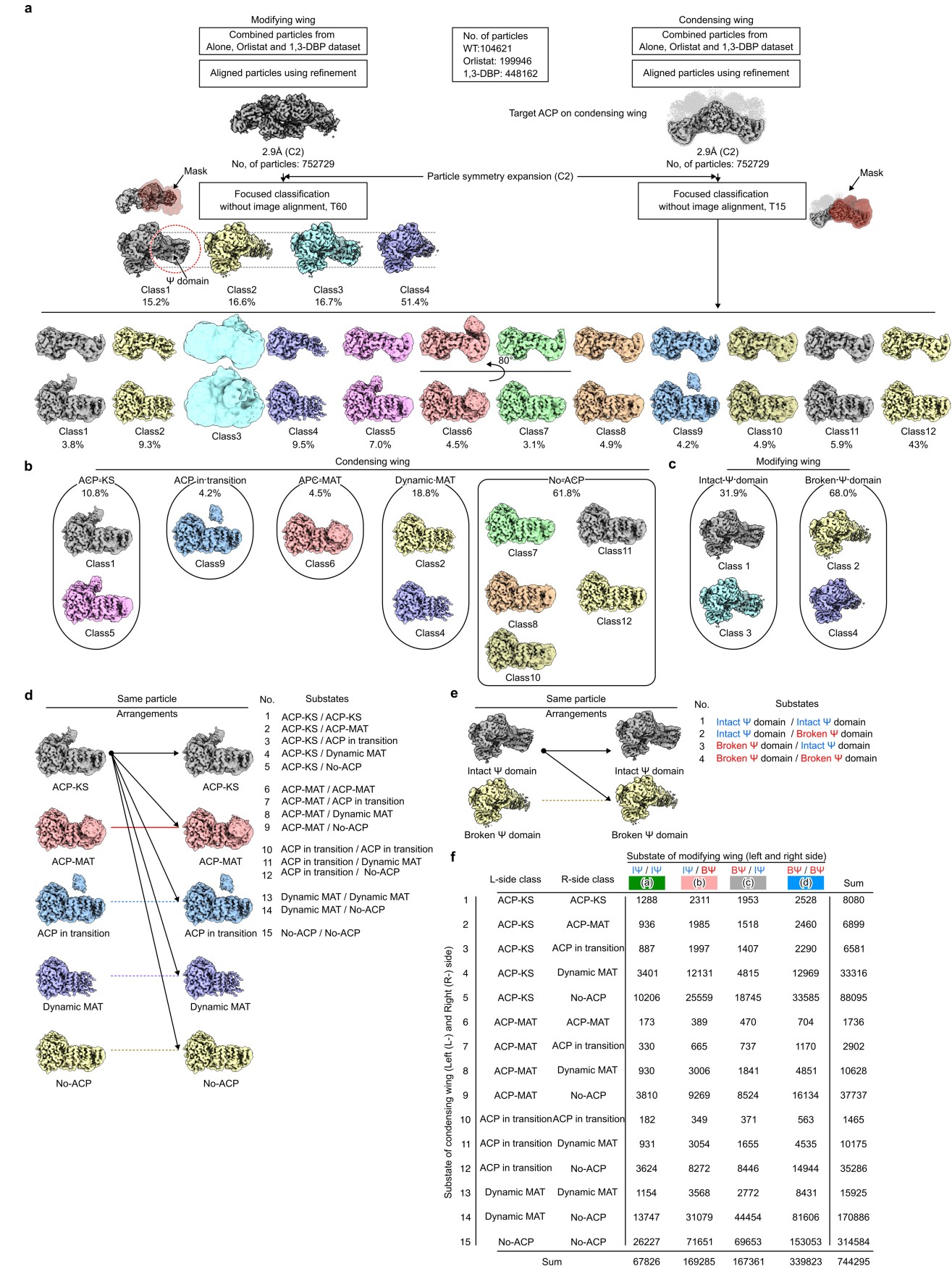

**Extended Data Fig. 6** | See next page for caption.

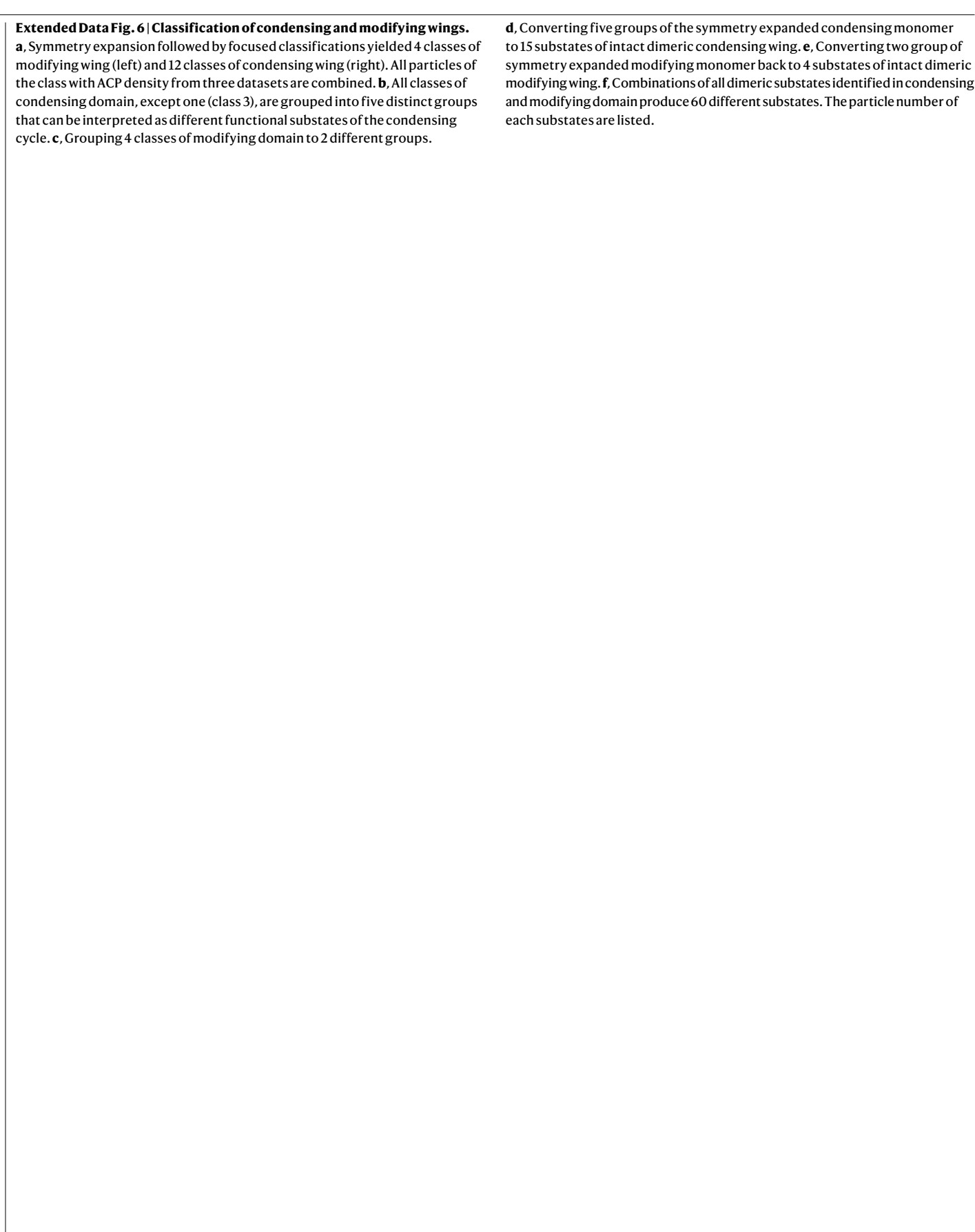

**Extended Data Fig. 6 | Classification of condensing and modifying wings.**
**a**, Symmetry expansion followed by focused classifications yielded 4 classes of modifying wing (left) and 12 classes of condensing wing (right). All particles of the class with ACP density from three datasets are combined. **b**, All classes of condensing domain, except one (class 3), are grouped into five distinct groups that can be interpreted as different functional substates of the condensing cycle. **c**, Grouping 4 classes of modifying domain to 2 different groups. **d**, Converting five groups of the symmetry expanded condensing monomer to 15 substates of intact dimeric condensing wing. **e**, Converting two group of symmetry expanded modifying monomer back to 4 substates of intact dimeric modifying wing. **f**, Combinations of all dimeric substates identified in condensing and modifying domain produce 60 different substates. The particle number of each substates are listed.

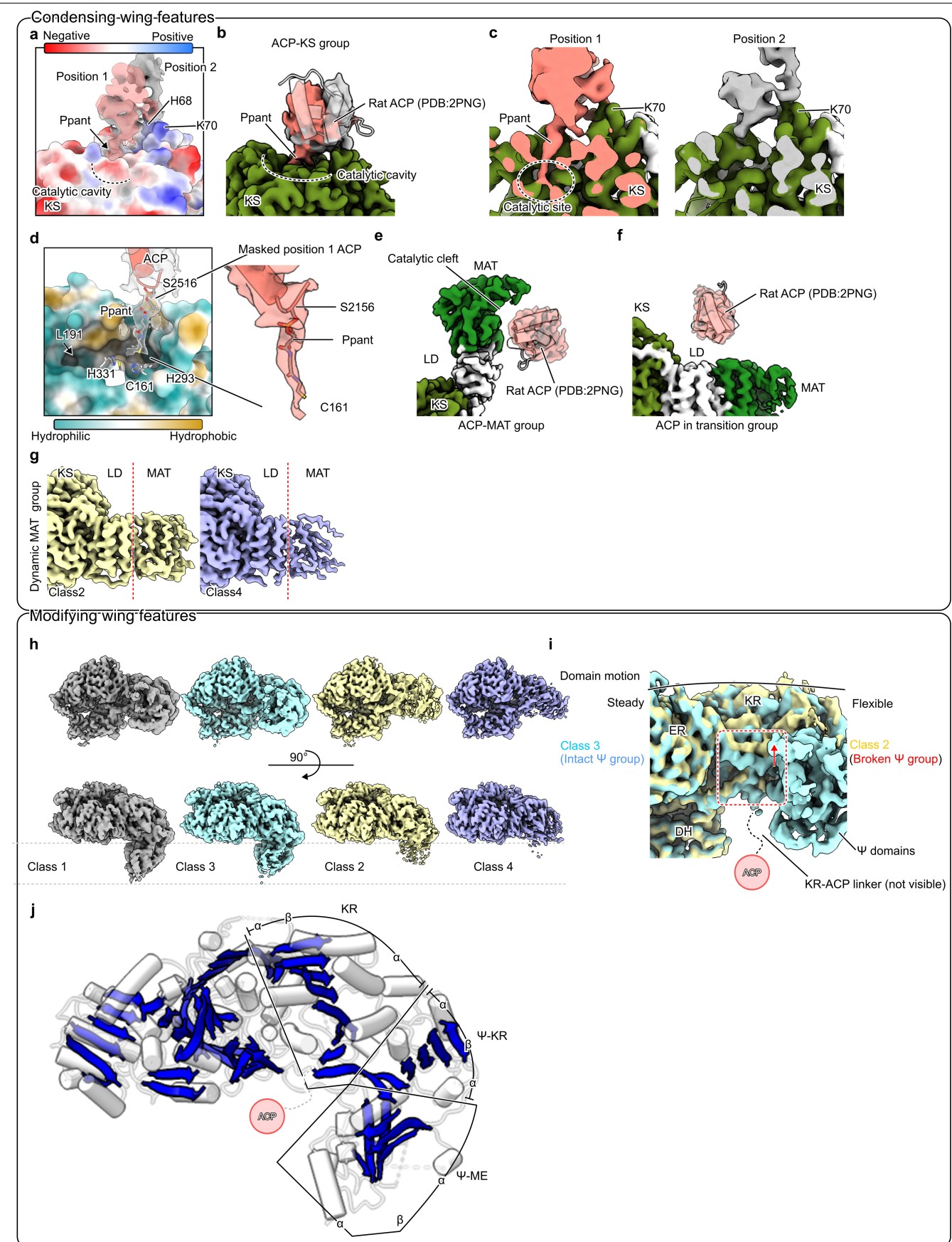

**Extended Data Fig. 7** | See next page for caption.

**Extended Data Fig. 7 | Selected structural features of condensing and modifying domain revealed from classification. a**, Two different engagements ACP (salmon and gray) of with the KS domain, which is shown as electrostatic surface. **b**, Rat ACP (PDB: 2PNG) is docked into the ACP density in ACP-KS group. **c**, ACP in gray shows stronger attachment on surface of KS domain while the other (salmon) shows a density deeply inserted into the catalytic cavity. **d**, The inserted density is interpreted as a Ppant to reaching the catalytic triad. **e** and **f**, Rat ACP (PDB: 2PNG) is docked into ACP density in ACP-MAT (**e**) and ACP in transition groups (**f**). Docking of Rat ACP to ACP density in three different groups are also shown in Supplementary Movies. **g**, Different level of dynamics of MAT catalytic cleft. **h**, Two different views (upper and lower rows) of modifying domain arranged from well resolved (left) to most dynamic (right) Ψ domain. **i**, Overlay the two classes in panel **h** reveals up-and-down motion of the Ψ domain. Link to ACP is marked as carbon. **j**, Structural architecture of the modifying domain, where α-helices are shown as grey cylinders and β-sheet are shown as blue ribbon.

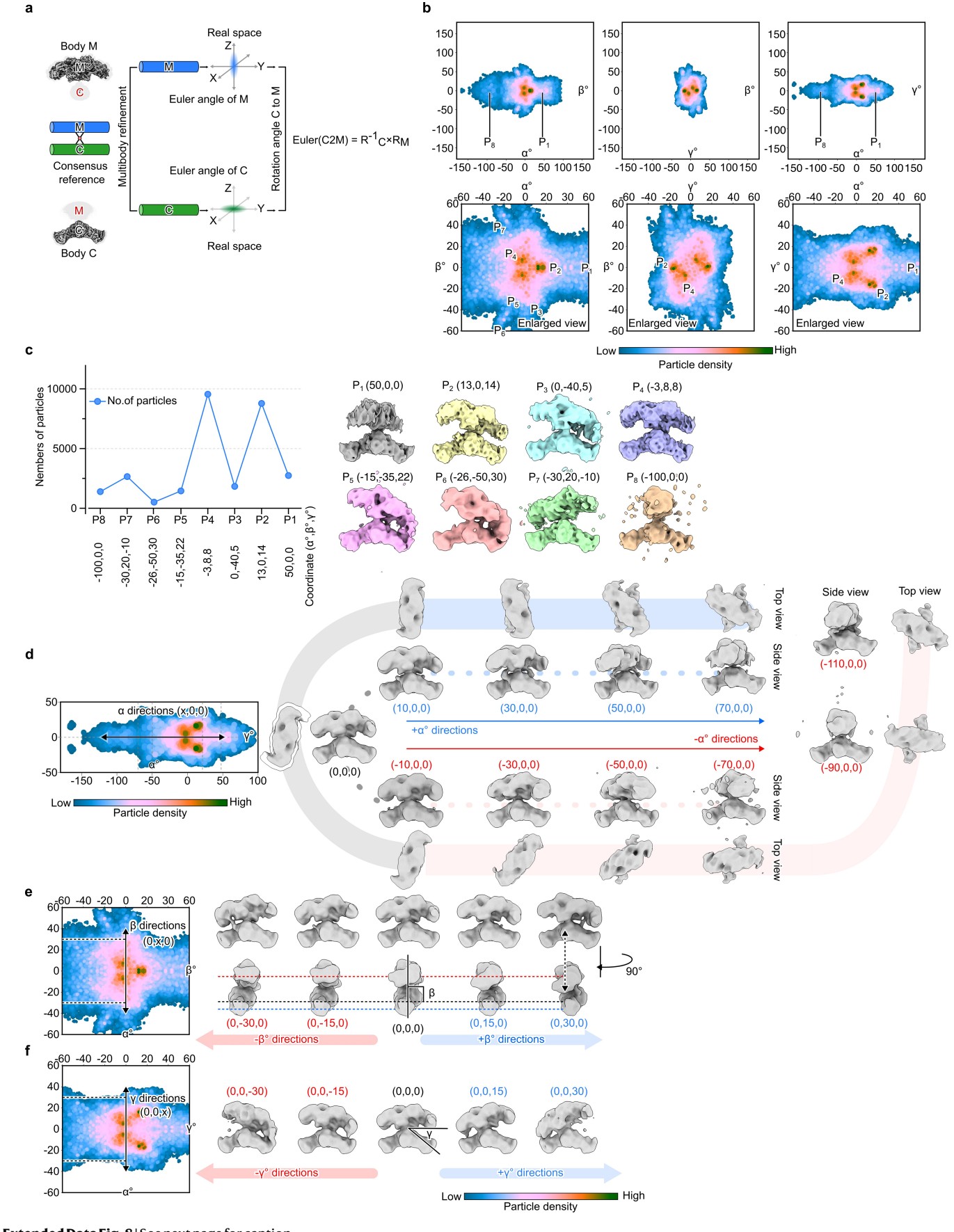

**Extended Data Fig. 8 |** See next page for caption.

**Extended Data Fig. 8 | Conformational landscape. a**, Schematic illustration of calculating the relative orientation between the modifying (M) and condensing (C) wings. **b**, 2D projection views of the landscape along three orthogonal axes (upper row) and enlarged view of the central regions (bottom row). **c**, Number of particles from 8 selected coordinates in the landscape (left), 3D reconstructions calculated from particles inside a sphere of 7° radius around these 8 coordinates. The number (n) of particles within the sphere, and the set resolution of the reconstruction (Å) calculated from these particles are $P_1$(n = 2744, 12 Å), $P_2$(8784, 10 Å), $P_3$(1839, 15 Å), $P_4$(9511, 10 Å) $P_5$(1454, 15 Å), $P_6$(512, 20 Å), $P_7$(2655, 15 Å), and $P_8$(1392, 15 Å), respectively (right). **d-f**, 3D reconstructions calculated from particle selected from different locations in the conformational landscape of the dataset combining three different samples, along α (a), β (b) and γ (c) directions. The reconstruction is calculated from locations at every 10° from +70° to −110° in α direction, from +30° to −30° at every 15° in β and γ directions. All reconstructions were calculated from a disc surrounding the selected coordinates with a radius of r, and total particle of n are (r = 6°, n = 1152), (6°, 1764), (6°, 1926), (6°, 5619), (6°, 4369), (6°, 3659), (6°, 1589), (6°, 939), (6°, 665), (6°, 921), and (6°, 905) along α direction, (r = 6°, n = 2037), (6°, 4031), (6°, 4369), (6°, 4031), (6°, 2037) along β direction, and (r = 6°, n = 512), (6°, 3536), (6°, 4369), (6°, 3536), (6°, 512) along γ direction. The maximum resolution of all reconstructions was set at 20 Å, and all maps are normalized and displayed at the same threshold.

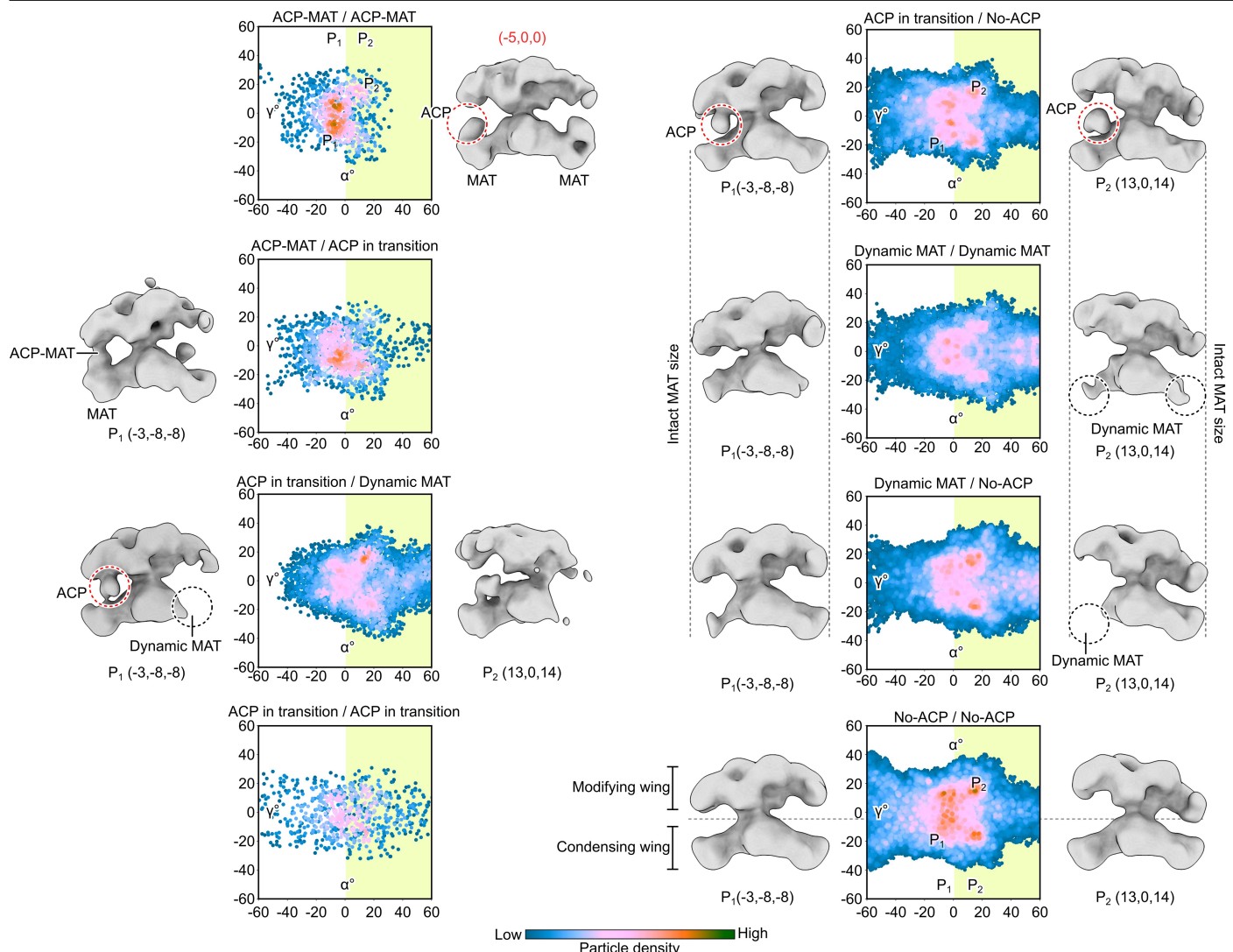

**Extended Data Fig. 9 | The conformational landscape of substates.**
Conformational landscape of individual substate identified through symmetry expanded classification with 3D reconstructions calculated from two fixed coordinates of selected substates. The reconstruction parameters of each substate are ACP-MAT/ACP-MAT (r = 14°, n = 172), ACP-MAT/ACP in transition $P_1$(12°, 156), ACP in transition/Dynamic MAT $P_1$(12°, 235) and $P_2$(12°, 245), ACP in transition/No-ACP $P_1$(9°, 409) and $P_2$(9°, 406), Dynamic MAT/Dynamic MAT $P_1$(10°, 312) and $P_2$(10°, 347), Dynamic MAT/No-ACP $P_1$(7°, 691) and $P_2$(7°, 885), and No-ACP/No-ACP $P_1$(7°, 1646) and $P_2$(7°, 1455). All reconstructions are normalized and visualized at threshold 3 with resolution of 25 Å. The overall conformational landscape is in the Supplementary Fig. 5.

# Extended Data Table 1 | Cryo-EM data collection, refinement and validation statistics

| | FASN alone | FASN + 1 mM Orlistat | FASN + 3 mM 1,3-DBP | FASN + 3 mM 1,3-DBP Class without ACP | FASN - combined modifying wing with classification | FASN - combined condensing wing with classification |
|---|---|---|---|---|---|---|
| Composite | (EMDB-44230) | (EMDB-44267) | (EMDB-44300) | (EMDB-44295) | (EMDB-44371) | (EMDB-44378) |
| Consensus | (EMDB-44231) | (EMDB-44268) | (EMDB-44301) | (EMDB-44296) | | (PDB: 9MJ9 – ACP position 1 with Ppant, Class5) |
| Focused modifying wing | (EMDB-44263) | (EMDB-44269) | (EMDB-44330) | (EMDB-44297) | | |
| Focused condensing wing | (EMDB-44264) | (EMDB-44270) | (EMDB-44329) | (EMDB-44298) | | |
| Modifying wing model | | | (PDB- 9B80) | | | |
| Condensing wing model | | | (PDB- 9B7Z) | | | |
| **Data collection and processing** | | | | | | |
| Magnification | 105,000x | 105,000x | 105,000x | 105,000x | 105,000x | 105,000x |
| Voltage (kV) | 300 | 300 | 300 | 300 | 300 | 300 |
| Electron exposure (e–/Å$^2$) | 45.8 | 45.8 | 45.8 | 45.8 | 45.8 | 45.8 |
| Defocus range (μm) | -0.8 ~ -1.5 | -0.8 ~ -1.5 | -0.8 ~ -1.5 | -0.8 ~ -1.5 | -0.8 ~ -1.5 | -0.8 ~ -1.5 |
| Pixel size (Å) | 0.835 (Sup res. 0.4175) | 0.835 (Sup res. 0.4175) | 0.835 (Sup res. 0.4175) | 0.835 (Sup res. 0.4175) | 0.835 (Sup res. 0.4175) | 0.835 (Sup res. 0.4175) |
| Symmetry imposed | C2 | C2 | C2 | C2 | C2 | C2 |
| Initial particle images (no.) | 5,896,040 | 7,516,131 | 5,305,809 | 5,305,809 | N/A | N/A |
| Final particle images (no.) | 104,621 | 199,946 | 448,162 | 347,527 | 752,729 | 752,729 |
| Map resolution (Å) | 3.1 Å (Consensus) 3.1 Å (Modifying) 3.0 Å (Condensing) | 3.3 Å (Consensus) 3.3 Å (Modifying) 3.0 Å (Condensing) | 2.7 Å (Consensus) 2.7 Å (Modifying) 2.5 Å (Condensing) | 2.3 Å (Consensus) 3.1 Å (Modifying) 2.3 Å (Condensing) | 2.9 Å | 2.9 Å |
| FSC threshold | 0.143 | 0.143 | 0.143 | 0.143 | 0.143 | 0.143 |

| **Refinement** | Models against to class 1 of FASN with 1,3-DBP | Model against ACP position 1, class5 |
|---|---|---|
| Initial model used (PDB code) | 2VZ8 (Modifying domain) / 3HHD (Condensing domain) | 9B7Z and 2PNG |
| Model resolution (Å) | 3.0 (Modifying wing) / 2.9 (Condensing wing) | N/A |
| FSC threshold | 0.5 | N/A |
| Map sharpening $B$ factor (Å$^2$) | -111.795 (Modifying wing) / -118.085 (Condensing wing) | N/A |
| Model composition | | N/A |
| Non-hydrogen atoms | 16154 (Modifying wing) / 12942 (Condensing wing) | 6576 |
| Protein residues | 2192 (Modifying wing) / 1708 (Condensing wing) | 870 |
| Ligands | | Ppant 1 |
| $B$ factors (Å$^2$) | | |
| Protein | 30.00/326.64/113.10 (Modifying wing) | 102.06/339.31/150.69 |
| Ligand | 44.54/248.07111.64 (Condensing wing) | 301.05/301.05/301.05 |
| R.m.s. deviations | | |
| Bond lengths (Å) | 0.003 (0) (Modifying wing)/ 0.008 (0) (Condensing wing) | 0.003 (0) |
| Bond angles (°) | 0.558 (2) (Modifying wing) / 1.275 (20) (Condensing wing) | 0.553 (4) |
| Validation | | |
| MolProbity score | 1.38 (Modifying wing) / 1.59 (Condensing wing) | 1.39 |
| Clashscore | 4.63 (Modifying wing) / 6.01 (Condensing wing) | 5.59 |
| Poor rotamers (%) | 0.00 (Modifying wing) / 0.29 (Condensing wing) | 0.1 |
| Ramachandran plot | | |
| Favored (%) | 97.23 (Modifying wing) / 96.13 (Condensing wing) | 0.00 |
| Allowed (%) | 2.77 (Modifying wing) / 3.87 (Condensing wing) | 2.42 |
| Disallowed (%) | 0.00 (Modifying wing) / 0.00 (Condensing wing) | 97.58 |

The table lists parameters used or obtained in cryo-EM experiments. It includes microscope settings for data acquisition, parameter settings for image processing, statistics of reconstructions, map and model validations.

| | |
|---|---|

# Reporting Summary

Please do not complete any field with "not applicable" or n/a. Refer to the help text for what text to use if an item is not relevant to your study.
For final submission: please carefully check your responses for accuracy; you will not be able to make changes later.

## Statistics

For all statistical analyses, confirm that the following items are present in the figure legend, table legend, main text, or Methods section.

| n/a | Confirmed | |
|---|---|---|
| ☐ | ☑ | The exact sample size ($n$) for each experimental group/condition, given as a discrete number and unit of measurement |
| ☐ | ☑ | A statement on whether measurements were taken from distinct samples or whether the same sample was measured repeatedly |
| ☐ | ☑ | The statistical test(s) used AND whether they are one- or two-sided<br>*Only common tests should be described solely by name; describe more complex techniques in the Methods section.* |
| ☑ | ☐ | A description of all covariates tested |
| ☑ | ☐ | A description of any assumptions or corrections, such as tests of normality and adjustment for multiple comparisons |
| ☐ | ☑ | A full description of the statistical parameters including central tendency (e.g. means) or other basic estimates (e.g. regression coefficient) AND variation (e.g. standard deviation) or associated estimates of uncertainty (e.g. confidence intervals) |
| ☑ | ☐ | For null hypothesis testing, the test statistic (e.g. $F$, $t$, $r$) with confidence intervals, effect sizes, degrees of freedom and $P$ value noted<br>*Give P values as exact values whenever suitable.* |
| ☑ | ☐ | For Bayesian analysis, information on the choice of priors and Markov chain Monte Carlo settings |
| ☑ | ☐ | For hierarchical and complex designs, identification of the appropriate level for tests and full reporting of outcomes |
| ☑ | ☐ | Estimates of effect sizes (e.g. Cohen's $d$, Pearson's $r$), indicating how they were calculated |

*Our web collection on statistics for biologists contains articles on many of the points above.*

## Software and code

Policy information about availability of computer code

| Data collection | CryoEM data was collected on a Titan Krios using SerialEM. |
|---|---|
| Data analysis | MotionCor2, Ctffind4.1.14, Relion 4.0.1-cu11.8, CryoSparc 4.2, Grapad Prism10.2.2, Phenix1.21, Coot0.9.6, cryoDRGN-2.3.0, Protein Prospector v6.6.4 and ChimeraX1.5, CryoROLE (released in GItHub, https://github.com/yifancheng-ucsf/cryorole) |

For manuscripts utilizing custom algorithms or software that are central to the research but not yet described in published literature, software must be made available to editors and reviewers. We strongly encourage code deposition in a community repository (e.g. GitHub). See the Nature Portfolio guidelines for submitting code & software for further information.

## Data

Policy information about availability of data

All manuscripts must include a data availability statement. This statement should provide the following information, where applicable:
- Accession codes, unique identifiers, or web links for publicly available datasets
- A description of any restrictions on data availability
- For clinical datasets or third party data, please ensure that the statement adheres to our policy

Composite maps of FASN alone (Composite map: EMDB-44230, consensus map: EMDB-44231, focused modifying domain map: EMDB-44263, and focused condensing domain map: EMDB-44264), with Orlistat (EMDB-44267, EMDB-44268, EMDB-44269 and EMDB-44270), and with 1,3-DBP (EMDB-44300, EMDB-44301, EMDB-44330 and EMDB-44329) are deposited into EMDB database. The atomic models (modifying domain: PDB-9B80, and condensing domain: PDB-9B7Z) are deposited, accordingly. Also, maps of condensing and modifying wings calculated from combining all three datasets are also deposited together with their symmetry expanded 3D focused classification maps: modifying domain (EMDB-44371) and condensing domain (EMDB-44378). The symmetry expanded map of ACP-KS group is an additional map associated with EMDB-44378 and linked with the atomic model of condensing wing with Ppant and partially docked ACP (PDB-9MJ9). pYC homology directed repair donor plasmid is available in Addgene (https://www.addgene.org/Yifan_Cheng/). Homo sapiens (HEK cells) SwissProt.2022.05.26. random.concat database (containing 567483 entries) is used for LC-MS analysis.

## Research involving human participants, their data, or biological material

Policy information about studies with [human participants or human data](). See also policy information about [sex, gender (identity/presentation), and sexual orientation]() and [race, ethnicity and racism]().

| | |
|---|---|
| Reporting on sex and gender | N/A |
| Reporting on race, ethnicity, or other socially relevant groupings | N/A |
| Population characteristics | N/A |
| Recruitment | N/A |
| Ethics oversight | N/A |

Note that full information on the approval of the study protocol must also be provided in the manuscript.

# Field-specific reporting

Please select the one below that is the best fit for your research. If you are not sure, read the appropriate sections before making your selection.

☑ Life sciences          ☐ Behavioural & social sciences          ☐ Ecological, evolutionary & environmental sciences

For a reference copy of the document with all sections, see [nature.com/documents/nr-reporting-summary-flat.pdf]()

# Life sciences study design

All studies must disclose on these points even when the disclosure is negative.

| | |
|---|---|
| Sample size | Sample sizes were not predetermined. For CryoEM, enough particles were collected to achieve best resolution. For statistic analysis, enough replicates were performed for robust statistic analysis. |
| Data exclusions | Microgrpahs were excluded based on rlnCtfMaxResolution value (cutt-off 5.0 to 5.5)  after CTF estimation. Individual particles were dicarded by 2D and 3D classification results based on  the standard in the field of single-particle cryoEM. All data processing pipelines are shown in Extended Data Figure 2 to 4 ,and 6. |
| Replication | Knock-in of affinity tag, immunoblotting, protein purification and FSEC analysis were conducted multiple times using different batch of cells. NADPH consumption, MS validation, cell growth assay were repeated listed number on the corresponding figure legends. |
| Randomization | Topaz training was conducted on randomly selected micrographs. FSC was calculated by randomly split the final particle set into 2. Randomization is not needed for other experiments. |
| Blinding | For all experiments performed in this study, blinding is not required. Becuase the study is discovery and mechanism focused, all experiements were designed as open labeled. |

# Behavioural & social sciences study design

All studies must disclose on these points even when the disclosure is negative.

| | |
|---|---|
| Study description | |
| Research sample | |
| Sampling strategy | |
| Data collection | |
| Timing | |
| Data exclusions | |
| Non-participation | |
| Randomization | |

# Ecological, evolutionary & environmental sciences study design

All studies must disclose on these points even when the disclosure is negative.

| | |
|---|---|
| Study description | |
| Research sample | |
| Sampling strategy | |
| Data collection | |
| Timing and spatial scale | |
| Data exclusions | |
| Reproducibility | |
| Randomization | |
| Blinding | |

Did the study involve field work?  ☐ Yes  ☐ No

## Field work, collection and transport

| | |
|---|---|
| Field conditions | |
| Location | |
| Access & import/export | |
| Disturbance | |

# Reporting for specific materials, systems and methods

We require information from authors about some types of materials, experimental systems and methods used in many studies. Here, indicate whether each material, system or method listed is relevant to your study. If you are not sure if a list item applies to your research, read the appropriate section before selecting a response.

### Materials & experimental systems

| n/a | Involved in the study |
|---|---|
| ☐ | ☑ Antibodies |
| ☐ | ☑ Eukaryotic cell lines |
| ☑ | ☐ Palaeontology and archaeology |
| ☑ | ☐ Animals and other organisms |
| ☑ | ☐ Clinical data |
| ☑ | ☐ Dual use research of concern |
| ☑ | ☐ Plants |

### Methods

| n/a | Involved in the study |
|---|---|
| ☑ | ☐ ChIP-seq |
| ☑ | ☐ Flow cytometry |
| ☑ | ☐ MRI-based neuroimaging |

## Antibodies

| | |
|---|---|
| Antibodies used | Anti-FLAG and anti-GAPDH antibodies were used for immunoblotting. |
| Validation | Antibodies: anti-FLAG-peroxidase (HRP) (Sigma-Aldrich, A8592), anti-GAPDH (Proteintech, CL594-60004), and anti-mouse IgG-HRP (Invitrogen, 31450) |

## Eukaryotic cell lines

Policy information about cell lines and Sex and Gender in Research

| Cell line source(s) | Wildtype HEK293T was purchased from ATCC (CRL-3216) |

| Authentication | No authentication was performed for the purchased cell line. |

| Mycoplasma contamination | No mycoplasma contamination test was conducted. |

| Commonly misidentified lines (See ICLAC register) | No commonely misidentified cell lines were used in this study. |

## Palaeontology and Archaeology

| Specimen provenance | |

| Specimen deposition | |

| Dating methods | |

☐ Tick this box to confirm that the raw and calibrated dates are available in the paper or in Supplementary Information.

| Ethics oversight | |

Note that full information on the approval of the study protocol must also be provided in the manuscript.

## Animals and other research organisms

Policy information about studies involving animals; ARRIVE guidelines recommended for reporting animal research, and Sex and Gender in Research

| Laboratory animals | |

| Wild animals | |

| Reporting on sex | |

| Field-collected samples | |

| Ethics oversight | |

Note that full information on the approval of the study protocol must also be provided in the manuscript.

## Clinical data

Policy information about clinical studies
All manuscripts should comply with the ICMJE guidelines for publication of clinical research and a completed CONSORT checklist must be included with all submissions.

| Clinical trial registration | |

| Study protocol | |

| Data collection | |

| Outcomes | |

## Dual use research of concern

Policy information about dual use research of concern

### Hazards

Could the accidental, deliberate or reckless misuse of agents or technologies generated in the work, or the application of information presented in the manuscript, pose a threat to:

| No | Yes | |
|----|-----|---|
| ☐ | ☐ | Public health |
| ☐ | ☐ | National security |
| ☐ | ☐ | Crops and/or livestock |
| ☐ | ☐ | Ecosystems |
| ☐ | ☐ | Any other significant area |

## Experiments of concern

Does the work involve any of these experiments of concern:

| No | Yes | |
|----|-----|---|
| ☐ | ☐ | Demonstrate how to render a vaccine ineffective |
| ☐ | ☐ | Confer resistance to therapeutically useful antibiotics or antiviral agents |
| ☐ | ☐ | Enhance the virulence of a pathogen or render a nonpathogen virulent |
| ☐ | ☐ | Increase transmissibility of a pathogen |
| ☐ | ☐ | Alter the host range of a pathogen |
| ☐ | ☐ | Enable evasion of diagnostic/detection modalities |
| ☐ | ☐ | Enable the weaponization of a biological agent or toxin |
| ☐ | ☐ | Any other potentially harmful combination of experiments and agents |

# Plants

| | |
|---|---|
| Seed stocks | |
| Novel plant genotypes | |
| Authentication | |

# ChIP-seq

## Data deposition

☐ Confirm that both raw and final processed data have been deposited in a public database such as GEO.

☐ Confirm that you have deposited or provided access to graph files (e.g. BED files) for the called peaks.

| | |
|---|---|
| Data access links<br>*May remain private before publication.* | |
| Files in database submission | |
| Genome browser session<br>(e.g. UCSC) | |

## Methodology

| | |
|---|---|
| Replicates | |
| Sequencing depth | |
| Antibodies | |
| Peak calling parameters | |
| Data quality | |

Software [                                    ]

# Flow Cytometry

## Plots

Confirm that:

☐ The axis labels state the marker and fluorochrome used (e.g. CD4-FITC).

☐ The axis scales are clearly visible. Include numbers along axes only for bottom left plot of group (a 'group' is an analysis of identical markers).

☐ All plots are contour plots with outliers or pseudocolor plots.

☐ A numerical value for number of cells or percentage (with statistics) is provided.

## Methodology

Sample preparation [                                    ]

Instrument [                                    ]

Software [                                    ]

Cell population abundance [                                    ]

Gating strategy [                                    ]

☐ Tick this box to confirm that a figure exemplifying the gating strategy is provided in the Supplementary Information.

# Magnetic resonance imaging

## Experimental design

Design type [                                    ]

Design specifications [                                    ]

Behavioral performance measures [                                    ]

Imaging type(s) [                                    ]

Field strength [                                    ]

Sequence & imaging parameters [                                    ]

Area of acquisition [                                    ]

Diffusion MRI    ☐ Used    ☐ Not used

## Preprocessing

Preprocessing software [                                    ]

Normalization [                                    ]

Normalization template [                                    ]

Noise and artifact removal [                                    ]

Volume censoring [                                    ]

## Statistical modeling & inference

Model type and settings [                                    ]

Effect(s) tested [                                    ]

Specify type of analysis:  ☐ Whole brain   ☐ ROI-based   ☐ Both

Statistic type for inference

(See Eklund et al. 2016)

Correction

## Models & analysis

| n/a | Involved in the study |
|---|---|
| ☐ | ☐ Functional and/or effective connectivity |
| ☐ | ☐ Graph analysis |
| ☐ | ☐ Multivariate modeling or predictive analysis |

Functional and/or effective connectivity

Graph analysis

Multivariate modeling and predictive analysis

