## [Peer Review File · Nature]

Structural dynamics of human fatty acid synthase in the condensing cycle

Corresponding Author: Dr Yifan Cheng

Version 0:

Reviewer comments:

Referee #1

(Remarks to the Author)

The manuscript by Choi et al. reports the conformational variability of the human FASN using single particle cryogenic electron microscopy (cryo-EM). FASN has been purified endogenously from HEK293, and treated with Orlistat, a drug against the TE domain of human FASN, and with 1,3-dibromopropanone (DBP), a crosslinker used for FAS analysis before. Three datasets were received from preparations: (i) no treatment - termed FASN alone, (ii) Orlistat and (iii) and 1,3-dibromopropanone (DBP). The authors focused on the characterization of the conformational dynamics of human FASN using, which is relevant for understanding acyl carrier protein (ACP) mediated substrates shuttling. Their structure scan visualize ACP docking to the catalytic KS or MAT domains, representing key functional substates during the carbon-carbon bond forming part of the fatty acid cycle.

A crucial aspect of this manuscript is the conformational analysis by assigning Euler angles to the two wings, treated as rigid bodies enabling to assign conformational landscapes of particles that differ in the orientations between the condensing and modifying wings. One of the main findings is that only small conformational variations are necessary for conducting fatty acid synthesis, contradicting - as the authors argue - the "swiveling and swinging" model. Instead, FASN can be fully functional with subtle "wiggling and jiggling" motions. The authors argue that the type of restricting conformational variability, the "wiggling and jiggling" model, is reflecting the real conformational variability of the protein in the crowded cellular environment. In a further crucial aspect of the manuscript, the authors captured FASN in specific functional states during the condensation cycle and structurally described a plethora of FASN substates (in which one reaction chamber is engaged in MAT or KS catalysis). They found that the enzymatic reactions carried out by the two monomers do not correlate.

The manuscript is very technical, with the functionally relevant statements being derived from the analysis of the conformational landscape and substate analysis. The paper is not easy to read for molecular biologist. Rather, in its current form, it seems specialized for the cryo-EM community. Additionally, the analysis does not provide much new insight in the function of mammalian FAS. While it is undoubtedly interesting to map the conformational variability of mammalian FAS, on the example of FASN, which has not done in this detail before, it does not offer a new understanding of the protein's molecular mechanism(s). Different than the authors' claim, there is no broader consensus that large conformational variability is essential for mammalian FAS function a for mammalian FAS („swiveling and swinging model"). In this regard, the view on the evolutionarily related iterative PKS is insightful: Iterative PKSs and mammalian FAS share the same X-shaped fold and perform the same type of chemistry. Iterative PKSs have recently been structurally characterized. Intriguingly, for the iterative PKS CalA3 <https://doi.org/10.1038/s41467-023-36989-w> and LovB <https://doi.org/10.1038/s41467-021-21174-8> conformational variability of the two wings could not be observed in cryo-EM (no swiveling and swinging). Structural elements, in the waist region between the wings, were accounted to cause rigidity. This implies that the X-shaped fold, seen in CalA3 and LovB, mammalian FAS and other related proteins can perform synthesis without the need for large conformational variability. The finding that the reaction chambers essentially work independently, matches the current perception of mammalian FAS mediated FA-synthesis. The two polypeptides of a dimeric mammalian FAS just share little interfaces (mainly via KS) such that is difficult to imagine how coordination of the reaction chamber could arise from this.

A higher-order conformational variability of the X-shaped fold, like swiveling and swinging by FASN, increases the likelihood of interactions with the domains within this confined space. The idea that higher order conformational variability can improve efficiency of such FASs and PKSs was termed „medium conformational constraint model" (see Weissman <https://doi.org/10.1038/nchembio.1883> and Buychuihan et al. <https://doi.org/10.1002/anie.202312476>).

While there is the impression, as detailed above, that biological relevance of the conformational landscape analysis is overestimated, other aspects remain almost unmentioned. Particularly, data contains structural information to the interaction between ACP and KS, and ACP and MAT. These interactions are highly interesting, as comprising the key steps in carbon-carbon bond formation. For example, (i) the MAT domain is highly interesting, because it has been characterized as extraordinarily fast compared to other transferases in this PKSs. How does the interface of ACP with MAT look like, is it minimal to feature such high turnover rates. (ii) Is there any explanation for the two different ACP docking site at the KS domain? (iii) How does the ACP in the floating state look like, and what does such a state mean for FASN function. It is noted that for modular PKSs these interactions have recently been characterized in structure, such that FAS and PKS could be compared in this respect.

In sum, this manuscript is very technical in its current form. However, data itself holds very intriguing information. It is recommended to streamline the conformational landscape analysis as well as better explore the yet under-examined data on the domain-domain interaction.

list of comments:

-terminology: „domains“ should not be used for condensing and modifying substructures, it describes the enzymatic domain. Usually „wing“ or „region“ is used. Throughout this review, the established terminology is used for describing the regions of FAS.

-line 46: I recommend saying „two key processes“; „step“ is mostly used for a single reactions

-line 53: „in bacteria and plants, enzyme propagating a fatty acid synthesis are...“

-line 55: not just mammals, but „animals“ or „metazoa“, please note that mammalian FAS should better be „animal FAS“ or „metazoan FAS“

-line 60: the KS condenses KS-bound ace(ty)l and ACP-bound malonyl, but not their CoA esters

-line 66: „The TE domain cleaves off the final product from the ACP when it reaches the desired length of 16 to 18 acyl chains“ is confusing. Better is „The TE domain cleaves off acyl chains from the ACP when they reach the desired length of 16 to 18 carbons“.

-paragraph starting line 110: „...refined the condensing and modifying domains separately from the same subsets of particles...“. Does that mean the refinement was done separately, but the integrity of the particle was maintained in the sense that the actual relative orientation of the wings is preserved? Please clarify this point

-line 71: „dimer interface“ can be confused with the interface between two protomers. I suggest to structure the sentence like this: „...and that the two dimeric domains (better wings) are arranged in parallel with a short linker between them.“

-line 120: acetyl is just shuttled during priming; thus, in a condensation cycle not just acetyl but also acyl is loaded into the KS active site, and then condensed with enolate originating from malonyl.

-line 125: it says 5 functional groups, but just four are listed in the following; according to Ext Fig. 6, it seems that class „dynamic MAT“ has been omitted. This is misleading and should be adjusted.

line 129: ACP binding is mediated by electrostatic interactions, which has been analyzed in yeast FAS by Gipson et al. <https://doi.org/10.1073/pnas.0913547107> and Anselmi et al. <https://doi.org/10.1021/ja103354w>; before analyzed at higher resolution in ref 29.

-paragraph starting line 130: How can the two distinct positions of ACP binding to the KS be explained. The authors should analyze whether one of these positions is induced by the 1,3-DBP crosslink.

-paragraph starting line 130: can the interface between domains be solved at near atomic detail?

-again paragraph starting line 130: The argument that tunnels of different lengths are likely to accommodate partially synthesized acyl chains of different lengths during the condensing cycle, disagrees with the current understanding of chain length regulation. Structural data with acyl chains of different chain length have so far not identified binding to different cavities, see e.g. ref 16 with C8 and C12 bound to KAS 1 of E. coli. Note that a KS with bound octanoic acid has been solved structurally by Rittner et al. <https://doi.org/10.1002/pro.3797>. A superposition with the C8:KS data would help in visualizing the binding tunnel (see Extended Data Fig. 6d). Is the mixture of charged and hydrophobic residues supposed to support any binding of partially synthesized acyl chains. These points should be explained in more detailed and connected to the current understanding of substrate binding to KS. Acyl chain binding to KS, including chain length regulation, has been reviewed by Heil et al. <https://doi.org/10.1002/cbic.201800809>.

-paragraph starting line 140: how should a floating ACP group be trapped in a defined position and orientation such that it can be traced in density?

-paragraph starting line 140: can the interface between domains be solved at near atomic detail?

-paragraph starting line 145: Is there any biological interpretation for the dynamic MAT? Why it should be associated with acetyl- or malonyl-CoA?

-paragraph starting with line 169: There is no „swiveling and swinging model“ that suggests that rotation of the two wings is essential for activity (see comment above).

-line 163: how is the assembly of condensing and modifying wing done in terms of their relative orientation?

-line 181: what does „same subset of particles“ means?

-line 183: Calculation of the Euler angles, assigned to each wing, is difficult to grasp in its details. Which particles are used for determining the relative orientation of wings. See also my comment to line 110.

-line 196: The authors interpret landscape of conformations as constraint by ACP engagement. However, this would just be possible if the ACP, as part of the modifying wing, binds to the catalytic domains of the condensing wing. As just a subset of particles show ACP binding to KS and MAT, this interpretation is incomprehensible.

-line 209: The authors should disclose the calculation leading to 76 Å? For the argument of constraining conformation landscape, this distance is treated as a threshold, such that details to its calculation are important.

- line 210: The authors should also explain that A2115 is the attachment point for ACP.
- line 268: Joshi et al did not suggest, that ACP can only engage with one monomer. They proved, that the mFAS is active within one monomer, but they also clearly state, that ACP can interact inter- and intra-subunit.
- line 309: Unclear meaning: With less fatty acids in the nutrients, FASN would be less constrained in conformational variability? Or: The low efficiency of mammalian FAS is the reason that fatty acids are supplied by nutrition rather than FASN derived?
- line 311-314: Further unclear point: Higher copy numbers of FASN will of course not change crowded cytosolic environment.
- line 315: very speculative and not supported by any data: given the references, FASN may form filaments?

Additional points:

- Has the tip of the PNS been analyzed for any density of an acyl chain bound to the PNS? Endogenously purified FAS should probably have intermediates of acyl chains bound. The location probability of the ACP in the complex might be affected by the substrate, that is bound to the PNS. This possibility should be mentioned at some point.
- Enzymatic activity is an ideal parameter for checking protein quality. Have protein preparations been tested in activity?
- The claim that large rotational motion between the condensing and modifying wing is unsupported by functional data. it would have been interesting to see enzymatic data on human FASN restricted in conformational variability. A possibility for engineering may be the exchange of the flexibly linkers with alpha helical element as found in LovB of MSA-line PKS.
- Has the strep tag been used for purification? Or for anything else?
- The authors should please comment on the absence of structures that show the ACP at the modifying domains.
- Fig. 2e suggests that no protein-protein interactions of ACP and KS were found. The authors are asked to comment on that.

- line 125: typo: „APC“ instead of “ACP”
- Fig. 1b: what does C16 (yes) and C16(no) mean?
- Figure 4d: Numbers on the axis have shifted.
- Fig. 4f: The purpose of the dashed lines (barely visible) is unclear.
- Fig. 5a: It would be appreciated if the substate groups P1-P4 are marked with a dot or circle in the landscape.
- Ext. Data Fig. 1c: correct CO₂
- Ext Data Fig. 10; typo „Daa“
- these references seem misplaced:
(ref 15): cited in the context of type II FAS: Paper is about a cryo-EM structure of Pks13, which is a type I PKS.
(ref 16): cited in the context of catalytic functions in the modifying region: Fantastic paper about the mechanism of the KS, analyzed with different acyl chains and type II KS from E. coli.

Referee #2

(Remarks to the Author)

Understanding the dynamics of the multifunctional vertebrate fatty acid synthase (FASN) and its biochemical (and perhaps even biological) implications is an important scientific challenge. In their manuscript, Choi and Li et al. address this challenge using single-particle cryogenic electron microscopy (cryo-EM) to resolve various conformational states of intact human FASN. To reduce heterogeneity, the authors employ Orlistat, which is specific for the thioesterase (TE) of human FASN, as well as 1,3-dibromopropanone (1,3-DBP) to specifically cross-link the acyl carrier protein (ACP) and ketosynthase (KS) domains. Using these cryo-EM datasets, the authors also develop a method to analyze the conformational landscape of FASN particles. Specifically, the authors report:

- (i) Full-length human endogenous FASN alone, at an overall resolution of 3.1 Å (3.1 Å modifying domain, 3.0 Å condensing domain)
- (ii) Full-length human endogenous FASN with 1 mM Orlistat at 3.3 Å (3.3 Å modifying, 3.0 Å condensing)
- (iii) Full-length human endogenous FASN with 3 mM 1,3-DBP at 2.7 Å (2.7 Å modifying, 2.5 Å condensing), with ACP bound to KS

By refining the modifying and condensing domains separately, combining particles from the three datasets, then performing further classification after symmetry expansion, the authors resolve various sub-states of the condensation cycle including states with ACP engaging with either KS or malonyl/acetyl transferase (MAT). Considering all identified substates together, the authors conclude the catalytic steps of the two monomers in FASN are not correlated.

To attempt correlating a specific sub-state with a specific catalytic conformation, the authors develop a procedure to analyze the distribution of FASN particles with different orientations between the modifying and condensing domains. From their conformational landscape analysis, the authors propose that the condensing cycle reactions occur in a relatively confined orientation between the modifying and condensing domain.

In summary, based on their findings, the authors draw the following key conclusions:

- 1) The enzymatic reactions catalyzed by the two FASN reaction chambers are not correlated;
- 2) FASN function does not require large rotational or swiveling motion between the modifying and condensing multidomains; and
- 3) Physiologically relevant dynamics of FASN are probably limited to the more restricted, so-called “wiggling and jiggling” motions.

Whereas each of these conclusions is supported by elegant structural observations, no data is presented to verify the catalytic relevance of these observations. Because no structurally characterized protein in this study represents an actual catalytically relevant state of FASN (they are either the apo form of the protein or irreversibly inhibited states attained by

mechanisms that do not directly relate to the chemistry of any of the constituent enzymes of FASN), the authors' inferences can at best be considered as plausible. While the challenge of definitively proving negative conclusions (e.g., absence of correlation of half-site activity, or that large-scale motions are not needed for synthesizing fatty acids) is not straightforward, the manuscript would be greatly strengthened if at their conclusions were backed up by independent but complementary lines of experimental evidence.

Other major concerns:

- 1) The authors' description of their ability to resolve the ACP domain is hyperbolic. In the one PDB file provided, only one of the ACP helices is visible.
- 2) Structural data for DBP-crosslinked FASN cannot establish which ACP domain is crosslinked to the KS domain of which subunit. The text reveals a bias toward intra-subunit crosslinking (e.g., Figure 1a), but this is by no means obvious from the actual structure. It is not even discussed explicitly. Classic work by Stoops, Henry and Wakil (JBC 259, 12482, 1983; not cited, although it should be) led to the conclusion that this was an inter-subunit mode of association, although that conclusion was questioned in ref. 28. Notwithstanding the high structural resolution claimed by the authors of their DBP crosslinked structure; their work falls short of answering this important unanswered question.
- 3) The use of the term "domain" to characterize the "condensation" and "modification" portions of FASN is confusing. Both are multidomains.
- 4) The authors must provide data verifying that the FASN protein preps used in their EM analysis have kinetic parameters expected of this multifunctional enzyme.
- 5) It is unclear how orlistat helps reduce heterogeneity, thereby allowing the authors to resolve the TE domains. Perhaps a comment about this could be helpful.

Minor points:

- 6) Line 119: Characterizing holo-ACP in the condensing cycle
- 7) Line 125-126: 4 or 5 functional groups?
- 8) Line 317: Give specific examples of other enzymes that function similarly when their activity is upregulated
- 9) Line 432: How long was FASN incubated with 1,3-DBP or Orlistat?
- 10) Line 77: swiveling and swinging
- 11) Line 151: fix
- 12) Line 164-166: rewrite sentence
- 13) Line 176: fix
- 14) Line 205: substrate or snapshot
- 15) Line 209: KS or MAT
- 16) Line 216: remove :
- 17) Line 227: incomplete sentence
- 18) Line 236: fix
- 19) Line 301-306: rewrite sentences
- 20) Line 309: in adult humans
- 21) Line 311: such as in cancer cells
- 22) Line 324: fix
- 23) Line 423: (Sigma-Aldrich, 125903)
- 24) Line 446: remove "respectively"
- 25) Line 463: particles
- 26) Line 468: Extended Data Figs. 2 and 4
- 27) Line 486: Extended Data Figs. 2-4
- 28) Line 489: domains
- 29) Line 511: Supplementary Table 1
- 30) Line 514: Sus scrofa
- 31) Line 873: remove track changes
- 32) Supplementary Fig. 1d: 1,3-DBP
- 33) Extended Data Fig. 2: "maps from Arctica" (?)
- 34) Extended Data Figs. 2-4:
 - a. Fix multiple instances of typo "interative"
 - b. Fix typo "Comebine with..."
 - c. Fix typo "Extraction of momdifying domain"
- 35) Extended Data Fig. 6: Fix typo "Comebine all particles ..."
- 36) Extended Data Fig. 10: Fix typo "Extended Daa"

Referee #3

(Remarks to the Author)

In the manuscript "Structural dynamics of human fatty acid synthase" Choi, Cheng and colleagues elucidate structures of human fatty acid synthase (hFAS) by cryo-EM. They used a strategy of tagging and purifying endogenous hFAS for single particle cryo-EM. They have succeeded in capturing structural snapshots in the condensing cycle of hFAS by visualizing ACP engagements with either KS or MAT domains. The authors have also acquired two datasets with an FDA-approved drug Orlistat, targeting the TE domain, and 1,3-dibromopropanone (1,3-DBP), which induces a cross-link with the phosphopantetheine arm and the KS catalytic cysteine. They have also developed a procedure to analyse the conformational landscape of particles with different orientations between the condensing and modifying domains. With this procedure of conformational landscape analysis they suggest that the formerly postulated hypothesis of a swivelling and swinging motion of hFAS which is thought to shuttle substrate to different enzymatic domains of hFAS by ACP, is not fully necessary. Instead, they suggest that only subtle wiggling and juggling motions are sufficient for ACP shuttling in hFAS. The

latter small scale motions appear more feasible in the crowded cellular milieu.

This study is a significant advance in the study and understanding of the structural basis of de novo lipogenesis, by hFAS. It also furthers our understanding how hFAS is targeted by inhibitors and drugs. In particular, it emphasises on how conformational landscapes influence the activity of this dynamics of this machine which is central to metabolism, but also in various malignancies. Knowledge of conformations adopted by hFAS will be essential to develop specific and allosteric drugs directed against hFAS.

While this study is interesting, its publication would require several points to be addressed by the authors:

- hFAS purification: this preparation is far from pure. I am referring to all the proteins visible in Extended Fig. 1B in the range of 75 kDa to 25 kDa. What are the other proteins co-purified with hFAS. Are these truncation products? Both issues would have substantial implications for the conclusions drawn by the authors.

- Orlistat and 1,3-DBP addition: Are these two compounds even bound? Is there any orthogonal validation to ensure their binding? While I understand that it might be difficult to address this in the case of Orlistat which is bound to the flexible and unresolved TE domain, the situation is different with 1,3-DBP. As the authors state themselves, 1,3-DBP should cross-link the catalytic Cys131 with the phosphopantetheine of the ACP. Considering the resolutions attained by the authors for the condensing domain this cross-link should be clearly visible. In the maps provided by the authors, I am having a difficult time finding this covalent attachment. Also this has substantial implications for the conclusions reached by the authors.

- The authors put forward a catalytic triad formed by C161, H293, and H331 in the KS. This is highly unusual as the two histidines would have similar pKa values. Do the authors have mutational evidence that supports this catalytic triad? In the model none of the histidine residues are in hydrogen bonding distance. In fact, by the same criteria applied by the authors Glu333 appears to be in suitable distance to support proton transfer through H331. The pKa values of this catalytic triad constellation would make much more sense!

- In all maps provided by the authors the ACP densities (including phosphopantetheine) are considerably weaker than the rest of FAS. Fortunately for the authors the phosphopantetheine residues allow them to establish topology of the pseudo-symmetric ACP. However, the weaker density of the ACP alludes to (conformational) heterogeneity. To which extent does this impact the statement of the authors that the ACP of endogenously purified hFAS contains holo-ACP? Do the MS/MS experiments indicated quantitative holo-ACP?

- Related to this issue: What are the specific contacts established in the recognition of ACP by hFAS-KS and hFAS-MAT? How do they compare to the interactions of ACP with bacterial KS and MAT? For that matter how do the interactions compare with the recently determined yFAS-ACP interactions?

- I have a bit of difficulty wrapping my head around the "floating ACP" position described by the authors? Surely there must be some specific contacts of the ACP to hFAS? Otherwise, it is difficult to imagine how classification would yield a singular position. One would expect a washing-out of density effect to ensue. Such contacts would have to be with the modifying domain which would be suppressed by the image processing scheme employed by the authors. This is in fact indicated in the conformational landscape analysis in Figure 5c. I encourage the authors to elaborate on this important point.

- Relatedly, can the topology of the ACP even be established in the floating ACP position? In which direction is the phosphopantetheine arm pointing? Does the position of the phosphopantetheine arm even support the authors' hypothesis that the floating ACP represents an intermediate in transition between KS and MAT?

- Regarding No-ACP and dynamic MAT groups: The authors speculate that these represent the dynamics in MAT are associated with capturing acetyl- or malonyl-CoA. While the authors correctly admit that ACP is not visualised, can they discern densities on the catalytic Ser581 which correspond to acetyl- or malonyl-esters?

- In line 166, the authors conclude that the absence of any coordination between ACP hopping and conformations adopted in two monomers suggests that the catalytic steps of the two monomers in hFAS are not correlated. I find this a bit of a stretch as considering the conformational dynamics within this structure, it is not intuitive to me how to derive such a statement from not visualizing certain states. How can the authors distinguish between classification/ alignment errors and functional significance? This is especially even more so, when none of the structural analysis performed by the authors involves the addition of substrates to initiate functional cycles. This statement asks for enzymological analysis which interrogates allosteric communication. The KS-MAT paper by Parthinkar, Grininger and colleagues where such enzymological assays were performed clearly demonstrate positive cooperativity in condensation steps, contradicting the authors' conclusion.

- The same applies to the entire paragraph starting from line 169: The authors use this paragraph to question the swiveling and swinging model. I personally find this problematic for two conceptual reasons: The pre-requisite for this statement by the authors is that they fail to reconstruct 3D structures where a single substate in the condensing domain would be associated with a specific mutual orientation of condensing or modifying domain. This does not occur in their analysis and therefore prompting the conclusion reached by the authors. However, they fail to address potential issues with sampling space. It is very conceivable that the numbers of particles sampled by the authors are not exhaustive enough for observing the low-populated elusive states. Secondly, they reach this conclusion in a resting state enzyme. It is very likely that an energetic barrier exists which prevents the enzyme from reaching cooperative behaviour in the absence of substrates. Therefore, to reach this conclusion the authors should at least analyse another dataset in the presence of substrates in turnover conditions. This should be followed up by conformational landscape analysis under these conditions.

- The conformational landscape analysis performed by the authors is an appropriate way to quantitate dynamics within hFAS. However, this analysis is not conceptually novel per se. Stark and colleagues have performed such analyses before for the bacterial ribosome, the 26S proteasome, the spliceosome, and the yeast FAS. Frank and colleagues have also performed such analyses for bacterial ribosome. I find it a bit strange that the authors do not cite these papers to put their own efforts into context with earlier developments in the field. This should also be addressed in the conclusion and discussion section.

- It is important that the inhibitors constrain the sampled conformational landscape. This is very much consistent with earlier findings by Stark and colleagues in the 26S proteasome that 20S proteasome inhibitors allosterically inhibit motion in the 19S regulatory particle, and the gamma-subunit the rotation of yeast FAS. Surely, this would warrant citations? More

importantly this alludes to an important principle in the function of macromolecular complexes which the authors fail to elaborate on.

- I lack some specific experiments on the conformational landscape analysis to elaborate on this: How does the conformational landscape look for hFAS in turnover conditions? How do the employed inhibitors Orlistat and 1,3-DBP modulate the conformational landscape in turnover conditions?

- The conformational landscape analysis in turnover conditions would also address the important question if the dynamics carefully investigated by the authors correspond to off-pathway states or on-pathway states? Whether there is indeed no coupling between monomers in hFAS? If swiveling really exists? How ACP movement is correlated with the conformation of both monomers?

- Spots of high particle densities in the conformational landscape should give rise to particles which are amenable to high-resolution reconstructions. Is this the case? 10000 particles if conformationally homogeneous should give rise to 3 Å structures. Should this not be the case, this would allude to a sampling problem where particle numbers are just not high enough. Alternatively, it would suggest that conformational landscape analysis has not reached convergence and some more additional hidden substates exist. Could the authors comment on this?

- Minor points: some aspects of the manuscript are not clearly worded and warrant some re-writing. This applies especially to the conceptual parts of the manuscript.

Version 1:

Reviewer comments:

Referee #1

(Remarks to the Author)

First and foremost, I would like to emphasize that this manuscript is highly interesting. It greatly improved during revisions. In principle, the ms merits publication in Nature. I am also grateful for the detailed responses to many of the points I have raised, and for sharing structural information as reviewer only material. However, I still have a few concerns that should be addressed before the ms is published:

(i) The work by Choi et al. refines the structural understanding of FAS by demonstrating, for the first time, that the inherently pronounced conformational variability is unlikely to be essential for fatty acid biosynthesis and is plausibly not fully utilized within the cellular environment. However, I would like to note that I continue to disagree with the authors' emphasis on stating that the "swiveling and swinging model" reflects the current perception of the FAS mechanism, and the new data by Choi et al. now revise this model. Although the large conformational variability has been demonstrated by several methods, there is no model that conformational variability is obligatory to run the FA cycle. (ii) Second, I appreciate the additional insight into ACP docking. However, several conclusions regarding ACP docking to the condensing wing, as drawn by the authors, remain unclear and should be clarified by providing additional information (without necessitating new data collection). (iii) The claim that monomers are unsynchronized or do not collaborate does not seem supported by cryo-EM data. A more careful discussion is necessary.

Major and minor points:

(1) line 55: it should probably mean „metazoans“ or „metazoa“ as a taxonomic group used in singular

(2) line 56/56/59...: wings are still named "domains"

(3) line 78 and line 179: „Consequently, continuous large rotational motion between the wings is vital for FASN function.“ Although it has been argued that swiveling can drag the ACP closer to KS to improve docking, a "swiveling and swinging model" does not reflect a broad current perception of the FAS function in the community. Neither the revised manuscript nor the rebuttal letter convincingly substantiate this model; e.g., literature with direct data to the "swiveling and swinging model" are not included (see statement line 179 without any citation „The "swiveling and swinging" model suggested that rotation between two wings is essential to position ACP with substrate to different enzymatic domains.“). The review cited in the rebuttal letter and manuscript PMID: 37856285 does not support this model but instead states (i) that large rotations can occur, which explains the knockout experiments by Joshi et al., and (ii) that there are two levels of conformational variability, namely local positional variability of the ACP through flexible linkage to the KR and TE, and overarching conformational variability of the wings. This does not imply the necessity of swiveling and swinging for fatty acid synthesis.

(4) line 134: The more detailed presentation and interpretation of data is in general appreciated. However, there are some open points. Two ACPs are docked to the KS in slightly different positions. (a) In MS analysis apo-ACP has been identified. How likely is it that in the one ACP position, the gray density, the ACP is without PTM and this position is not physiological, at least not reflecting one step of the proposed two-step model. Or there could also be the modification with malonyl such that the two states are not sequential but refer to transacylation and condensation. ACP:KS complexes have been characterized for bacterial FAS type II by Burkart and coworkers (see e.g. 10.1038/s41467-020-15455-x). Are positions similar and may the bacterial structure help in understanding ACP docking to human FAS KS? (b) I am grateful for providing maps. It seems that the helix in the condensation wing model does not very well fit the density. And it does not align with the full ACP docked. Probably a repositioning of the helix is necessary. (c) The authors interpret that ACP docks to MAT transiently. They state „density resembles the shape of ACP and loosely attaches to a well resolved catalytic cleft in the MAT domain“. While this is plausible, I suggest that also here the density should be presented with docked ACP, and the pdb file shared. Again, I am grateful that authors shared files. I agree that the position of the extra-density well aligns with ACP at the

active site (close to binding cleft). References 36 and 37 do not refer to fast MAT kinetics. Better its references to fast kinetics of MAT are: Rangan and Smith; 10.1074/jbc.271.49.31749; and Rittner et al.; 10.1021/acschembio.7b00718). (d) In the rebuttal letter, the authors argue that for ACP in transition „ACP density is not floating away from the condensing wing, but with some attachment with some part of the condensing wing.“ This is not mentioned in the manuscript. Structural information provided as reviewer only material dock ACP in distance to about 15 Å to the condensing wing, which seems too far away to have direct contact. A more careful discussion on ACP in transition seems appropriate.

(5) line 178: To the unsynchronized monomers (also referring to Rebuttal Letter: Comment to Reviewer#3 line 166): Choi et al claim that there is no synchronization between the monomer (in the original version of the manuscript the term „no correlation“ was used). Reviewer#3 addresses an important point to the heterogenous loading of ACP which may hamper synchronization. Of note, there is a new paper by the Grininger lab further substantiating cooperativity in the KS mediated elongation. Gusenda et al. suggest that cooperativity involves ACP binding such that the binding of ACP to one KS protomer favors the binding to the other one <https://doi.org/10.1002/anie.202412195>. Cooperativity means that there is synchronization; however, synchronization does not mean that it is absolute. Thus, cooperativity in the KS-mediated condensation reaction can be manifested in an increased prevalence of ACPs docking to both KSs as it would happen without cooperativity. Data by Choi et al. cannot rule out such synchronization at the level of cooperativity as observed by Gusenda et al. Thus, it is recommended that authors are more cautious in their statements regarding “no collaboration” or “no synchronization” of monomers.

(6) line 304: The authors conclude that substrate-shuttling by human FAS follows all concepts - free diffusion, medium conformational constraint and excessive conformational constraint. This does not align with Kira Weissman’s original concept that a megasynthase can follow in principle just one of these concepts: The free diffusion model means that there is no steering effect by the catalytic core (the fungal FAS follows rather this concept, although Stark and coworkers revised the view slightly). The medium conformational constraint model means that there is higher order (large scale) conformational variability of the catalytic core that supports or hinders the ACP from reaching a subset of catalytic domains. The excessive conformational constraint model means that ACP is guided to the catalytic domains by the higher order conformational variability. The free diffusion of ACP, as the local conformational variability, is implicit in medium conformational constraint and excessive conformational constraint. Given data presented in paragraph starting with line 220, substrate shuttling in human FAS seems to follow the medium conformational constraint model.

(7) line 334: The authors state: „Likely, the open reaction chamber in PKS also allows enzymatic reactions across different molecules to increase overall enzyme productivity.“ This statement should be toned down. It is a hypothesis, without direct data.

In sum, I strongly support publication of the manuscript. I suggest that files of ACP docking are shared with the public.

Referee #2

(Remarks to the Author)

None

Referee #3

(Remarks to the Author)

The revised manuscript "Structural dynamics of human fatty acid synthase" by Yifan Cheng and colleagues, addresses many aspects raised by the other reviewers and myself. More specifically, overall the revised manuscript is toned down in the most contentious statements and offers a series of control experiments essential to validate the statements made by the authors. I very much appreciated and commend the efforts made by the authors to resolve many of the raised issues. These include especially the more advanced characterization of the biochemical preparations of FASN used for structural analysis, the enzymatic assays and clarification of the methods employed for conformational landscape analysis. The PEG6000 experiment indeed validates the utility of the conformational landscape analyses as performed by the authors in this manuscript. I appreciate the balanced discussion throughout the revised manuscript and inclusion of the citations I had suggested in the original review. This has certainly contributed to the legibility and quality of the revised manuscript.

Nevertheless, I do not feel that the main criticism which had voiced in the original review has been satisfactorily addressed. I encourage the authors to address these issues below:

The authors have not captured any substates in the modifying wing of the FASN. Therefore their statements with regards to conformational dynamics to the overall catalytic reactions in FASN are unjustified and speculative. The only substates addressed by the authors are those within the condensing wing. Therefore, I would appreciate if the authors restricted their statements within the entire manuscript solely to the condensing domain.

- Change the title to: "Structural dynamics of human fatty acid synthase in the condensing cycle"

- Abstract: We captured conformational snapshots of various functional substates in the CONDENSING CYCLE... Together, we reveal that FASN function does not require large rotational motion between its two major functional domains in the CONDENSING CYCLE, and that the CONDENSING CYCLE catalytic reactions... Our data thus provide a new composite view of FASN dynamics during the fatty acid synthesis CONDENSING CYCLE.

The same applies to all statements within the entire revised manuscript!

Version 2:

Reviewer comments:

Referee #1

(Remarks to the Author)

My concerns have been addressed. The manuscript is now ready for publication.

Referee #3

(Remarks to the Author)

I congratulate the authors for a nice manuscript that has been revised in a substantial manner from initial submission. I thank the authors for acknowledging the comments of the Reviewer #1 and my own. Based on my reading they have also fully acknowledged these by editing the manuscript and modifying the title. The only caution I would like to point out that in several instances within the manuscript the authors claim that large conformational dynamics are unnecessary for the fatty acids synthase cycle. I again strongly advise the authors to restrict their statements to the condensing cycle only, as they have captured no states and no observations of the modifying cycle. Please ensure that these necessary modifications are followed through to avoid misunderstandings.

We thank all reviewers for their constructive comments. Following their suggestions, we have performed additional experiments, which we summarized here:

1. We performed mass spectrometry experiments to detect phosphopantetheine (Ppant) modification of purified endogenous FASN. We found that our FASN samples contain both modified and unmodified S2156, i.e. apo-ACP and holo-ACP, and other post-translational modifications, most of which were previously reported. Our result that only some of endogenous ACP is modified with Ppant is consistent with early studies (PMID: 7567999 and 26851298). We can now conclude that our purified endogenous FASN is functional and physiologically relevant.
2. We established a robust pipeline to measure enzymatic activity of FASN. We measure NADPH consumption by fluorescence in the presence of all substrates, NADPH, acetyl-CoA and malonyl-CoA, followed by detecting intermediate enzymatic products linked to Ppant tip by mass spectrometry. Together, we confirm that FASN consumes NADPH and ACP shuttles enzymatic intermediate products between different enzymatic domains. This experiment ensures that our purified endogenous FASN is functional.
3. We studied FASN in the buffer that contains 30mg/ml PEG6000. Conformational landscape analysis reveals a noticeable reduction in rotational motion (along α axis) between the two wings in the presence/absence of all substrates. Functional study by the assay mentioned above show that FASN is fully functional in PEG6000 buffer. Together, this new data supports our conclusion that large rotations is not required for FASN function.

In the revision, we include these new data and revised text to address all comments of reviewers. New or extensively revised text are in blue color. In the following, we provide a detailed point-to-point response.

Referees' comments:

Referee #1 (Remarks to the Author):

The manuscript by Choi et al. reports the conformational variability of the human FASN using single particle cryogenic electron microscopy (cryo-EM). FASN has been purified endogenously from HEK293, and treated with Orlistat, a drug against the TE domain of human FASN, and with 1,3-dibromopropanone (DBP), a crosslinker used for FAS analysis before. Three datasets were received from preparations: (i) no treatment - termed FASN alone, (ii) Orlistat and (iii) and 1,3-dibromopropanone (DBP). The authors focused on the characterization of the conformational dynamics of human FASN using, which is relevant for understanding acyl carrier protein (ACP) mediated substrates shuttling. Their structure scan visualize ACP docking to the catalytic KS or MAT domains, representing key functional substates during the carbon-carbon bond forming part of the fatty acid cycle.

A crucial aspect of this manuscript is the conformational analysis by assigning Euler angles to the two wings, treated as rigid bodies enabling to assign conformational

landscapes of particles that differ in the orientations between the condensing and modifying wings. One of the main findings is that only small conformational variations are necessary for conducting fatty acid synthesis, contradicting - as the authors argue - the “swiveling and swinging” model. Instead, FASN can be fully functional with subtle “wiggling and jiggling” motions. The authors argue that the type of restricting conformational variability, the “wiggling and jiggling” model, is reflecting the real conformational variability of the protein in the crowded cellular environment. In a further crucial aspect of the manuscript, the authors captured FASN in specific functional states during the condensation cycle and structurally described a plethora of FASN substates (in which one reaction chamber is engaged in MAT or KS catalysis). They found that the enzymatic reactions carried out by the two monomers do not correlate.

We appreciate this summary of our work, which largely captures key points of our study.

The manuscript is very technical, with the functionally relevant statements being derived from the analysis of the conformational landscape and substate analysis. The paper is not easy to read for molecular biologist. Rather, in its current form, it seems specialized for the cryo-EM community.

We appreciate this comment. In the revised manuscript, we moved most of the original Fig. 3 into Extended Data Figure 9 and combined the key description panel with the original Fig. 4. Similarly, we revised some text extensively to smooth the transition between different sections, making it easier for readers to grasp the key contents of our studies. We believe our revised manuscript is now more readable for readers outside of cryo-EM community.

Additionally, the analysis does not provide much new insight in the function of mammalian FAS. While it is undoubtedly interesting to map the conformational variability of mammalian FAS, on the example of FASN, which has not done in this detail before, it does not offer a new understanding of the protein’s molecular mechanism(s).

With respect, we disagree with the assessments that our studies do not offer new understanding of FASN’s molecular mechanism. We argue that we provide following novel mechanistic insights to understand FASN function:

- By capturing various substates of condensing cycle, we reveal that ACP engagement with enzymatic domains are evolutionally conserved as electrostatic interactions and reveal unambiguously that the catalytic reactions carried out by two monomers are unsynchronized.
- We provide structural and functional data that large rotation between two wings of FASN is NOT required for acyl chain synthesis.
- By combining captured substates with analysis of conformational landscapes, we address the question of how the orientation between two wings of FASN affect its

ACP shuttling. Our analyses clarify three different models of ACP engagement within a reaction chamber that were previously proposed (PMID: 26284673).

- Our studies further reveal that the “open” reaction chamber of individual FASN is dynamic and variable. It is functional when confined to small central region of the landscape, within which the angular orientations between two FASN wings are small enough to allow ACP to freely diffuse from one enzymatic domain to another.
- Furthermore, by comparison with the enclosed reaction chamber of yeast FAS, new insights provided from our studies raise new question and hypothesis that would require further study to test. Without ACP engagement with the condensing wing, FASN molecules can have very large angles between their two wings, making their reaction chambers widely open, within which ACP is incapable of reaching all enzymatic domains to complete acyl chain synthesis. Then, what is the biological implication of having FASN molecules with such wide-open reaction chambers being idle in cells? We hypothesize that, when multiple FASN molecules are close enough, ACP can shuttle substrates across molecules, making idle FASN with wide-open reaction chamber functional again. While this is a speculative hypothesis, the scenario of having high concentration of FASN molecules in specific sites to increase overall fatty acid synthesis have been seen in cells, such as during Dengue and classic swine fever viral infection (PMID: 20855599, 34132567).

Different than the authors' claim, there is no broader consensus that large conformational variability is essential for mammalian FAS function a for mammalian FAS („swiveling and swinging model“).

The rotation model, which may or may not be termed as “swiveling and swinging” model widely to the community studying FAS, has been a major model in describing function of FASN. This was suggested when the first metazoan FASN structure was determined (PMID: 16513975, 18772430). This model was further enhanced by a series of other studies, including early single particle cryo-EM (PMID: 19151726) and real-time high-speed AFM (PMID: 29023094). These previous observations were summarized in a more recent review (PMID: 37856285), which stated that “*conformational dynamics of the animal FAS fold comprises two main components: higher-order dynamics determined by the repositioning of the condensing and modifying wing (which is the swiveling and swinging motion) and locally high variability of the ACP and TE domains*”. The notion that large rotational motion between condensing and modifying wings is essential for substrate shuttling and FASN function have triggered many studies, including improving the methodology to characterize such motion (PMID: 30291265).

Thus, we argue that providing a detailed analysis of dynamic behavior between two wings of FASN and linking such motions to its function (data added in this revision) provide new understanding of FASN's molecular mechanism.

In this regard, the view on the evolutionarily related iterative PKS is insightful: Iterative PKSs and mammalian FAS share the same X-shaped fold and perform the same type of chemistry. Iterative PKSs have recently been structurally characterized. Intriguingly, for the iterative PKS CalA3 <https://doi.org/10.1038/s41467-023-36989-w> and LovB <https://doi.org/10.1038/s41467-021-21174-8> conformational variability of the two wings could not be observed in cryo-EM (no swiveling and swinging).

Indeed, some PKS proteins do not undergo large rotational motion between their two wings, such as LovB, which has a stable hinge between the two wings. However, there are also many PKS proteins that do have large rotational motion. A very recent structural studies of PKS13, mycobacterial polyketide synthase, revealed significant rotational motion between its condensing domain and HINGE/C-terminal domain (PMID: 36782050). In the study of PKS CalA3 (PMID: 36899013, cited by the reviewer), authors also pointed out that *“For fully functional PKSs with ACP that adopts the same architecture, the catalytic cycle may be hindered by steric constraints and mostly the lack of energetically coupled conformational dynamics. Directly opposite, the structural dynamics of modular PKSs are firmly established, featured by the end-to-end flip of ψ KR/KR in PikAIII, the rotational ψ KR/KR in Lsd14 and DEBS module 1, and the turnstile mechanism of AT in DEBS module 1. They both illustrate the structural rearrangements during substrate shuttling by the ACP domains”*. A more recent study released in bioRxiv (<https://doi.org/10.1101/2024.07.22.604177>) also observed significant rotations between two wings of animal fatty acid synthase-like polyketide synthases (AFPKs).

Thus, we would argue that, in many of these previous studies, large rotational motions between the two wings are observed and are thought to be part of the function. Our study, with the newly added data listed above, provide evidence to clarify this point.

Structural elements, in the waist region between the wings, were accounted to cause rigidity. This implies that the X-shaped fold, seen in CalA3 and LovB, mammalian FAS and other related proteins can perform synthesis without the need for large conformational variability.

We agree with the Reviewer that some PKS family members have more rigid waist region between the wings, such as LovB, and that they can perform synthesis without the need for large conformational variability. Indeed, our studies provided experimental evidence that this is also true for FASN, it can be fully functional in acyl chain synthesis without the need of large rotation between their two wings. This is one of the major conclusions of our study.

However, the structural element between the two wings of FASN, as well as many other PKS proteins, is not rigid and allows large rotational motion between the two. We, as well as previous studies, do observe that FASN molecules having very large rotational angles between the two wings. A key question is then: what is the functionality of having large angle between two wings in FASN? Our studies provide a reasonable speculation to connect this observation with increased fatty acid synthesis during viral infection.

The finding that the reaction chambers essentially work independently, matches the current perception of mammalian FAS mediated FA-synthesis. The two polypeptides of a dimeric mammalian FAS just share little interfaces (mainly via KS) such that is difficult to imagine how coordination of the reaction chamber could arise from this.

If the swiveling and swing motion is part of the FAS function, then each enzymatic reaction step in the condensing cycle would require a specific orientation between the two wings. In such a scenario, one would expect some kind of coordination, synchronization or asynchronization between the two reaction chambers.

Our findings provide definitive structural evidence that the two reaction chambers are unsynchronized.

A higher-order conformational variability of the X-shaped fold, like swiveling and swinging by FASN, increases the likelihood of interactions with the domains within this confined space. The idea that higher order conformational variability can improve efficiency of such FASs and PKSs was termed „medium conformational constraint model“ (see Weissman <https://doi.org/10.1038/nchembio.1883> and Buychuihan et al. <https://doi.org/10.1002/anie.202312476>).

We agree with this comment and our findings do not contradict with it but extend it further.

What we concluded is that large motion is not required for catalytic cycle. But rotation between the two wings within the central region in the landscape could be beneficial to shuttle ACP among different enzymic domains. We show that landscape of individual substates overlap significantly but also have somewhat different localized hot region, suggesting that a specific substate may favor certain orientation between the two wings. Thus, rotation between the two wings may help shuttling of ACP from one enzymatic domain to another and improve FASN efficiency, supporting “medium conformational constrain model”.

Furthermore, our studies also reveal two other aspects of ACP shuttling in connection with the angle between two wings. Within the central region, ACP can freely diffuse within the open reaction chamber at almost any angles between the two wings. This supports one of the models proposed by Weissman, free diffusion model. Outside of the central region, reaction chamber becomes widely open so that ACP is no longer capable of engaging condensing wing, thus in consistent with the other model of Weissman, i.e. excessive conformational constrain model.

We do appreciate this comment. We now revised the paragraph in the conclusion and discussion to articulate our thoughts.

While there is the impression, as detailed above, that biological relevance of the conformational landscape analysis is overestimated, other aspects remain almost uncommented. Particularly, data contains structural information to the interaction

between ACP and KS, and ACP and MAT. These interactions are highly interesting, as comprising the key steps in carbon-carbon bond formation.

While we disagree with the comment of overestimation about our landscape analysis, with respect, we do appreciate the points that our data contains other aspects that worth comment. We have now performed further experiments as outlined above, confirming that the purified endogenous FASN is fully functional as holo-enzyme, giving us confidence to interpret the structural information mentioned by the reviewer. We now revised the manuscript, both text and figures, to describe in more details about these interactions.

For example, (i) the MAT domain is highly interesting, because it has been characterized as extra-ordinarily fast compared to other transferases in this PKSs. How does the interface of ACP with MAT look like, is it minimal to feature such high turnover rates.

Regarding MAT domain, we have identified three main substates. In the ACP-MAT, the catalytic cavity is clearly resolved with charged residues in the surface of the MAT domain. The ACP is loosely attached to MAT. We revised the text to describe this in more details.

Without ACP, conformation of MAT is classified to two groups. In the No-ACP group, MAT is well resolved, but no density of substrate within the catalytic cleft. In the dynamic MAT group, density of half of the MAT, especially the catalytic cleft, becomes weaker or disappearing.

We can only speculate that different conformations, such as loosely engaged ACP and/or highly flexible MAT could be caused by high turnover rate. However, it is difficult to validate. Nonetheless, we appreciate this suggestion and revised text accordingly.

(ii) Is there any explanation for the two different ACP docking site at the KS domain?

By classification, we resolved 2 classes of ACP-KS, which could be interpreted as two substates of ACP engagement with KS. We now revise the figure to make these two different conformations clearer (revised Fig. 2d and Extended Data Fig. 8a and b). In the revised Fig. 2d and Extended Data Fig 8b, gray density shows a clear helical density of ACP interacts with H68 and K70 on KS surface, while phosphopantetheine (Ppant) density is weaker. The salmon-colored map shows a well resolved Ppant density connecting to the catalytic cysteine 161. Together, they could be interpreted as two-steps engagement of ACP with KS, where the first step is to dock ACP to surface of KS catalytic cavity via electrostatic interaction (gray density), followed by engaging Ppant with the catalytic site of KS (salmon density). We now revised the text to elaborate this.

(iii) How does the ACP in the floating state look like, and what does such a state mean for FASN function. It is noted that for modular PKSs these interactions have recently been characterized in structure, such that FAS and PKS could compared in this respect.

The class with an ACP density not attached to KS or MAT were identified through classification. The term “floating” could be confusing, as commented by the Reviewer 3. However, our data does not support to either the hopping or crawling model of how does ACP shuttle between MAT and KS.

In sum, this manuscript is very technical in its current form. However, data itself holds very intriguing information. It is recommended to streamline the conformational landscape analysis as well as better explore the yet under-examined data on the domain-domain interaction.

We appreciate these very constructive comments from the reviewer and revise our manuscript extensively.

list of comments:

-terminology: „domains“ should not be used for condensing and modifying substructures, it describes the enzymatic domain. Usually „wing“ or „region“ is used. Throughout this review, the established terminology is used for describing the regions of FAS.

Thanks for pointing this out, and we revised the text and use “wing” throughout the manuscript.

-line 46: I recommend saying „two key processes“; „step“ is mostly used for a single reactions

-line 53: „in bacteria and plants, enzyme propagating a fatty acid synthesis are...“

-line 55: not just mammals, but „animals“ or „metazoa“, please note that mammalian FAS should better be „animal FAS“ or „metazoan FAS“

Revised accordingly.

-line 60: the KS condenses KS-bound ace(lyl) and ACP-bound malonyl, but not their CoA esters

Revised.

-line 66: „The TE domain cleaves off the final product from the ACP when it reaches the desired length of 16 to 18 acyl chains“ is confusing. Better is „The TE domain cleaves off acyl chains from the ACP when they reach the desired length of 16 to 18 carbons“.

Revised.

-paragraph starting line 110: „...refined the condensing and modifying domains separately from the same subsets of particles...“. Does that mean the refinement was done separately, but the integrity of the particle was maintained in the sense that the actual relative orientation of the wings is preserved? Please clarify this point

Particles of entire FASN were picked. After excluding the bad particles by classifications, final particle set was subjected to independent refinement of each wing, without classification. During this process, each wing in every particle was refined separately against its own reference model, after computationally subtracting the other wing. Therefore, each intact particle has two Euler angle assignments, one for each wing. From these two sets of Euler angle assignments, we calculate the relative orientation between the two wings. The landscape is generated by plotting this relative orientation between the two wings of all particles in the final set.

-line 71: “dimer interface” can be confused with the interface between two protomers. I suggest to structure the sentence like this: “...and that the two dimeric domains (better wings) are arranged in parallel with a short linker between them.”

Revised.

-line 120: acetyl is just shuttled during priming; thus, in a condensation cycle not just acetyl but also acyl is loaded into the KS active site, and then condensed with enolate originating from malonyl.

Revised accordingly.

-line 125: it says 5 functional groups, but just four are listed in the following; according to Ext Fig. 6, it seems that class „dynamic MAT“ has been omitted. This is misleading and should be adjusted.

Revised.

line 129: ACP binding is mediated by electrostatic interactions, which has been analyzed in yeast FAS by Gipson et al. <https://doi.org/10.1073/pnas.0913547107> and Anselmi et al. <https://doi.org/10.1021/ja103354w>; before analyzed at higher resolution in ref 29.

Revised, and appropriate citations are added.

-paragraph starting line 130: How can the two distinct positions of ACP binding to the KS be explained. The authors should analyze whether one of these positions is induced by the 1,3-DBP crosslink.

We appreciate this suggestion. We now analyzed our data as suggested by the reviewer. We separate all classes to three datasets. As shown in the figure attached here, particle distributions of all classes in three datasets are rather similar. Thus, the two distinct positions of ACP binding to KS is not caused by 1,3-DBP crosslink. We explained this in the response to a comment above and revised the text accordingly.

This comment also raises the question as what is the influence of small molecules? In our analysis, we did not observe any significant change in ACP engagement (shown in this attached figure, and now included in the Supplementary Fig. 4a and overall rotational motion between two wings (Supplementary Fig. 4)

-paragraph starting line 130: can the interface between domains be solved at near atomic detail?

We resolved atomic resolution of KS domain, indicating H68 and K70 are key residues to provide electrostatic surface to ACP. However, ACP density, except for Ppant and few residues, is not good enough to provide atomic level of interpretation.

-again paragraph starting line 130: The argument that tunnels of different lengths a likely to accommodate partially synthesized acyl chains of different lengths during the condensing cycle, disagrees with the current understanding of chain length regulation. Structural data with acyl chains of different chain length have so far not identified binding to different cavities, see e.g. ref 16 with C8 and C12 bound to KAS 1 of E. coli. Note that a KS with bound octanoic acid has been solved structurally by Rittner et al. <https://doi.org/10.1002/pro.3797>. A superposition with the C8:KS data would help in visualizing the binding tunnel (see Extended Data Fig. 6d). Is the mixture of charged and hydrophobic residues supposed to support any binding of partially synthesized acyl chains. These points should be explained in more detailed and connected to the current understanding of substrate binding to KS. Acyl chain binding to KS, including chain length regulation, has been reviewed by Heil et al. <https://doi.org/10.1002/cbic.201800809>.

We appreciate this comment. Indeed, the explanation was very speculative without additional experimental data to definitive support it. In the revised manuscript, we removed this speculation.

-paragraph starting line 140: how should a floating ACP group be trapped in a defined position and orientation such that it can be traced in density?

Reviewer 3 also raised the similar question about the meaning of “floating ACP” group. Indeed, the term “floating ACP” is misleading. In class with “floating ACP”, what we observed is an unaccounted density with the size similar to ACP located in between KS and MAT domain and we interpret it as a transitional state between ACP engagement with KS and MAT. This density was observed in classes identified from classification of the symmetry expanded condensing wing, in which it appears not engaging with either KS nor MAT (Fig. 2h and class 9 of Extended Data Fig. 7a and b). We thus termed it “floating”, meaning that it is in transition between KS and MAT. In addition, we also observed similar density in reconstructions of the entire FASN with defined orientations between the two wings (Fig. 4c, Extended Data Fig. 10 and Supplementary Fig. 5 and 6, which was analyzed by using cryoDRGN). In these reconstructions, this ACP density is not floating away from the condensing wing, but with some attachment with some part of

the condensing wing. And we cannot assign it to any specific functional state. Thus, in the revised manuscript, we change “floating ACP” to “ACP in transition”.

-paragraph starting line 140: can the interface between domains be solved at near atomic detail?

The MAT domain is resolved to near atomic resolution for building accurate atomic model. However, ACP density is not good enough to provide atomic level of interpretation. We make this clear in the revised manuscript.

-paragraph starting line 145: Is there any biological interpretation for the dynamic MAT? Why it should be associated with acetyl- or malonyl-CoA?

In previous studies (PMID: 31811668 and 20132826), it was observed that the conformation of catalytic cleft of MAT domain is very heterogeneous in the presence of substrates, indicating highly dynamic nature of this catalytic cleft with substrates. As pointed out by the reviewer, MAT has a high turnover rate. We thus speculated that the dynamic MAT is associated with capturing acetyl- or malonyl-CoA. We do agree that this interpretation is speculative. The higher dynamic of MAT domain maybe a consequence of the high turnover rate in general. We now removed it in the revised manuscript.

-paragraph starting with line 169: There is no „swiveling and swinging model“ that suggests that rotation of the two wings is essential for activity (see comment above).

With respect, we disagree with this point. As mentioned above, a more recent review (ref 22, PMID: 37856285) states that “*Hence, conformational dynamics of the animal FAS fold comprises two main components: higher-order dynamics determined by the repositioning of the condensing and modifying wing (which is the swiveling and swinging motion) and locally high variability of the ACP and TE domains, Knowledge about the dynamics is essential to understand substrate shuttling in animal FASs*”. Our study clarified that the swiveling and swing motion does exist, but it is more stochastic than following defined path and is not essential for the shuttling substrate by ACP across different enzymatic domains.

-line 163: how is the assembly of condensing and modifying wing done in terms of their relative orientation?

Converting the symmetry expanded domain back to the entire complex is a standard procedure in symmetry expansion operation. It is basically a sorting procedure. For example, for ACP-KS/ACP-MAT group, we sort out and group any particle with one domain being classified as ACP-KS and the other classified as ACP-MAT. The whole condensing wing is then refined with ACP-KS in one side of the wing and the ACP-MAT in the other side. The same procedure is repeated for the modifying domain, followed by combining the two domains together. Note that, in the last step of combining condensing and modifying domains together, we can only assign left/right for one domain but not both.

-line 181: what does „same subset of particles“ means?

The reconstruction of condensing and modifying domains are calculated from the same particle set. Such set could be the particle set before symmetry expanded classification (ref to the whole particle set), or after symmetry expanded classifications (outlined above). We now revised the text to make this description clear. The procedure is also explained in response to a comment above.

-line 183: Calculation of the Euler angles, assigned to each wing, is difficult to grasp in its details.

We explained the procedure of calculating the orientation landscape in the Method. Briefly, for each particle, with the two sets of Euler angles assigned, one to condensing and one to modifying wings during refinement, the orientation between the two wings can be calculated by formula listed in Extended Data Fig. 9a.

Which particles are used for determining the relative orientation of wings. See also my comment to line 110.

Each and every particle in the dataset (combined data of all substates, or of each individual substates) are used to calculate the relative orientation. Plotting the orientation of all particles in the dataset in a Cartesian coordinate system generate the landscape.

-line 196: The authors interpret landscape of conformations as constraint by ACP engagement. However, this would just be possible if the ACP, as part of the modifying wing, binds to the catalytic domains of the condensing wing. As just a subset of particles show ACP binding to KS and MAT, this interpretation is incomprehensible.

The landscape can be calculated from any specific subsets of particles, as far as each wing of every particle in the subset can be refined and Euler angle assigned. Shown in Fig. 3f of the revised manuscript, we compare the landscapes of ACP-KS/ACP-KS, ACP-KS/No-ACP and No-ACP/No-ACP substates and show that the constraints of landscape increase with the engagement of ACP to KS. Thus, ACP engagement clearly constrain landscape. Similar comparisons are also shown in Extended Data Fig. 10.

-line 209: The authors should disclose the calculation leading to 76 Å? For the argument of constraining conformation landscape, this distance is treated as a threshold, such that details to its calculation are important.

This is now added to Fig. 3c.

-line 210: The authors should also explain that A2115 is the attachment point for ACP.

Corrected.

-line 268: Joshi et al did not suggest, that ACP can only engage with one monomer. They proved, that the mFAS is active within one monomer, but they also clearly state, that ACP can interact inter- and intra-subunit.

We thank the Reviewer for pointing out that we mis-state the previous studies. We have now revised the whole paragraph.

-line 309: Unclear meaning: With less fatty acids in the nutrients, FASN would be less constrained in conformational variability? Or: The low efficiency of mammalian FAS is the reason that fatty acids are supplied by nutrition rather than FASN derived?

We now revised the conclusion and discussion section.

-line 311-314: Further unclear point: Higher copy numbers of FASN will of course not change crowded cytosolic environment.

We now revised this paragraph to clearly articulate our thoughts.

-line 315: very speculative and not supported by any data: given the references, FASN may form filaments?

We did not speculate FASN may form filament. During viral infection, FASN expression is upregulated, viral proteins redistribute FASN to viral replication site to increase local concentration of FASN and fatty acid synthesis (PMID: 20855599, 34132567). We speculate that such congregation of FASN could facilitate ACP shuttling across different FASN molecules with large angle between the two wings. We revised the original paragraph in the revised manuscript to better articulate our thoughts.

Additional points:

-Has the tip of the PNS been analyzed for any density of an acyl chain bound to the PNS? Endogenously purified FAS should probably have intermediates of acyl chains bound. The location probability of the ACP in the complex might be affected by the substrate, that is bound to the PNS. This possibility should be mentioned at some point.

We now performed mass spectrometry analysis of multiple batches of purified endogenous FASN. We do observe intermediates of acyl chains of different sizes. In the revised manuscript, we now include new mass spectrometry data. In addition, we also performed mass spectrometry analysis of purified endogenous FASN incubated with all substrates, NADPH, acetyl-CoA and malonyl-CoA, from which we also detect acyl chain of different lengths in the tip of Ppant after incubating purified endogenous FASN with. We now include these new data in the revised manuscript (Extended Data Fig. 1f and g, and 2).

- Enzymatic activity is an ideal parameter for checking protein quality. Have protein preparations been tested in activity?

We appreciate this point. We have now performed assay to demonstrate that the purified endogenous purified FASN is functional, as it consumes NADPH and produces different size of enzymic intermediates.

We established a new robust enzymatic activity assay, in which we use florescence instead of UV absorption to measure NADPH consumption after incubating purified endogenous FASN with all necessary substrates, NADPH, acetyl-CoA and malonyl-CoA. After NADPH consumption measurement, we subject the sample to mass spectrometry to detect the intermediates of acyl chain of different lengths. In this way, we are certain that NADPH was consumed by the entire enzymatic cycle of FASN to elongate acyl chain. In addition, we performed this assay on purified FASN in PEG6000 buffer that reduces rotational range of FASN. New data are now included in the revised manuscript (Fig. 1d, 5c, and Extended Data Fig. 2).

-The claim that large rotational motion between the condensing and modifying wing is unsupported by functional data. it would have been interesting to see enzymatic data on human FASN restricted in conformational variability. A possibility for engineering may be the exchange of the flexibly linkers with alpha helical element as found in LovB of MSA-line PKS.

We appreciate this comment. A major revision of the manuscript is reflected in addressing this question.

Since we are working with endogenous proteins, it is not easy to engineer the protein. For recombinant full length FASN, it is difficult to ensure physiologically relevant Ppant modification in ACP. Instead, we used PEG6000 to reduce the rotational motion of FASN. We confirmed that the rotational range of FASN is indeed reduced in PEG6000 buffer, yet the protein is fully functional in consuming NADPH and producing enzymatic intermediates (Fig. 5 and Extended Data Fig. 2e). Furthermore, our preliminary analysis of FASN in the presence of additional substrates in PEG6000 buffer, under the turnover condition, shows that similar dense central region. See attached figure here.

Thus, we can conclude that reducing the probability of large rotation between the two wings does not cause significant change in NADPH consumption. Detailed analysis of substates and landscapes under this turnover condition would require much

larger dataset and more extensive processing, exceeding the timeline for resubmitting this revised manuscript. We thus only include the new results of FASN landscape under PEG6000 buffer without added substrates in the revised manuscript. It supports our conclusion that reduced rotation motion of FASN by PEG6000 does not impact FASN function (Fig. 5).

-Has the strep tag been used for purification? Or for anything else?

No, we did not use the Strep tag for purification. Our collaborator has used it for other studies that is beyond the scope of this work.

-The authors should please comment on the absence of structures that show the ACP at the modifying domains.

In this study, we only focused on capturing substates with ACP engaged with condensing wing. Capturing ACP engaging with specific enzymatic domain in the modifying wing is more technically complicated and requires substantial efforts. We will follow up with this in our follow up studies.

-Fig. 2e suggests that no protein-protein interactions of ACP and KS were found. The authors are asked to comment on that.

In Fig. 2e, to show clearly the engagement of Ppant with the catalytic triad, we only show the masked density of the ACP helix from which Ppant attached. The rest of ACP density are present (Fig. 2d).

-line 125: typo: „APC“ instead of “ACP”

Corrected.

-Fig. 1b: what does C16 (yes) and C16(no) mean?

We revised the figure.

-Figure 4d: Numbers on the axis have shifted.

-Fig. 4f: The purpose of the dashed lines (barely visible) is unclear.

-Fig. 5a: It would be appreciated if the substate groups P1-P4 are marked with a dot or circle in the landscape.

-Ext. Data Fig. 1c: correct CO₂

-Ext Data Fig. 10; typo „Daa“

-these references seem misplaced:

(ref 15): cited in the context of type II FAS: Paper is about a cryo-EM structure of Pks13, which is a type I PKS.

(ref 16): cited in the context of catalytic functions in the modifying region: Fantastic paper about the mechanism of the KS, analyzed with different acyl chains and type II KS from E. coli.

All corrected.

Referee #2 (Remarks to the Author):

Understanding the dynamics of the multifunctional vertebrate fatty acid synthase (FASN) and its biochemical (and perhaps even biological) implications is an important scientific challenge. In their manuscript, Choi and Li et al. address this challenge using single-particle cryogenic electron microscopy (cryo-EM) to resolve various conformational states of intact human FASN. To reduce heterogeneity, the authors employ Orlistat, which is specific for the thioesterase (TE) of human FASN, as well as 1,3-dibromopropanone (1,3-DBP) to specifically cross-link the acyl carrier protein (ACP) and ketosynthase (KS) domains. Using these cryo-EM datasets, the authors also develop a method to analyze the conformational landscape of FASN particles.

Specifically, the authors report:

- (i) Full-length human endogenous FASN alone, at an overall resolution of 3.1 Å (3.1 Å modifying domain, 3.0 Å condensing domain)
- (ii) Full-length human endogenous FASN with 1 mM Orlistat at 3.3 Å (3.3 Å modifying, 3.0 Å condensing)
- (iii) Full-length human endogenous FASN with 3 mM 1,3-DBP at 2.7 Å (2.7 Å modifying, 2.5 Å condensing), with ACP bound to KS

By refining the modifying and condensing domains separately, combining particles from the three datasets, then performing further classification after symmetry expansion, the authors resolve various sub-states of the condensation cycle including states with ACP engaging with either KS or malonyl/acetyl transferase (MAT). Considering all identified substates together, the authors conclude the catalytic steps of the two monomers in FASN are not correlated.

To attempt correlating a specific sub-state with a specific catalytic conformation, the authors develop a procedure to analyze the distribution of FASN particles with different orientations between the modifying and condensing domains. From their conformational landscape analysis, the authors propose that the condensing cycle reactions occur in a relatively confined orientation between the modifying and condensing domain.

In summary, based on their findings, the authors draw the following key conclusions:

- 1) The enzymatic reactions catalyzed by the two FASN reaction chambers are not correlated;
- 2) FASN function does not require large rotational or swiveling motion between the modifying and condensing multidomains; and
- 3) Physiologically relevant dynamics of FASN are probably limited to the more restricted, so-called “wiggling and jiggling” motions.

We appreciate these summaries, which indeed capture major findings presented in our manuscript.

Whereas each of these conclusions is supported by elegant structural observations, no data is presented to verify the catalytic relevance of these observations. Because no

structurally characterized protein in this study represents an actual catalytically relevant state of FASN (they are either the apo form of the protein or irreversibly inhibited states attained by mechanisms that do not directly relate to the chemistry of any of the constituent enzymes of FASN), the authors' inferences can at best be considered as plausible. While the challenge of definitively proving negative conclusions (e.g., absence of correlation of half-site activity, or that large-scale motions are not needed for synthesizing fatty acids) is not straightforward, the manuscript would be greatly strengthened if at their conclusions were backed up by independent but complementary lines of experimental evidence.

Indeed, in our original submission, we were not able to verify the catalytic relevance of our structural observations. We have now performed additional experiments, included in the revision, to support our structural observations, which we believe providing experimental data to support the conclusion drawn from our structural analysis.

The major new experiments are described in the beginning of this rebuttal. More specific to this comment, we established a procedure that is reliable and robust to demonstrate the functionality of our purified endogenous FASN. With this new assay procedure, we demonstrate that our sample consumes NADPH and produces enzymatic intermediates (Fig 1d, and Extended Data Fig. 2). We demonstrated that placing purified FASN in PEG6000 buffer reduces the rotational motion between the condensing and modifying wings but still consuming NADPH and producing enzymatic intermediates (Fig 5, and Extended Data Fig. 2). Furthermore, our preliminary conformational analysis of FASN in the presence of additional substrates in PEG6000 buffer, under the turn-over condition, shows that similar dense central region (Figure attached above).

We believe this new data support the conclusion that the large rotational motion between the two wings is not a requirement for FASN function. We also performed mass spectrometry analysis of purified endogenous FASN, showing that ACP is properly modified by Ppant, and contains enzymatic intermediates (Extended Data Fig 1f, g and Supplementary Figure 1.c).

Indeed, we intended to reduce the motion by using small molecules, Orlistat and 1,3-DBP. However, our analysis shows that the distribution of particles among different substates remain almost identical (Figure attached above) and the conformational landscape remain largely unchanged (Supplementary Figure. 4b). Thus, we argue that the conclusion drawn from our studies are valid and physiologically relevant.

Other major concerns:

1) The authors' description of their ability to resolve the ACP domain is hyperbolic. In the one PDB file provided, only one of the ACP helices is visible.

Indeed, in structural biology, "resolve" often meant reaching certain resolution and could be misleading in this case. In the revised manuscript, we change it to "capture a density that we interpret as ACP". This revised statement is justified by the identified a specific ACP helix that is modified with additional density at S2156 that we interpret as

phosphopantetheine (Ppant) modification. Our new mass spectrometry data shows that ACP in our endogenous FASN is indeed modified with Ppant, giving us confidence to make such interpretation (Extended Data Fig. 1f, g, and Supplementary Fig. 1c).

2) Structural data for DBP-crosslinked FASN cannot establish which ACP domain is crosslinked to the KS domain of which subunit. The text reveals a bias toward intra-subunit crosslinking (e.g., Figure 1a), but this is by no means obvious from the actual structure. It is not even discussed explicitly. Classic work by Stoops, Henry and Wakil (JBC 259, 12482, 1983; not cited, although it should be) led to the conclusion that this was an inter-subunit mode of association, although that conclusion was questioned in ref. 28. Notwithstanding the high structural resolution claimed by the authors of their DBP crosslinked structure; their work falls short of answering this important unanswered question.

With respect, we disagree with this comment. In schematic figure (Fig. 1a), we specifically avoid imply either cis or trans dimer configuration, but using a red disc to mark the joint. We revise the figure legend to emphasize this.

1,3-DBP crosslink the tip of Ppant with the catalytic C161 (PMID: 6630195, pointed by the Reviewer). A later study (PMID: 10206962) showed that 1,3-DBP crosslink ACP to both KS domains in the dimer. Indeed, the question of which KS does ACP engage, intra or inter domain, is important. Since it has already been addressed, or at least partially, by this earlier study, it is not a key question we intend to address in our study. We now cite this paper (PMID 6630195).

Indeed, we cannot establish specifically which ACP domain is engaged with KS domain of which subunit, with or without 1,3-DBP, because we cannot resolve the hinge to high enough resolution to identify either cis or trans configuration. Based on crystal structure of FASN (PMID: 18772430, Supplementary Figure 3 and 4), both trans and cis dimer configuration are possible. Our landscape analysis cannot exclude either possibility. What we do conclude from our landscape analysis is that ACP mostly engage with KS on the same side, evidenced from the landscape of ACP-KS substates. Therefore, our data did not contract with these early studies.

3) The use of the term “domain” to characterize the “condensation” and “modification” portions of FASN is confusing. Both are multidomains.

We change the term to “wing”.

4) The authors must provide data verifying that the FASN protein preps used in their EM analysis have kinetic parameters expected of this multifunctional enzyme.

As summarized in the beginning of the rebuttal, we established a robust pipeline to measure enzymatic activity of FASN. We measure NADPH consumption by florescent of NADPH and detect enzymatic intermediate linked to Ppant tip by mass spectrometry. Together, our assay shows that our samples consume NADPH and ACP is shuttling

enzymatic intermediate products between different enzymatic domains. This experiment ensures that our purified endogenous FASN is functional. We now included the assay and results in Fig1d, and Extended Data Fig. 2.

5) It is unclear how orlistat helps reduce heterogeneity, thereby allowing the authors to resolve the TE domains. Perhaps a comment about this could be helpful.

Indeed, while it was our intention to reduce heterogeneity by Orlistat, our data shows otherwise. Orlistat does not reduce rotational motion between two wings, nor reduced conformational heterogeneities as revealed from extensive classification. Thus, it does not facilitate visualizing TE domain.

Minor points:

6) Line 119: Characterizing holo-ACP in the condensing cycle

Revised.

7) Line 125-126: 4 or 5 functional groups?

Corrected.

8) Line 317: Give specific examples of other enzymes that function similarly when their activity is upregulated

We now included two examples, Dengue and classical swine fever viral infections, and cite the relevant references.

9) Line 432: How long was FASN incubated with 1,3-DBP or Orlistat?

Approximately 30 minutes prior to making cryo-EM grids. We now updated the method.

10) Line 77: swiveling and swinging

Corrected

11) Line 151: fix

12) Line 164-166: rewrite sentence

13) Line 176: fix

14) Line 205: substrate or snapshot

15) Line 209: KS or MAT

16) Line 216: remove :

17) Line 227: incomplete sentence

18) Line 236: fix

19) Line 301-306: rewrite sentences

20) Line 309: in adult humans

- 21) Line 311: such as in cancer cells
- 22) Line 324: fix
- 23) Line 423: (Sigma-Aldrich, 125903)
- 24) Line 446: remove “respectively”
- 25) Line 463: particles
- 26) Line 468: Extended Data Figs. 2 and 4
- 27) Line 486: Extended Data Figs. 2-4
- 28) Line 489: domains
- 29) Line 511: Supplementary Table 1
- 30) Line 514: Sus scrofa
- 31) Line 873: remove track changes
- 32) Supplementary Fig. 1d: 1,3-DBP
- 33) Extended Data Fig. 2: “maps from Arctica” (?)
- 34) Extended Data Figs. 2-4:
 - a. Fix multiple instances of typo “interative”
 - b. Fix typo “Comebine with...”
 - c. Fix typo “Extraction of momdifying domain”
- 35) Extended Data Fig. 6: Fix typo “Comebine all particles ...”
- 36) Extended Data Fig. 10: Fix typo “Extended Daa”

All these are corrected in the revised manuscript. We appreciate the Reviewer pointing them out.

Referee #3 (Remarks to the Author):

In the manuscript "Structural dynamics of human fatty acid synthase" Choi, Cheng and colleagues elucidate structures of human fatty acid synthase (hFAS) by cryo-EM. They used a strategy of tagging and purifying endogenous hFAS for single particle cryo-EM. They have succeeded in capturing structural snapshots in the condensing cycle of hFAS by visualizing ACP engagements with either KS or MAT domains. The authors have also acquired two datasets with an FDA-approved drug Orlistat, targeting the TE domain, and 1,3-dibromopropanone (1,3-DBP), which induces a cross-link with the phosphopantetheine arm and the KS catalytic cysteine. They have also developed a procedure to analyse the conformational landscape of particles with different orientations between the condensing and modifying domains. With this procedure of conformational landscape analysis they suggest that the formerly postulated hypothesis of a swivelling and swinging motion of hFAS which is thought to shuttle substrate to different enzymatic domains of hFAS by ACP, is not fully necessary. Instead, they suggest that only subtle wiggling and juggling motions are sufficient for ACP shuttling in hFAS. The latter small scale motions appear more feasible in the crowded cellular milieu.

This study is a significant advance in the study and understanding of the structural basis of de novo lipogenesis, by hFAS. It also furthers our understanding how hFAS is targeted by inhibitors and drugs. In particular, it emphasises on how conformational

landscapes influence the activity of this dynamics of this machine which is central to metabolism, but also in various malignancies. Knowledge of conformations adopted by hFAS will be essential to develop specific and allosteric drugs directed against hFAS.

We appreciate this summary of our study and positive comments about our findings.

While this study is interesting, its publication would require several points to be addressed by the authors:

We have now performed further experiments and revised the manuscript accordingly.

- hFAS purification: this preparation is far from pure. I am referring to all the proteins visible in Extended Fig. 1B in the range of 75 kDa to 25 kDa. What are the other proteins co-purified with hFAS.

We agree with the Reviewer that the SDS-PAGE gel shown in Extended Data Fig. 1d is far from pure. However, it is one of the advantages of single particle cryo-EM that the sample for structural study does not need to be very pure. Our target protein can be isolated in image processing, while particles of other contaminants can be discarded. Indeed, from this sample, we also determined a $\sim 6\text{\AA}$ reconstruction that matches the shape and size of bacterial ArnA, which is a homohexamer of 75kDa monomer. This is a contaminant from ALFA-nanobody purification from *E. coli*. The hexamer ArnA has a molecular weight of $\sim 450\text{kDa}$, close enough with FASN dimer ($\sim 546\text{kDa}$), making them hard to be separated by SEC. The identity of this contaminate is also confirmed by mass spectrometry. Since bacterial ArnA is not our target, we did not report it in the manuscript. In the revised manuscript, we include this information in Supplementary Figure 2, as it is not directly related to the focus of this study. The second strong band is the contaminant of eGFP conjugated ALFA-nanobody (Supplementary Fig. 2e).

Are these truncation products? Both issues would have substantial implications for the conclusions drawn by the authors.

We did mass spectrometry of the 250kDa band (Extended Data Fig. 1e). With over 90% coverage, our purified endogenous FASN is the full-length protein with proper modifications. While we cannot absolutely sure that there is no truncated FASN fragments, such particles, even exist, were excluded during classifications. All structural features we discussed in the manuscript are from particles with both intact condensing and modifying wings, as all particles have two sets of Euler angles assignments, one for each wing (Extended Data Figure. 3-5).

On the other hand, we do recognize the potential complications when we perform assays to quantify enzymatic activities of FASN, since any truncated FASN could still be enzymatically active in certain domains. Therefore, to eliminating such complications, we performed more stringent purification to ensure sufficient purity of purified endogenous FASN without obvious additional bands in the SDS PAGE gel (Extended Data Fig. 2c).

We did three biological replicates (three completely independent purifications from three different batches of cells) and obtained reproducible results of NADPH consumptions.

- Orlistat and 1,3-DBP addition: Are these two compounds even bound? Is there any orthogonal validation to ensure their binding? While I understand that it might be difficult to address this in the case of Orlistat which is bound to the flexible and unresolved TE domain, the situation is different with 1,3-DBP. As the authors state themselves, 1,3-DBP should cross-link the catalytic Cys131 with the phosphopantetheine of the ACP. Considering the resolutions attained by the authors for the condensing domain this cross-link should be clearly visible. In the maps provided by the authors, I am having a difficult time finding this covalent attachment. Also this has substantial implications for the conclusions reached by the authors.

It is not clear at all to us what portion of FASN have either Orlistat or 1,3-DBP bound. As mentioned above, in response to Reviewer 2, our initial intention is to use such drug/cross-linker to reduce conformational dynamics, a common approach of structural biology. However, we did this in a rather crude one, by simply incubate the Orlistat/1,3-DBP with purified FASN for ~30 minutes prior to plunge freezing cryo-EM grids. We resolved ACP bound to KS domain in all three samples. As mentioned in response to Review 1 above with attached figure (also in Supplementary Figure 4a), we do not observe significant changes in populations of any specific substates from three datasets.

In terms of the landscapes from three different datasets, there is no significant change especially within the central region where ACP can reach all enzymatic domains. The only small differences we observed is the appearance of several hot spots from Orlistat and 1,3-DBP datasets. However, we could not conclude any specific efforts from our data either about Orlistat or 1,3-DBP. Their contribution in our current study was simply the increase of total number of particles in the final datasets.

Retrospectively, we now have a better understanding about the effect of 1,3-DBP. It cross-links sulfide in the tip of Ppant with the sulfide of C161 of the catalytic triad in the KS domain. However, for endogenous FASN, not every ACP is modified by Ppant (Extended Data Fig. 1f). This phenomenon was also observed previously and was thought to be physiological (PMID: 7567999). Unmodified apo ACP cannot be cross-linked by 1,3-DBP. Those modified, Ppant itself could be linked to intermediate enzymatic products, thus also preventing it from being cross-linked to C161. As shown in the mass spectrometry analysis, even without adding any substrate, we detected heterogenous intermediate enzymatic products from our endogenous FASN sample (Extended Data Fig. 1g).

Therefore, simply incubated 1,3-DBP with purified endogenous FASN can only cross-link holo-ACP without intermediate enzymatic products, which are likely not a major portion of particles in our datasets.

- The authors put forward a catalytic triad formed by C161, H293, and H331 in the KS. This is highly unusual as the two histidines would have similar pKa values. Do the

authors have mutational evidence that supports this catalytic triad? In the model none of the histidine residues are in hydrogen bonding distance. In fact, by the same criteria applied by the authors Glu333 appears to be in suitable distance to support proton transfer through H331. The pKa values of this catalytic triad constellation would make much more sense!

We appreciate this comment. The catalytic triad formed by C161, H293 and H331 was proposed by previous studies (PMID: 34156235 and 30908841), based not only on the crystal structure (3HHD and 6OKC) but also on accumulated mutagenesis studies (such as PMID: 12196027). We now cite these publications in the revised manuscript. E333 was not considered to be part of the catalytic triad. In 3HHD, the distance from the base H293 to acidic E333 is $\sim 10\text{\AA}$, which is too far to be considered as part of the catalytic triad. And unlike C161, H293 and H331, E333 is not evolutionary conserved among β -ketoacyl synthase. Furthermore, we are unable to conduct mutagenesis studies of E333 on endogenous FASN, nor using recombinant FASN, as it is unlikely being properly modified by Ppant (PMID: 37308485). Thus, although our structure shows that E333 is closer to H293 or H331 than that in the crystal structure (3HHD), we thus prefer not to re-define the catalytic triad in the KS domain.

- In all maps provided by the authors the ACP densities (including phosphopantetheine) are considerably weaker than the rest of FAS. Fortunately for the authors the phosphopantetheine residues allow them to establish topology of the pseudo-symmetric ACP. However, the weaker density of the ACP alludes to (conformational) heterogeneity. To which extent does this impact the statement of the authors that the ACP of endogenously purified hFAS contains holo-ACP? Do the MS/MS experiments indicated quantitative holo-ACP?

We again appreciate this comment. In the revision, we now included mass spectrometry analysis and assessed the function of FASN in terms of NADPH consumption and production of intermediates (Extended Data Fig. 1 and 2). We unambiguously identified Ppant modification on S2156 by mass spectrometry (Extended Data Fig. 1f and Supplementary Fig. 1c). We also observed different molecular mass on the Ppant-S2156 residues after incubating purified FASN with NADPH, acetyl-CoA and malonyl-CoA, which we interpret as intermediate enzymatic products (Extended Data Fig. 2e).

- Related to this issue: What are the specific contacts established in the recognition of ACP by hFAS-KS and hFAS-MAT? How do they compare to the interactions of ACP with bacterial KS and MAT? For that matter how do the interactions compare with the recently determined γ FAS-ACP interactions?

From our structures, we were able to isolate two different engagements of ACP with the catalytic site of KS, both are mediated by electrostatic interactions. We can identify two residues, H68 and K70, located in the entrance of the catalytic cavity that make electrostatic interactions with the canonical domain of ACP. It is difficult to establish an exact side chain beyond this observation, as the interaction is dynamic and transient. Since all three reviewers asked the similar question, we now revised Fig. 2d and Extended

Data Figure 8a-b to illustrate this. In this regard, yFAS-ACP interaction is also electrostatic in the recently determined structure (PMID: 37949058). In another recent work on PKS (PMID 39179672), interactions between ACP and KS are similar as our shown in our structure.

-I have a bit of difficulty wrapping my head around the "floating ACP" position described by the authors? Surely there must be some specific contacts of the ACP to hFAS? Otherwise, it is difficult to imagine how classification would yield a singular position. One would expect a washing-out of density effect to ensue. Such contacts would have to be with the modifying domain which would be suppressed by the image processing scheme employed by the authors. This is in fact indicated in the conformational landscape analysis in Figure 5c. I encourage the authors to elaborate on this important point.

- Relatedly, can the topology of the ACP even be established in the floating ACP position? In which direction is the phosphopantetheine arm pointing? Does the position of the phosphopantetheine arm even support the authors' hypothesis that the floating ACP represents an intermediate in transition between KS and MAT?

Since these two comments are related, we address them here together. Indeed, the term "floating ACP" could be misleading. In class with "floating ACP", what we observed is an unaccounted density with the size similar to ACP located in between KS and MAT domain and we interpret it as a transitional state between ACP engagement with KS and MAT. This density was observed in classes identified from classification of the symmetry expanded condensing wing, in which it appears not engaging with either KS nor MAT (Fig. 2h and class 9 of Extended Data Fig. 7a and b). We thus termed it "floating", meaning that it is in transition between KS and MAT. In addition, we also observed similar density in reconstructions of the entire FASN with defined orientations between the two wings (Fig. 5c, Extended Data Fig. 10 and Supplementary Figure 5 and 6), which was analyzed by using cryoDRGN). In these reconstructions, this ACP density is not floating away from the condensing wing, but with some attachment with some part of the wing. And we cannot assign it to any specific functional state. Thus, in the revised manuscript, we change "floating ACP" to "ACP in transition".

- Regarding No-ACP and dynamic MAT groups: The authors speculate that these represent the dynamics in MAT are associated with capturing acetyl- or malonyl-CoA. While the authors correctly admit that ACP is not visualised, can they discern densities on the catalytic Ser581 which correspond to acetyl- or malonyl-esters?

We agree with the Reviewer that this is a pure speculation. In the No-ACP group, while the MAT domain is well resolved, we do not see any density within the catalytic cleft. In the dynamic MAT group, since half of the MAT is not resolved, no additional density is seen within the catalytic cleft. Thus, what we state is pure speculation. Since this similar question was asked by Reviewer 1, we removed this speculation in the revision.

- In line 166, the authors conclude that the absence of any coordination between ACP

hopping and conformations adopted in two monomers suggests that the catalytic steps of the two monomers in hFAS are not correlated. I find this a bit of a stretch as considering the conformational dynamics within this structure, it is not intuitive to me how to derive such a statement from not visualizing certain states. How can the authors distinguish between classification/ alignment errors and functional significance? This is especially even more so, when none of the structural analysis performed by the authors involves the addition of substrates to initiate functional cycles. This statement asks for enzymological analysis which interrogates allosteric communication. The KS-MAT paper by Parthinkar, Grininger and colleagues where such enzymological assays were performed clearly demonstrate positive cooperativity in condensation steps, contradicting the authors' conclusion.

Fig. 4a (in the revised manuscript) shows that, the landscape of different classes where we can assign specific ACP engagements of each monomer with either KS, MAT or none have similar central regions. This observation suggests that within the central region under the same orientation between the two wings, ACP of each monomer can have different engagements with catalytic domains. We thus can conclude that the catalytic steps of the two monomers in FASN are unsynchronized, meaning that with ACP engagement in one monomer, the ACP of the other monomer could potentially be in any substate of the condensing cycle. We realize that the word "no correlation" may mislead to "no cooperativity" of the two monomers. In the revised manuscript, we change it to "unsynchronized".

Our observation that the enzymatic reactions of two monomer is unsynchronized does not support nor against the conclusion of positive cooperativity of KS reported in the paper mentioned by the Reviewer (PMID: 31811668).

- The same applies to the entire paragraph starting from line 169: The authors use this paragraph to question the swiveling and swinging model. I personally find this problematic for two conceptual reasons: The pre-requisite for this statement by the authors is that they fail to reconstruct 3D structures where a single substate in the condensing domain would be associated with a specific mutual orientation of condensing or modifying domain. This does not occur in their analysis and therefore prompting the conclusion reached by the authors. However, they fail to address potential issues with sampling space. It is very conceivable that the numbers of particles sampled by the authors are not exhaustive enough for observing the low-populated elusive states.

With respect, we disagree with the Reviewer on this point, as we elaborate here:

What we described in this paragraph is rather logic. According to "swiveling and swinging model", which is rather the same as "excessive conformational constraint" model, each substate identified by classification (Extended Data Fig. 7f) should produce a reconstruction with two wings resolved to a specific orientation, even at modest resolution. Instead, the reconstructions we can calculate from each substate are still two separate wings, but not both together. Indeed, if the number of particles increase significantly, it is

possible to generate a series reconstruction of the same substate with different orientational angle between two wings. If that is the case, it would produce the same conclusion that there is no one-to-one correlation between any specific substate with a specific orientational angle between two wings.

This promoted us to further analyze the landscape, without increasing the number of particles further from what we already have in our datasets. Combining the classification (Extended Data Fig. 7f) with landscape analysis, we did reconstruct such structures, as shown in Fig. 4 and Extended Data Fig. 10, where the structure of any specific substate we identified can be reconstructed. As shown in Fig. 4c, e and f, specific substates with clear feature of ACP can be reconstructed in multiple different orientations between the two wings. Furthermore, each substate has its own landscape, shown in Extended Data Fig. 10. Therefore, we can unambiguously conclude that there is no one-to-one correlation of any substates with any specific orientation between the two wings.

Secondly, they reach this conclusion in a resting state enzyme. It is very likely that an energetic barrier exists which prevents the enzyme from reaching cooperative behaviour in the absence of substrates.

Our FASN is purified from endogenous source, which are properly modified with Ppant, and even contains intermediate enzymatic products as shown in mass spectrometry data included now in the revision (Extended Data Fig. 1f, Extended Data Fig. 2e, and Supplementary Fig. 1c). We also observed ACP engagements with KS or MAT. Thus, they are not non-functional enzyme. Each particle could be in different “resting state” without supplement of additional substrates but correlate with various functional states these FASN particles trapped when being purified.

Therefore, to reach this conclusion the authors should at least analyse another dataset in the presence of substrates in turnover conditions. This should be followed up by conformational landscape analysis under these conditions.

Nonetheless, we appreciate this suggestion very much. In addition to analyze the landscape FASN in PEG6000 buffer, as summarized in the beginning of this rebuttal, we also incubate 500~1000-fold excess acetyl-CoA, malonyl-CoA and NADPH with FASN in PEG6000 buffer 5~10 minutes prior to making cryo-EM grids, collected a modest cryo-EM data and performed similar analysis. The landscape with this dataset shows very similar overall feature as the landscape of FASN without added substrates. Thus, under the turnover condition does not change the landscape dramatically (Figure attached above).

Detailed analysis of substates and landscapes under this turnover condition would require much larger dataset and more extensive processing, exceeding the timeline for resubmitting this revised manuscript. We thus only include the new results of FASN landscape under PEG6000 buffer without added substrates in the revised manuscript. It supports our conclusion that reduced rotation motion of FASN by PEG6000 does not impact FASN function (Fig. 5).

- The conformational landscape analysis performed by the authors is an appropriate way to quantitate dynamics within hFAS. However, this analysis is not conceptually novel per se. Stark and colleagues have performed such analyses before for the bacterial ribosome, the 26S proteasome, the spliceosome, and the yeast FAS. Frank and colleagues have also performed such analyses for bacterial ribosome. I find it a bit strange that the authors do not cite these papers to put their own efforts into context with earlier developments in the field. This should also be addressed in the conclusion and discussion section.

We appreciate this comment. Indeed, many others have performed the landscape analysis, including Holger Stark and Joachim Frank, particularly when they studied the ratchet motion between large and small subunits of ribosome. This also includes the manifold analysis developed by Joachim and his colleagues. Furthermore, multi-body refinement implemented in Relion, 3DVA in CryoSPARC and cryoDRGN were all developed to analyze conformational landscape, focusing on the conformational flexibilities.

Our analysis focus on characterizing rotational relationship between two “near” rigid bodies. It is true that the ratchet motion between large and small subunits of ribosome are in this category. A key difference between the analysis performed by many previous studies (not all) and our analysis is that we do not use PCA, but visualize the landscape directly in real space using Cartesian coordinate of three orientational angles. This is indeed somewhat similar as Stark lab used in their early study of ribosome (Fig. 4b, c, d of PMID: 20631791), but different from their study of 26S proteasome, which was after PCA analysis.

There is also a difference between our analysis with other machine learning based analysis of conformational landscape, including 3DVA and cryoDRGN, which focus on continuous flexibilities of the reconstructions. There, the landscape is interpreted in latent space. We did perform cryoDRGN on FASN dataset. Giving the entire dataset, cryoDRGN failed to analyze the landscape. However, isolating particles around a specific orientational coordinate, cryoDRGN produced results that matches our analysis (shown in Supplementary Figure 5 and 6).

In the revised manuscript, we now cite all these previous studies. We are also preparing a separate manuscript to describe our approach in detail, and to compare the results/performances of our approach with that of previous methods.

Indeed, even for conceptual idea of analyzing the orientational landscape between two rigid bodies as we did here, we are not the first one. There is one previous study, which attempted the same approach to study ribosome ratchet motion between small and large subunits (PMID: 3282006). However, in this paper, the formula used to calculate the angles between two rigid bodies was incorrect, resulting all landscapes with a Gaussian distribution as reported in that publication. As far as we know, the method described there was used in only by one another study, similarly produced landscapes with Gaussian distributions.

Our landscape analysis formula is correct, as we validated it by calculating reconstruction from selected coordinates and produced reconstructions that match the orientational angles. We did not cite this specific paper, for collegiality reason.

- It is important that the inhibitors constrain the sampled conformational landscape. This is very much consistent with earlier findings by Stark and colleagues in the 26S proteasome that 20S proteasome inhibitors allosterically inhibit motion in the 19S regulatory particle, and the gamma-subunit the rotation of yeast FAS. Surely, this would warrant citations? More importantly this alludes to an important principle in the function of macromolecular complexes which the authors fail to elaborate on.

We appreciate this comment. As discussed above, we cannot conclude beyond observing subtle in the landscape from the samples with Orlistat and 1,3-DBP applied. We thus refrained from further discussing of the effects of these two small molecules. Nonetheless, the purpose of using small molecule to reduce conformational heterogeneity was inspired by these previous studies, including the work from Stark lab. In the revised manuscript, we now elaborate this and cited this paper.

- I lack some specific experiments on the conformational landscape analysis to elaborate on this: How does the conformational landscape look for hFAS in turnover conditions?

As described above, we now performed additional experiments, including collecting data from FASN incubated with all substrates, this under the turnover condition. As show in the Figure attached above, the central hot region remains similar, supporting our conclusion drawn from sample without incubating with additional substrates.

How do the employed inhibitors Orlistat and 1,3-DBP modulate the conformational landscape in turnover conditions?

We did not study purified FASN incubated with all substrates as well as Orlistat or 1,3-DBP. We intended to do these experiments, but they are beyond the scope of the current manuscript.

To support a main conclusion of our study that large rotations are not required for FASN function, we did study FASN in PEG6000 buffer, which constrains the landscape to central region but does not alter the function of FASN (Fig 5b, and figure attached above).

- The conformational landscape analysis in turnover conditions would also address the important question if the dynamics carefully investigated by the authors correspond to off-pathway states or on-pathway states? Whether there is indeed no coupling between monomers in hFAS? If swiveling really exists? How ACP movement is correlated with the conformation of both monomers?

These are indeed great questions and suggestions, which we appreciate and will follow up in our further studies. However, we did capture ACP engagement with KS or MAT, and verified that they are Ppant modified ACP. Thus, we argue that the conclusion derived from our landscape analysis and new experiments included are physiologically relevant.

- Spots of high particle densities in the conformational landscape should give rise to particles which are amenable to high-resolution reconstructions. Is this the case? 10000 particles if conformationally homogeneous should give rise to 3 Å structures. Should this not be the case, this would allude to a sampling problem where particle numbers are just not high enough. Alternatively, it would suggest that conformational landscape analysis has not reached convergence and some more additional hidden substates exist. Could the authors comment on this?

This is correct, that coordinates with high particle densities in the landscape should correspond to a reconstruction with a higher resolution. Shown in Extended Data Fig. 9c, reconstructions calculated from coordinates P2 (8,784 particles) and P4 (9,551 particles) with a radius of 7°, yielded higher overall resolution (~10Å) compared to others, such as P8 with the same radius (1,392 particles around 15Å). The reconstruction was performed based on the Euler angles from the condensing domain without further refinement of either wing. Thus, the condensing domain has higher resolution. We also did reconstructions based on the Euler angles assigned to the modifying domain (data not shown), these reconstructions maintained the same orientational relationship between the two wings but got a higher resolution of the modifying wing than that of the condensing wing.

The resolution of a reconstruction derived from a specific location of the landscape mainly depends on two factors, the radius of the region selected around the specific coordinate and the total number of the particles within the region, which is also affected by the local density within the region. In the dataset of FASN, the particle number of a local hotspot is still not sufficient to produce a high-resolution structure. We thus test this with datasets of completely different protein system, such as characterizing ratchet motion between two subunits of ribosome, the large subunit (LSU) and the small subunit (SSU). In one dataset that we tested, a clear hotspot in the landscape was seen. A reconstruction from this hot

spot (with a radius of only 1° at [0°,0°,0°] containing 217,199 particles) based on the Euler angles from LSU achieved a slightly higher overall resolution (2.48 Å) compared to the one from the entire dataset (1,946,044 particles with 2.49 Å), without further refinement. Moreover, if we assess the local resolution distribution, it becomes evident that the density quality in the SSU region within this hot spot is significantly enhanced to that from

the entire dataset. This is expected, as given the restrictive 1° radius setting, which limits the rotational motion to within 1 degree, thereby noticeably improving the density quality in the SSU region.

A related question from this comment is that whether the numbers of sampled particles would influence the accuracy of the conformational landscape. In our studies, we used a Relative Nearest Neighbor Distance (RND) to color local particle density in the landscape. This scale is invariant of the total number of particles in any specific dataset. Thus, we can compare landscapes of the same protein under different conditions without influenced by the number of total particles in each dataset. We illustrated this in the figure attached here, where the landscape are shown with different total particle number in the dataset. Thus, landscapes of substates, or different samples can be compared directly without concerning the particle numbers of each dataset (Figure attached right). We are currently preparing a separate manuscript to describe all these details.

- Minor points: some aspects of the manuscript are not clearly worded and warrant some re-writing. This applies especially to the conceptual parts of the manuscript.

We now revised the many parts of the manuscript to make it clear. New text or the part with extensive revision are colored in blue.

Referees' comments:

Referee #1 (Remarks to the Author):

First and foremost, I would like to emphasize that this manuscript is highly interesting. It greatly improved during revisions. In principle, the ms merits publication in Nature. I am also grateful for the detailed responses to many of the points I have raised, and for sharing structural information as reviewer only material.

We thank the Reviewer for her/his support and appreciate the further suggestions/comments. In this re-submission, we revised the text to address these remaining/additional concerns.

However, I still have a few concerns that should be addressed before the ms is published:

(i) The work by Choi et al. refines the structural understanding of FAS by demonstrating, for the first time, that the inherently pronounced conformational variability is unlikely to be essential for fatty acid biosynthesis and is plausibly not fully utilized within the cellular environment. However, I would like to note that I continue to disagree with the authors' emphasis on stating that the "swiveling and swinging model" reflects the current perception of the FAS mechanism, and the new data by Choi et al. now revise this model. Although the large conformational variability has been demonstrated by several methods, there is no model that conformational variability is obligatory to run the FA cycle.

Again, we appreciate this comment. Indeed, there is no experimental evidence that conformational variability is obligatory to FASN function. We now revised the paragraph, pointing out that although early structural studies of FASN promoted notion that "rotational motion would drag ACP to two different faces of FASN" (cited from reference 18), there is no direct experimental evidence that such motion is obligatory for FASN function.

(ii) Second, I appreciate the additional insight into ACP docking. However, several conclusions regarding ACP docking to the condensing wing, as drawn by the authors, remain unclear and should be clarified by providing additional information (without necessitating new data collection).

We now revised the text to address this comment. Detailed responses and revision are described below.

(iii) The claim that monomers are unsynchronized or do not collaborate does not seem supported by cryo-EM data. A more careful discussion is necessary.

We revised the text and respond to this comment below.

Major and minor points:

(1) line 55: it should probably mean „metazoans“ or „metazoa“ as a taxonomic group used in singular

Corrected

(2) line 56/56/59...: wings are still named “domains“

We now revised this paragraph to separate architectural description of FASN monomer from dimer. When describe monomer, we use “domain”. When describing dimer, we define “wing” as dimeric condensing or modifying wings and used the term throughout the remaining manuscript. We hope the revised text is now clear.

(3) line 78 and line 179: „Consequentially, continuous large rotational motion between the wings is vital for FASN function.“ Although it has been argued that swiveling can drag the ACP closer to KS to improve docking, a “swiveling and swinging model” does not reflect a broad current perception of the FAS function in the community. Neither the revised manuscript nor the rebuttal letter convincingly substantiate this model; e.g., literature with direct data to the “swiveling and swinging model” are not included (see statement line 179 without any citation „The “swiveling and swinging” model suggested that rotation between two wings is essential to position ACP with substrate to different enzymatic domains.“). The review cited in the rebuttal letter and manuscript PMID: 37856285 does not support this model but instead states (i) that large rotations can occur, which explains the knockout experiments by Joshi et al., and (ii) that there are two levels of conformational variability, namely local positional variability of the ACP through flexible linkage to the KR and TE, and overarching conformational variability of the wings. This does not imply the necessity of swiveling and swinging for fatty acid synthesis.

Line 78: We now revised this paragraph, as explained above in response to point (i).

Line 179: We now revised this paragraph, removed the statement of “swiveling and swinging” model.

(4) line 134: The more detailed presentation and interpretation of data is in general

appreciated. However, there are some open points. Two ACPs are docked to the KS is slightly different positions. (a) In MS analysis apo-ACP has been identified. How likely is it that in the one ACP position, the gray density, the ACP is without PTM and this position is not physiological, at least not reflecting one step of the proposed two-step model. Or there could also be the modification with malonyl such that the two states are not sequential but refer to transacylation and condensation. ACP:KS complexes have been characterized for bacterial FAS type II by Burkart and coworkers (see e.g. 10.1038/s41467-020-15455-x). Are positions similar and may the bacterial structure help in understanding ACP docking to human FAS KS?

We appreciate this comment. Indeed, with electrostatic interactions, even without Ppant modification, ACP can still dock at the catalytic cavity of KS. Thus, our interpretation of two-step engagement was too speculative. We now removed this sentence.

(b) I am grateful for providing maps. It seems that the helix in the condensation wing model does not very well fit the density. And it does not align with the full ACP docked. Probably a repositioning of the helix is necessary.

We thank the reviewer for pointing this out. We now remodeled the helix so that it aligns with the full ACP docked. We now deposit one more PDB to show monomeric condensing domain with the helix of ACP and the Ppant attached to the end of the helix. The deposition is in process and accession codes will be available shortly.

(c) The authors interpret that ACP docks to MAT transiently. They state „density resembles the shape of ACP and loosely attaches to a well resolved catalytic cleft in the MAT domain“. While this is plausible, I suggest that also here the density should be presented with docked ACP, and the pdb file shared. Again, I am grateful that authors shared files. I agree that the position of the extra-density well aligns with ACP at the active site (close to binding cleft).

We revised the Extended Data Fig. 8 to show docked rat ACP (Extended Data Fig. 8, panel b, e and f). We prefer not to deposit PDB with docked rat ACP, concerning that readers may take it as an actual atomic model of how ACP docked to MAT. Instead, we now included three movies to show ACP docking in ACP-KS, ACP-MAT and ACP in transition groups.

References 36 and 37 do not refer to fast MAT kinetics. Better its references to fast kinetics of MAT are: Rangan and Smith; 10.1074/jbc.271.49.31749; and Rittner et al.; 10.1021/acscchembio.7b00718).

Appreciate the suggestion, corrected.

(d) In the rebuttal letter, the authors argue that for ACP in transition „ACP density is not floating away from the condensing wing, but with some attachment with some part of the condensing wing.“ **This is not mentioned in the manuscript.** Structural information provided as reviewer only material dock ACP in distance to about 15 Å to the condensing wing, which seems too far away to have direct contact. A more careful discussion on ACP in transition seems appropriate.

In Fig. 4c and 4e, we observe ACP density in between KS and MAT. In some cases (such as shown in Fig. 4c middle panel in third and fourth rows), ACP is attached to condensing wing. In other cases (such as Fig. 4c middle panel in first row) is away from the condensing wing but attached to the modifying wing. The classification shown in Fig. 2h is an average, where density is seen as floating in between KS and MAT. Thus, from our data, we cannot conclude ACP either engage with the condensing wing or detached from it while in transition. We thus think that “in transition” is the right description but prefer not to elaborate further.

(5) line 178: To the unsynchronized monomers (also referring to Rebuttal Letter: Comment to Reviewer#3 line 166): Choi et al claim that there is no synchronization between the monomer (in the original version of the manuscript the term „no correlation“ was used). Reviewer#3 addresses an important point to the heterogenous loading of ACP which may hamper synchronization. Of note, there is a new paper by the Grininger lab further substantiating cooperativity in the KS mediated elongation. Gusenda et al. suggest that cooperativity involves ACP binding such that the binding of ACP to one KS protomer favors the binding to the other one <https://doi.org/10.1002/anie.202412195>. Cooperativity means that there is synchronization; however, synchronization does not mean that it is absolute. Thus, cooperativity in the KS-mediated condensation reaction can be manifested in an increased prevalence of ACPs docking to both KSs as it would happen without cooperativity. Data by Choi et al. cannot rule out such synchronization at the level of cooperativity as observed by Gusenda et al. Thus, it is recommended that authors are more cautious in their statements regarding “no collaboration” or “no synchronization” of monomers.

We are aware of this new paper by Grininger lab, which used the same method as their previous work. The system uses free ACP and truncated FASN without modifying domain. Cooperativity was concluded from the Hill coefficient larger than 1. This is a very different system than the system used in our study, where no free ACP exist and full length FASN

is used. Nonetheless, we do realize that without seeing something in our dataset cannot rule out its existence. We thus revised the text to tone down our statement.

(6) line 304: The authors conclude that substrate-shuttling by human FAS follows all concepts - free diffusion, medium conformational constraint and excessive conformational constraint. This does not align with Kira Weissman's original concept that a megasynthase can follow in principle just one of these concepts: The free diffusion model means that there is no steering effect by the catalytic core (the fungal FAS follows rather this concept, although Stark and coworkers revised the view slightly). The medium conformational constraint model means that there is higher order (large scale) conformational variability of the catalytic core that supports or hinders the ACP from reaching a subset of catalytic domains. The excessive conformational constraint model means that ACP is guided to the catalytic domains by the higher order conformational variability. The free diffusion of ACP, as the local conformational variability, is implicit in medium conformational constraint and excessive conformational constraint. Given data presented in paragraph starting with line 220, substrate shuttling in human FAS seems to follow the medium conformational constraint model.

We greatly appreciate this comment. Indeed, the reviewer is correct. We now revised this paragraph in the Discussion.

(7) line 334: The authors state: „Likely, the open reaction chamber in PKS also allows enzymatic reactions across different molecules to increase overall enzyme productivity.“ This statement should be toned down. It is a hypothesis, without direct data.

We now revised this paragraph to specify that this is our speculation.

In sum, I strongly support publication of the manuscript. I suggest that files of ACP docking are shared with the public.

We thank the Reviewer for his/her support.

Referee #3 (Remarks to the Author):

The revised manuscript "Structural dynamics of human fatty acid synthase" by Yifan Cheng and colleagues, addresses many aspects raised by the other reviewers and myself. More specifically, overall the revised manuscript is toned down in the most contentious statements and offers a series of control experiments essential to validate

the statements made by the authors. I very much appreciated and commend the efforts made by the authors to resolve many of the raised issues. These include especially the more advanced characterization of the biochemical preparations of FASN used for structural analysis, the enzymatic assays and clarification of the methods employed for conformational landscape analysis. The PEG6000 experiment indeed validates the utility of the conformational landscape analyses as performed by the authors in this manuscript. I appreciate the balanced discussion throughout the revised manuscript and inclusion of the citations I had suggested in the original review. This has certainly contributed to the legibility and quality of the revised manuscript.

Nevertheless, I do not feel that the main criticism which had voiced in the original review has been satisfactorily addressed. I encourage the authors to address these issues below:

We appreciate the comments and revise the text accordingly.

The authors have not captured any substates in the modifying wing of the FASN. Therefore their statements with regards to conformational dynamics to the overall catalytic reactions in FASN are unjustified and speculative. The only substates addressed by the authors are those within the condensing wing. Therefore, I would appreciate if the authors restricted their statements within the entire manuscript solely to the condensing domain.

We appreciate the suggestion of focusing our conclusion on the condensing cycle. Indeed, we have not captured any specific substate in the modifying cycle. Thus, our conclusion should be strictly constraint to the condensing cycle. In the revised manuscript, we now thoroughly adapted this writing.

- Change the title to: "Structural dynamics of human fatty acid synthase in the condensing cycle"

We appreciate this suggestion and have revised the title accordingly.

- Abstract: We captured conformational snapshots of various functional substates in the CONDENSING CYCLE... Together, we reveal that FASN function does not require large rotational motion between its tow major functional domains in the CONDENSING CYCLE, and that the CONDENSING CYCLE catalytic reactions... Our data thus provide a new composite vie of FASN dynamics during the fatty acid synthesis CONDENSING CYCLE.

We revised the abstract accordingly.

The same applies to all statements within the entire revised manuscript!

We have revised the manuscript and make it clear that our findings are restricted to condensing cycle.

Point to point response:

1. We require that the cryo-EM reporting table is included in the Extended Data. The table is currently included as Supplementary Table 1. Please note that we only allow up to 10 Extended Data items (tables and figures), and that Extended Data tables also have to be submitted in .jpg, .tif or .eps format. See 'EXTENDED DATA' and 'DATA DEPOSITION' below for detailed guidance. We suggest that you move ED figure 1 to the Supplement instead as it has a large alignment with small text, which doesn't display well in the HTML format of our Extended data.

Following the suggestion, we moved the cryo-EM reporting table to Extended Data Table 1 and move the original Extended Data Figure 1 to the original Supplementary Figure. As such, the total number of extended data items are 10.

All panels of original Extended Data Fig. 1 and the original Supplementary Figure 1 are re-organized into new Supplementary Figure 1 and 2, with protein purification data in Supplementary Figure 1 and mass spectrometry data in Supplementary Figure 2.

2. The number of main text references should be 60 in total or less - currently there are 70. Also, there are no methods references. Please create a separate reference list for the methods with continuous numbering.

We now separated the references of method section from the main text. The main text now has a total of 54 references.

3. Please ensure all main figure legends are 300 words or less.

We shortened the legends of Fig. 3 and 4. Now legends of all five main figures are less than 300 words.

Also, please reduce subheadings to 40 characters (with spaces) or less.

All subheadings are 40 characters (with spaces) or less.

4. Please ensure that the text size in all figures is at least 5 pt Arial.

We confirm that text size in all figures is at least 5 pt Arial.

5. We require that the cryo-EM reporting table is included in the Extended Data. [### The table is currently included as Supplementary Table 3. #####] Please note that we only allow up to 10 Extended Data items (tables and figures), and that Extended Data tables also have to be submitted in .jpg, .tif or .eps format. See 'EXTENDED DATA' and 'DATA DEPOSITION' below for detailed guidance.

As stated above, and following the instruction, we moved the original Extended Data Figure 1 to Supplementary Figure. Moved the Cryo-EM Reporting Table to be Extended Data Table 1. Thus, the total number of Extended Data items are 10.

6. Please organize your article file as follows: title & front matter, summary, main text, references, methods, data (&code) availability statement, additional references (with continuous numbering), acknowledgements, author contributions, additional information, figure legends, extended data figure legends.

The manuscript is now organized following exactly as instructed.

7. There are potential third-party rights issues in some figures (i.e. schematics, illustrations). It is your responsibility to obtain the right to use any items (figures, tables, images, videos or text boxes) that are

reproduced (or adapted) from material for which you do not hold copyright and to give proper attribution to the creators of that work. This includes work that has previously been published elsewhere, but also templated from e.g. BioRender. Regardless if third-party material is included, please fill out our Third Party Rights Table and submit it with the revised manuscript. If you generated all illustrations yourself, this can remain empty (except for the author / manuscript number fields).

We confirm that there is no third-party issue in any of our figures. All panels are created by authors.

8. Please provide a supplementary information guide (see 'SUPPLEMENTARY INFORMATION' below). The supplementary videos currently do not have titles/legends, please provide them here.

We now added a table of content in the first page of Supplementary Information document.

9. Please check the comments in the attached checklists (Reporting Summary/ Editorial Policy Checklist) and revise them and/or the manuscript accordingly.

All comments in the Reporting Summary and Checklist are addressed.

10. For any Supplementary Figures, please check and confirm that:

* If data is presented as bar charts, individual data points are shown using overlaid dot plots.

The only bar chart is Supplementary Figure 5a, in which, every bar has only one data point. We included the data point using overlaid dot plots.

* The n number (i.e. the sample size used to derive statistics) is provided and defined as a precise value (not a range), using the wording “n=X samples/cells/independent experiments” etc. where applicable.

Supplementary Figure 1c: In the figure panel, we corrected the label to “(n=6 independent biological replicates)” and “(n=8 independent biological replicates). We also labeled the insertion panel with “(n=3 independent biological replicates)”

* Any chart axis, error bars, scale bars, symbols and colour scales are defined.

Scales in panels of all Supplementary Figures are now defined.

* Any statistical tests used for data analysis are specified and exact p-values are provided either on the figures themselves, in the legend or in the Source Data file.

p-values are labeled

* Wherever representative data such as micrographs are shown, the legend indicates how many times the experiment was repeated with the same results.

The total number of micrographs are indicated in the legend of Extended Data Fig. 2a.

11. Please ensure that all PDB/EMDB accession codes are correctly given in the manuscript and Data Availability statement, and that the datasets are made publicly available as soon as possible.

Now, all PDB and EMDDB accession codes are correctly given.

TRANSPARENT PEER REVIEW: Nature offers a transparent peer review option for original research manuscripts. We encourage increased transparency in peer review by publishing the reviewer comments and the authors' rebuttal letters if the authors agree. This material is made available as a supplementary

peer review file. **Please state in your cover letter either ‘I wish to participate in transparent peer review’ to opt in, or ‘I do not wish to participate in transparent peer review’ to opt out.** Failure to state your preference will result in delays in accepting your manuscript for publication. If you wish to opt in to transparent peer review please provide your response to reviewers as a Word file where possible.

We prefer transparent review. It is stated above in the cover letter.

Note: We allow redactions to authors’ rebuttal and referee comments in the interest of confidentiality. If you are concerned about the release of confidential data, let us know specifically what information you would like to have redacted. We cannot incorporate redactions for any other reasons. Referee names will be published in the peer review file if the referees have signed their comments to authors, or if they explicitly agree to release their name. For more information, see our FAQ page.

There are indeed unpublished data included in the rebuttal but not in the manuscript. However, we do not have concern of including them in the published rebuttal for transparent review processes.

ORCID--IMPORTANT: All authors identified as ‘corresponding author’ on the manuscript must have an ORCID associated with their Nature account before submitting the final version of the manuscript. While non-corresponding authors do not have to link their ORCIDs, they are encouraged to do so. Please note that it will NOT be possible to add/modify ORCIDs at the proof stage. Thus, if they wish to have their ORCID added to the paper they must follow the above procedure prior to acceptance. If you have any issues attaching an ORCID identifier to your Nature account, please contact the Platform Support Helpdesk.

All authors have ORCID.

In order to avoid delays with publication of your manuscript, please read the guidelines below carefully before resubmission of your manuscript.

STATISTICS: When revising your manuscript, you should ensure that any statistical analysis used is sound and that it conforms to our guidelines. A collection of articles explaining the basics of statistical analysis and advice on how to best present it can be found here.

We confirm that statistics in this manuscript follows the guidelines of Nature.

REPRODUCIBILITY: To ensure that the quality and transparency of methods and statistical reporting (as discussed here) are sound before the paper is published, we have reviewed your Reporting summary and Editorial policy checklist editorially. I have attached two documents: one listing specific issues related to your manuscript and one containing an annotated version of the Reporting summary. Please ensure that, as well as the more general points below, the points highlighted in the attached documents are addressed in full, both on these forms and within the manuscript. Both forms should be uploaded as a “Related Manuscript” file type. The Reporting summary will be published with your paper.

All comments are addressed, missing information is provided, and mistakes are corrected.

LENGTH: In print, biological sciences papers do not normally exceed 8 pages on average; the final print length, however, is at the editor’s discretion. The typical length of an 8-page article with 5 modest (quarter-page) display items is 4300 words. If a composite figure (with multiple panels) must occupy at least half a page in order for all the elements to be visible, the text length may need to be reduced accordingly to accommodate such figures. Essential but technical details can be moved into the Methods or Supplementary Information (see below).

In this case, we feel the current length of the paper is appropriate, so no further shortening is necessary; you should not significantly add to the text when revising.

In the final revision, we removed some redundant text and slightly shortened the main text by about 200 words. We did not add any significant text.

TITLE: Titles cannot exceed 75 characters (including spaces); they must not contain punctuation.

The title of our manuscript contains 72 characters, including space.

SUMMARY PARAGRAPH: Papers start with a fully referenced, bold paragraph, ideally of about 200 words, aimed at readers in other disciplines. Numbers, abbreviations, acronyms or measurements should be avoided unless essential. The summary paragraph consists of 2 to 3 sentences of basic-level introduction to the field; a brief account of the background and rationale of the work; a statement of the main conclusions (introduced by the phrase 'Here we show' or its equivalent); and a conclusion of 2 to 3 sentences putting the main findings into general context so it is clear how the results described in the paper have moved the field forward. A downloadable, annotated example is available here.

We have 185 words in the Summary paragraph.

MAIN TEXT: If further introductory material is necessary, the main text can begin with up to 500 words of introduction expanding on the background to the work (some overlap with the summary is acceptable), before proceeding to a concise, focused account of the findings, and ending with 1 or 2 short paragraphs of discussion. Sections are separated with subheadings (up to 40 characters including spaces) to aid navigation.

We confirm that the main text of the manuscript follows this guideline.

REFERENCES: As a guideline, most papers should include no more than 50 main text references; all additional references can be cited in (and listed after) the Methods section, as detailed below.

Total number of references of the main text is 54. Additional references cited in the Method section are listed separately as "additional references".

FIGURE LEGENDS: These should be listed sequentially after the main text references and not in the figure files. Each legend should begin with a brief title for the whole figure and continue with a short description of each panel and the symbols used. Legends should not exceed 300 words each.

We confirm that the legend for all five main figures are less than 300 words.

Each figure legend should contain, for each panel where relevant, the following information:

* the exact sample size (n) for each experimental group/condition, given as a number, not a range;

All figure legends are in compliance with this requirement.

* a description of the sample collection allowing the reader to understand whether the samples represent technical or biological replicates (including how many animals, litters, cultures, etc);

All figure legends are in compliance with this requirement.

* a statement of how many times the experiment shown was replicated;

In compliance.

* definitions of statistical methods and measures:

* very common tests (e.g. t-test, simple Chi-square tests, Wilcoxon and Mann-Whitney tests) can be identified by name only, but more complex techniques should be described in the Methods;

In compliance

* whether tests are one-sided or two-sided;

N/A

* whether there are adjustments for multiple comparisons;

N/A

* the statistical test results (e.g., P values);

In compliance

* the definition of 'center values' as median or average;

In compliance

* the definition of error bars as s.d. or s.e.m.

In compliance

Descriptions that are too long for the figure legend should be included in the Methods section.

In compliance

METHODS: The Methods section, which provides the full, step-by-step instructions that would allow other researchers to replicate the results, is included after the main text figure legends. The Methods section will not appear in print but will appear online in the full-text HTML and PDF versions. The Methods section should be written as concisely as possible but should contain all elements necessary to allow interpretation and reproduction of the results. If there are additional references (in the Methods section, Supplementary Information, etc), their numbering should continue from the last entry in the main text reference list, and they should be listed following the Methods section. Specialized methods that require chemical structures, figures, or tables cannot be accommodated in the Methods section of the main text file. If such information is part of the Methods, the entire Methods section must instead be included within a Supplementary Information text file.

In compliance

MAIN TEXT STATEMENTS: Several statements (which will not appear in print but will appear online in the full-text HTML and PDF) are required after the Methods (and additional references, if present). First, there should be an Acknowledgements section, listing grant/financial support. Next, we require a detailed Author Contribution statement; the specific contributions of each author, particularly in terms of which authors performed which specific experiments, must be listed. This is followed by a Competing Interest statement. Financial and non-financial interests should be noted here, as well as any patents; patent information should include at a minimum patent number, what is covered by the patent, and who submitted the patent application. Finally, an Additional Information statement should include information regarding reprints and permissions and name the author(s) to whom correspondence and requests for materials should be addressed. Formatting details and an example are available here.

In compliance

DATA AND CODE AVAILABILITY STATEMENTS: Any manuscript reporting original research must include a Data Availability statement that makes transparent to the reader the conditions of access to the “minimum dataset” that is necessary to interpret, verify and extend the research in the article. This minimum dataset may be provided through deposition in public community/discipline-specific repositories, custom proprietary repositories (for certain types of datasets), or general repositories like Figshare, Zenodo and Dryad. We strongly discourage providing large datasets in Supplementary Information; the preferred approach is to make data available in repositories. More information on Nature Portfolio’s reporting standards and guidance on preparing your Data Availability statement can be found here.

Both Data Availability and Code Availability statements are included at the end of Method.

For all studies using custom code or mathematical algorithms that are deemed central to the conclusions, a Code Availability statement must be included, indicating whether and how the code or algorithm can be accessed, including any restrictions to access. The Code Availability statement is listed as a separate section after the Data Availability statement but before any additional references. Code should be deposited in a DOI-minting repository such as Zenodo, Gigantum or Code Ocean and cited in the reference list. Authors are encouraged to manage subsequent code versions and to use a license approved by the open source initiative. Additional details can be found here.

The program (CryoROLE) used in this study was developed for this study and is released in GitHub as open source. We have included a Code availability statement.

DISPLAY ITEMS: We suggest that you take stock of all data that have been generated throughout the review process and ensure that only the data most central to the conclusions are presented in the main text figures. Any figures included within the main text file during the review process must be removed from the final main text file and uploaded as separate, individual files; they will be integrated into the main paper in print and online. An overview of the key features of this presentation may be found here.

In compliance

Figures should be comprehensible to readers in other disciplines and assist in understanding of the paper. Main text figures (but **not** Extended Data) must be provided in production-quality versions in an editable format (i.e., .ai, .cmx, .cdr, .doc, .eps, .pdf, .ppt, .ps, .psd, .svg and .xls); we cannot accept figures in .cvs, .gif, .jpg, .png and .tif formats. We highly encourage you to consult our artwork guidelines. They should be as small and simple as is compatible with clarity. All panels of a figure should be logically connected and assembled on a single page in a rectangular shape; any essential alignments (parts horizontal, vertical, spacings, etc) should be indicated. Each panel of a multipart figure should be sized so that the whole figure can be proportionally reduced and reproduced on the printed page at the smallest size at which essential details are visible. Nature's standard figure sizes are either 9 or 18 cm wide; the maximum permitted height is 17 cm. Panels should be arranged to fit these widths while minimizing excess space around the panels. Tables should be prepared using the Table menu in Word. As we must be able to edit the figures so that they conform to our house style, the submission of files that are incorrectly formatted, flattened, or of insufficient resolution may delay final acceptance of your manuscript.

All main figures are prepared according to the artwork guidelines and uploaded as PDF files with production quality and editable format.

THIRD PARTY RIGHTS: You must provide proof that you have secured permission to use any third party materials that appear in any part of your manuscript, including Extended Data and Supplementary

Information. Please fill out a Third Party Rights Table, and upload this with the final version of your manuscript. Third party materials include any figures, tables, images, videos or text boxes that are reproductions or adaptations of items that have previously been published elsewhere and/or are owned by a third party. This includes pictures taken by professional photographers, maps and images downloaded from the internet. You will need to obtain the right to use each of these items before your paper can be accepted for publication. You will also need to give proper attribution to the copyright holders in your paper. Please ensure you upload any necessary grants of rights alongside the final version of your manuscript. More information is available on our Rights and permissions page. Failure to obtain the appropriate rights and to supply a completed third party rights table will delay the publication of your article. The editorial assistant (cc'd) can help with any questions.

No third-party materials are used in this study.

COVER ARTWORK: We welcome submissions of artwork for consideration for our cover. More information can be found in our guide for cover artwork. The file name(s) should include the manuscript reference number and be labelled as a cover suggestion; a short description is also preferred. Illustrations should be selected more for their aesthetic appeal than for their scientific content. We cannot promise that your suggestions will be selected for the cover, as competition is intense.

CHEMICAL STRUCTURE PRESENTATION: Any chemical structures in the main text or Extended Data figures must conform to our chemical structure style guide. This guide lists the ChemDraw preferences and stylesheet that must be used to draw all structures. The style and size of chemical structures should not be modified from the default settings in the template, unless absolutely necessary (see the guide for examples), in which case 80% size and 5 pt font is the smallest size possible. Please export any ChemDraw (.cdx) files as a PDF, retaining editing capabilities — we find that 'print to pdf' works well for this — and upload this with your manuscript.

No chemical structure was used in our study or being described in the manuscript.

IMAGE INTEGRITY: We strongly advise that you go carefully through all the data (including Extended Data and Supplementary Information) to ensure there are no accidental image/data duplications, other image manipulations or data errors. Such issues generally require correction after publication. Any image provided for publication, either in print or online (including Extended Data and Supplemental Information), may be subject to a quality control process to check for image integrity and manipulation. A discussion of our standards regarding how images should be prepared and presented can be found here.

In compliance.

EXTENDED DATA: Extended Data do not appear in print but are included online within the full-text HTML and integrated in the downloadable PDF. Extended Data are an integral part of the paper, and only data that directly contribute to the main message should be included. All Extended Data must be referred to in the main text, and their legends should be listed sequentially at the end of the main text file, not in the Extended Data files. Extended Data should be assembled into a maximum of 10 A4 size, multi-panelled display items. They must be supplied as individual files in .jpg, .tif or .eps format **only**. They should be of the same quality as the main figures, but there are important differences in their formatting. More specific instructions are provided here. If you need to describe a complex process, we encourage you to add a schematic of the main finding as part of the Extended Data to aid readers unfamiliar with the immediate discipline.

In compliance.

SUPPLEMENTARY INFORMATION: Supplementary Information (SI) is online-only, peer-reviewed material that is essential background to the study (e.g., large data sets, more complex methods, and calculations), but which is too large or impractical, or of interest only to a few specialists, to justify inclusion

in the print version of the paper (see here for further details). While SI should not typically contain data figures (any figures additional to those appearing in the main text should be formatted as Extended Data), we require that the raw, uncropped data for gels be presented as an SI figure (see below). Tables may be included in SI, but only if they are unsuitable for formatting as Extended Data (e.g., tables containing large data sets that cannot fit a single page or raw data tables that are best suited to Excel files). If a manuscript has SI, each discrete SI item (e.g., videos, tables) must be referred to at an appropriate point in the main text file. You must also provide a Word file entitled "SI Guide", containing a cover page with manuscript title and author information; a table of contents (preferably with page numbers); and then any SI text, notes, figures, and titles and legends for any separate SI files; for additional information see here.

In compliance.

We recommend that you pay careful attention to the formatting of the SI because it is not subedited. After the paper has been accepted, SI files can only be amended for critical changes to the scientific content, not for style.

In compliance.

CELL LINE IDENTIFICATION: To help curb the inadvertent use of cross-contaminated or misidentified cell lines, we ask that you check your reagents against the list of commonly misidentified cell lines maintained by the International Cell Line Authentication Committee, which is also accessible through the NCBI BioSample database. If you have used a cell line that is on this list, you must provide a scientific justification and state the identity issue in the Methods. The editors reserve the right to demand that the data be removed from the paper if the justification is deemed unsatisfactory. In addition, authors must identify the source of cell lines (with catalog number if obtained from a vendor or cell bank) and report whether the cell lines have been authenticated, including the method used, the results, and the date authentication testing was last performed for that cell line. You should be able to provide the test results upon request. Mycoplasma contamination testing status must also be reported. These requirements will be particularly scrutinized for cancer studies, where the issue of cell line misidentification has been well documented. Resources on cell line authentication are available here.

In compliance.

SOURCE DATA (GRAPHS): To increase transparency, we strongly encourage you to provide, in spreadsheet form, the data underlying the graphical representations used in figures. For all experiments presenting data from animal models, this is a requirement and is not optional. This is in addition to our well-established data-deposition policy for specific types of experiments and large datasets. Online readers of the manuscript will be able to access the graphical source data directly from the figure legend. Spreadsheets must be submitted in .xls, .xlsx or .csv formats. One file per figure is permitted. If there is a multi-panelled figure, the source data for each panel should be clearly labeled in the file; alternatively the source data for a figure can be included in multiple, clearly labeled sheets within an Excel file. File sizes of up to 30 MB are permitted, but it is expected that the vast majority of graphical source data files will be considerably smaller than this. When submitting these files with your manuscript, you should select the "Source Data" file type and use the title field in the file description tab to indicate the figure(s) to which the source data pertain. Source data should not be provided as Extended Data.

N/A

RAW DATA (GELS): You must provide the original source images for all data obtained by electrophoretic separation (e.g., EMSA, northern/Southern/western blots, etc). The raw images must be assembled into a single .pdf or .tif file (multiple gels on a single page is encouraged). The file should be uploaded as Supplementary Figure 1. The full scanned images must be in uncropped form and contain labeled size/molecular weight markers and loading controls. There should be an accurate indication of how the gels were cropped for the final figure. The figure legends and raw data files should indicate whether

controls (such as beta-actin) were run on the same gel as loading controls, or on separate gels as sample processing controls (see here for guidance). While the data can be displayed in a relatively informal style, there must be a correspondence between each source data image and a specific main text or Extended Data figure. The main text or Extended Data figure legends should refer to the uncropped scans explicitly (e.g., “For gel source data, see Supplementary Figure 1.”). For examples, see here or here.

In compliance, provided in Supplementary Fig. 9. And it was cited in the legend of Fig. 1 and Extended Data Fig. 1.

DATA DEPOSITION: The following specific points may be relevant to your paper, so please ensure that you provide the following information:

* Sequences for any RNAi/small RNA constructs must be included.

N/A

* Accession numbers for gene expression data or RNA sequencing data must be listed.

N/A

* Papers reporting protein structures must conform to our standards listed in the Guide to Authors. The Data Availability statement must state that the X-ray crystallographic coordinates and structure factor files (or comparable NMR or cryoEM data) have been deposited in the appropriate, named, public database, along with all relevant accession number(s). You must use the standard Nature templates for structural data; there are separate links to tables for X-ray crystallographic, NMR and cryoEM structures. These tables must be presented as Extended Data; if the number of entries causes the table to exceed a page, it must be divided into two Extended Data items. The contour level of any electron density maps presented, as well as the type of map (i.e., Fo-Fc or 2Fo-Fc), should be explicitly stated in the figure legend.

Cryo-EM density maps and atomic structures determined in this study are deposited to EMDB and PDB databases.

* For every new chemical compound, a complete description of the synthesis and the physical characterization (i.e., NMR, MS, etc) must be included in the Supplementary Information (see here).

N/A

* Papers containing new or revised formal taxonomic nomenclature for animals, whether living or extinct, are accepted conditional on the provision of LSIDs (Life Science Identifiers) by means of registration of such nomenclature with ZooBank, the online registration system for the International Code of Zoological Nomenclature (ICZN). ZooBank LSIDs can be resolved and the associated information viewed through any standard web browser by appending the LSID to the prefix "<http://zoobank.org/>".

N/A

* We strongly encourage deposition of 3D morphological data in a suitable repository such as MorphoBank, MorphoSource or similar; the relevant accession numbers should be listed in the Data Availability statement.

N/A

* For animal experiments, you must confirm that all experiments were performed in accordance with relevant guidelines and regulations. There should be a statement identifying the institutional and/or

licensing committee approving the experiments, including any relevant details. Sex and other characteristics of animals that may influence the results must be described. Details of housing and husbandry must be included if they are likely to influence experimental results. Further details can be found here.

N/A

* Human genotype data (e.g., SNP array data) should be deposited into a public database (dbGAP or EGA) with a controlled access policy.

N/A

* A full clinical and pathological characterization of patients/human subjects and samples should be provided in tabular format, including the magnitude of response for each patient (partial, complete, stable disease), the site of the biopsy, whether or not that lesion was progressing and mutational status if appropriate.

N/A

We will not send your revised paper for further review. If the revised paper is in our format (as detailed above), in accessible style and of appropriate length, we shall begin the acceptance process.

To the best of our ability, the final version is in compliance with all instructions here.

In order to accept your paper, we require the following electronic files:

* A cover letter describing your response to any editorial comments and detailing any format changes during revision, particularly if the overall length is affected.

Included.

* A point-by-point response (preferably in Word) to any remaining issues raised by our referees.

Included.

* The final version of your text as a Word document. Word Equation Editor/MathType should be used only for formulae that cannot be produced using normal text or symbol font. If this is not possible, the manuscript can be supplied as a single plain vanilla TeX or LaTeX file that includes all references and abbreviations, with no special formatting, as well as a PDF version that is uploaded as a 'related manuscript file'.

Included.

* Production-quality versions of all figures (see above).

Included.

* The final version of the Extended Data.

Included.

* The final version of any Supplementary Information, presented as one file (ideally a PDF) if feasible, as well as a separate SI Guide.

Included.

* Source Data, if appropriate.

N/A

* For optimal quality videos we encourage H.264 encoding and a standard aspect ratio of 16:9 (4:3 is second best), without compression.

In compliance.

* Completed and signed copies of the following **five (or six) forms**, uploaded as a "Related Manuscript File" file type:

- 1) Biology editorial checklist; completed and included
- 2) Manuscript checklist; completed and included
- 3) Reporting summary; completed and included
- 4) Editorial policy checklist; completed and included
- 5) Third-party rights table; N/A, No third-party items used in this study.
- 6) Code and software submission checklist (if applicable). completed and included

Nature has now transitioned to a unified Rights Collection system which will allow our author services team to quickly and easily collect the rights and permissions required to publish your work. Once your paper is accepted, you will receive an email in approximately 10 business days providing you with a link to complete the grant of rights. If you choose to publish Open Access, our author services team will also be in touch at that time regarding any additional information that may be required to arrange payment for your article. If you have any questions please contact asjournals@springernature.com.

You may need to take specific actions to achieve compliance with funder and institutional open access mandates. If your research is supported by a funder that requires immediate open access (e.g. according to Plan S principles) then you should select the gold OA route, and we will direct you to the compliant route where possible. If you select the subscription publication route our standard licensing terms will need to be accepted, including our self-archiving policies. Those standard licensing terms will supersede any other terms that you or any third party may assert apply to any version of the manuscript.